# TRAINING NEURAL NETWORKS FOR AND BY INTERPOLATION

## ABSTRACT

In modern supervised learning, many deep neural networks are able to interpolate the data: the empirical loss can be driven to near zero on all samples simultaneously. In this work, we explicitly exploit this interpolation property for the design of a new optimization algorithm for deep learning. Specifically, we use it to compute an adaptive learning-rate in closed form at each iteration. This results in the Adaptive Learning-rates for Interpolation with Gradients (ALI-G) algorithm. ALI-G retains the main advantage of SGD which is a low computational cost per iteration. But unlike SGD, the learning-rate of ALI-G uses a single constant hyper-parameter and does not require a decay schedule, which makes it considerably easier to tune. We provide convergence guarantees of ALI-G in the stochastic convex setting. Notably, all our convergence results tackle the realistic case where the interpolation property is satisfied up to some tolerance. We provide experiments on a variety of architectures and tasks: (i) learning a differentiable neural computer; (ii) training a wide residual network on the SVHN data set; (iii) training a Bi-LSTM on the SNLI data set; and (iv) training wide residual networks and densely connected networks on the CIFAR data sets. ALI-G produces state-of-the-art results among adaptive methods, and even yields comparable performance with SGD, which requires manually tuned learning-rate schedules. Furthermore, ALI-G is simple to implement in any standard deep learning framework and can be used as a drop-in replacement in existing code.

## 1 INTRODUCTION

Training a deep neural network is a challenging optimization problem: it involves minimizing the average of many high-dimensional non-convex functions. In practice, the main algorithms of choice are Stochastic Gradient Descent (SGD) (Robbins & Monro, 1951) and adaptive gradient methods such as AdaGrad (Duchi et al., 2011) or Adam (Kingma & Ba, 2015). In recent work, the ability to interpolate – i.e. to achieve near zero loss on all training samples simultaneously – has been used to show convergence of SGD (Ma et al., 2018, Vaswani et al., 2019a, Zhou et al., 2019). This property is usually satisfied in supervised deep learning because of the empirical success of over-parameterized architectures. However, while the convergence analyses provide a better theoretical understanding of SGD, they do not help improve its practical behavior.

In this work, we open a different line of enquiry, namely: *can the interpolation property be used to design a robust and efficient optimization algorithm for deep learning?* In order to answer this question, we begin by giving the following two desiderata of an optimization algorithm for deep learning: (i) an inexpensive computational cost per iteration (typically a call to a stochastic first-order oracle); and (ii) adaptive learning-rates that do not require a manually designed schedule.

We present ALI-G (Adaptive Learning-rates for Interpolation with Gradients), an algorithm that takes advantage of interpolation by design and satisfies both properties mentioned above. Key to the ALI-G algorithm are the following two ideas. First, an adaptive learning-rate can be computed for the non-stochastic gradient direction when the minimum value of the objective function is known (Polyak, 1969, Shor, 1985, Brännlund, 1995, Nedić & Bertsekas, 2001a;b). And second, one such minimum value is usually approximately known for interpolating models: for instance, it is close to zero for a model trained with the cross-entropy loss. By carefully combining these two ideas,

we create a stochastic algorithm that provably converges fast in the convex setting and that obtains state-of-the-art results with neural networks.

Procedurally, ALI-G is close to many existing algorithms, such as Deep Frank-Wolfe (Berrada et al., 2019), APROX (Asi & Duchi, 2019) and $L_4$ (Rolinek & Martius, 2018). And yet uniquely, thanks to its careful design and analysis, ALI-G enables accurate optimization of a wide class of deep neural networks using only a single hyper-parameter that does not need to be decayed. This makes ALI-G well-suited to the deep learning setting, where hyper-parameter tuning is widely regarded as an onerous and time consuming task. Since ALI-G is easy to implement in any deep learning framework, we believe that it can prove to be a practical and reliable optimization tool for deep learning.

**Contributions.** We summarize the contributions of this work as follows:
- We design an optimization algorithm that uses a single hyper-parameter for its learning-rate and does need any decaying schedule. In contrast, the closely related APROX (Asi & Duchi, 2019) and $L_4$ (Rolinek & Martius, 2018) use respectively two and four hyper-parameters for their learning-rate.
- We provide convergence rates of ALI-G in various stochastic convex settings. Importantly, our theoretical results take into account the error in the estimate of the minimum objective value. To the best of our knowledge, our work is the first to establish convergence rates for interpolation with approximate estimates.
- We demonstrate state-of-the-art results for ALI-G on learning a differentiable neural computer; training variants of residual networks on the SVHN and CIFAR data sets; and training a Bi-LSTM on the Stanford Natural Language Inference data set.

## 2 THE ALGORITHM

### 2.1 PROBLEM SETTING

**Loss Function.** We consider a supervised learning task where the model is parameterized by $\mathbf{w} \in \mathbb{R}^p$. Usually, the objective function can be expressed as an expectation over $z \in \mathcal{Z}$, a random variable indexing the samples of the training set:

$$f(\mathbf{w}) \triangleq \mathbb{E}_{z \in \mathcal{Z}}[\ell_z(\mathbf{w})], \tag{1}$$

where each $\ell_z$ is the loss function associated with the sample $z$. We assume that each $\ell_z$ is non-negative, which is the case for the large majority of loss functions used in machine learning. For instance, suppose that the model is a deep neural network with weights $\mathbf{w}$ performing classification. Then for each sample $z$, $\ell_z(\mathbf{w})$ can represent the cross-entropy loss, which is always non-negative. Other non-negative loss functions include the structured or multi-class hinge loss, and the $\mathcal{L}_1$ or $\mathcal{L}_2$ loss functions for regression.

**Regularization.** It is often desirable to employ a regularization function $\phi$ in order to promote generalization. In this work, we incorporate such regularization as a constraint on the feasible domain: $\Omega = \{\mathbf{w} \in \mathbb{R}^p : \phi(\mathbf{w}) \leq r\}$ for some value of $r$. In the deep learning setting, this will allow us to assume that the objective function can be driven close to zero without unrealistic assumptions about the regularization. Our framework can handle any constraint set $\Omega$ on which Euclidean projections are computationally efficient. This includes the feasible set induced by $\mathcal{L}_2$ regularization: $\Omega = \{\mathbf{w} \in \mathbb{R}^p : \|\mathbf{w}\|_2^2 \leq r\}$, for which the projection is given by a simple rescaling of $\mathbf{w}$. Finally, note that if we do not wish to use any regularization, we define $\Omega = \mathbb{R}^p$ and the corresponding projection is the identity.

**Problem Formulation.** The learning task can be expressed as the problem $(\mathcal{P})$ of finding a feasible vector of parameters $\mathbf{w}_\star \in \Omega$ that minimizes $f$:

$$\mathbf{w}_\star \in \underset{\mathbf{w} \in \Omega}{\operatorname{argmin}} f(\mathbf{w}). \tag{$\mathcal{P}$}$$

Also note that $f_\star$ refers to the minimum value of $f$ over $\Omega$: $f_\star \triangleq \min_{\mathbf{w} \in \Omega} f(\mathbf{w})$.

### 2.2 THE POLYAK STEP-SIZE

Before outlining the ALI-G algorithm, we begin with a brief description of the Polyak step-size, from which ALI-G draws some fundamental ideas.

**Setting.** We assume that $f_\star$ is known and we use non-stochastic updates: at each iteration, the full objective $f$ and its derivative are evaluated. We denote by $\nabla f(\mathbf{w})$ the first-order derivative of $f$ at $\mathbf{w}$ (e.g. $\nabla f(\mathbf{w})$ can be a sub-gradient or the gradient). In addition, $\|\cdot\|$ is the standard Euclidean norm in $\mathbb{R}^p$, and $\Pi_\Omega(\mathbf{w})$ is the Euclidean projection of the vector $\mathbf{w} \in \mathbb{R}^p$ on the set $\Omega$.

**Polyak Step-Size.** At time-step $t$, using the Polyak step-size (Polyak, 1969, Shor, 1985, Brännlund, 1995, Nedić & Bertsekas, 2001a;b) yields the following update:

$$\mathbf{w}_{t+1} = \Pi_\Omega\left(\mathbf{w}_t - \gamma_t \nabla f(\mathbf{w}_t)\right), \text{ where } \gamma_t \triangleq \frac{f(\mathbf{w}_t) - f_\star}{\|\nabla f(\mathbf{w}_t)\|^2}, \tag{2}$$

where we loosely define $\frac{0}{0} = 0$ for simplicity purposes.

**Interpretation.** It can be shown that $\mathbf{w}_{t+1}$ lies on the intersection between the linearization of $f$ at $\mathbf{w}_t$ and the horizontal plane $z = f_\star$ (see Figure 1, more details in Proposition 2 in the supplementary material). Note that since $f_\star$ is the minimum of $f$, the Polyak step-size $\gamma_t$ is necessarily non-negative.

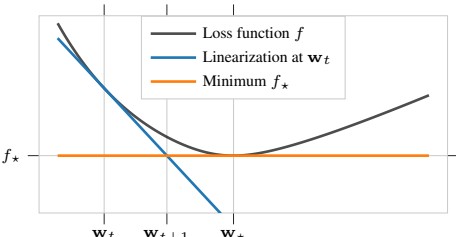

**Limitations.** Equation (2) has two major short-comings that prevent its applicability in a machine learning setting. First, each update requires a full evaluation of $f$ and its derivative. Stochastic extensions have been proposed in Nedić & Bertsekas (2001a;b), but they still require frequent evaluations of $f$. This is expensive in the large data setting, and even computationally infeasible when using massive data augmentation. Second, when applying this method to the non-convex setting of deep neural networks, the method sometimes fails to converge.

Figure 1: *Illustration of the Polyak step-size in 1D. In this case, and further assuming that $f_\star = 0$, the algorithm coincides with the Newton-Raphson method for finding roots of a function.*

Therefore we would like to design an extension of the Polyak step-size that (i) is inexpensive to compute in a stochastic setting (e.g. with a computational cost that is independent of the total number of training samples), and (ii) converges in practice when used with deep neural networks. The next section introduces the ALI-G algorithm, which achieves these two goals in the interpolation setting.

## 2.3 The ALI-G Algorithm

We now present the ALI-G algorithm. For this, we suppose that we are in an interpolation setting: the model is assumed to be able to drive the loss function to near zero on all samples simultaneously.

**Algorithm.** The main steps of the ALI-G algorithm are provided in Algorithm 1. ALI-G iterates over three operations until convergence. First, it computes a stochastic approximation of the learning objective and its derivative (line 3). Second, it computes a step-size decay parameter $\gamma_t$ based on the stochastic information (line 4). Third, it updates the parameters by moving in the negative derivative direction by an amount specified by the step-size and projecting the resulting vector on to the feasible region (line 5).

---

**Algorithm 1** *The ALI-G algorithm*

---

**Require:** maximal learning-rate $\eta$, initial feasible $\mathbf{w}_0 \in \Omega$, small constant $\delta > 0$
 1: $t = 0$
 2: **while** not converged **do**
 3:     Get $\ell_{z_t}(\mathbf{w}_t)$, $\nabla \ell_{z_t}(\mathbf{w}_t)$ with $z_t$ drawn i.i.d.
 4:     $\gamma_t = \min\left\{\frac{\ell_{z_t}(\mathbf{w}_t)}{\|\nabla \ell_{z_t}(\mathbf{w}_t)\|^2 + \delta}, \eta\right\}$
 5:     $\mathbf{w}_{t+1} = \Pi_\Omega\left(\mathbf{w}_t - \gamma_t \nabla \ell_{z_t}(\mathbf{w}_t)\right)$
 6:     $t = t + 1$
 7: **end while**

---

**Comparison with the Polyak Step-Size.** There are three main differences to the update in equation (2). First, each update only uses the loss $\ell_{z_t}$ and its derivative rather than the full objective $f$ and its derivative. Second, the learning-rate $\gamma_t$ is clipped to $\eta$, the maximal learning-rate hyper-parameter. We emphasize that $\eta$ remains constant throughout the iterations, therefore it is a single hyper-parameter and does not need a schedule like SGD learning-rate. Third, the minimum $f_\star$ has been replaced by the lower-bound of $0$. All these modifications will be justified in the next section.

**The ALI-G$^\infty$ Variant.** When ALI-G uses no maximal learning-rate, we refer to the algorithm as ALI-G$^\infty$, since it is equivalent to use an infinite maximal learning-rate. Note that ALI-G$^\infty$ requires no hyper-parameter for its step-size.

## 3   JUSTIFICATION AND ANALYSIS

### 3.1   INTERPOLATION ENABLES INEXPENSIVE STOCHASTIC UPDATES

By definition, the interpolation setting gives $f_\star = 0$, which we used in ALI-G to simplify the formula of the learning-rate $\gamma_t$. More subtly, the interpolation property also allows the updates to rely on the stochastic estimate $\ell_{z_t}(\mathbf{w}_t)$ rather than the exact but expensive $f(\mathbf{w}_t)$. Intuitively, this is possible because in the interpolation setting, each training sample can use its own learning-rate without harming progress on the other ones. Recall that ALI-G$^\infty$ is the variant of ALI-G that uses no maximal learning-rate. The following result formalizes the convergence guarantee of ALI-G$^\infty$ in the stochastic convex setting:

**Theorem 1** (Convex and Lipschitz)**.** *We assume that $\Omega$ is a convex set, and that for every $z \in \mathcal{Z}$, $\ell_z$ is convex and $C$-Lipschitz. Let $\mathbf{w}_\star$ be a solution of ($\mathcal{P}$), and assume that the interpolation property is approximately satisfied: $\forall z \in \mathcal{Z}$, $\ell_z(\mathbf{w}_\star) \leq \varepsilon$, for some interpolation tolerance $\varepsilon \geq 0$. Then ALI-G$^\infty$ applied to $f$ satisfies:*

$$f\left(\frac{1}{T+1}\sum_{t=0}^{T}\mathbf{w}_t\right) \leq \varepsilon\sqrt{\left(\frac{C^2}{\delta}+1\right)} + \frac{\|\mathbf{w}_0 - \mathbf{w}_\star\|\sqrt{C^2+\delta}}{\sqrt{T+1}}. \tag{3}$$

In other words, by assuming interpolation, ALI-G provably converges while requiring only $\ell_{z_t}(\mathbf{w}_t)$ and $\nabla\ell_{z_t}(\mathbf{w}_t)$ (stochastic estimation per sample) to compute its learning-rate. In contrast, the Polyak step-size, which does not exploit interpolation, would require $f(\mathbf{w}_t)$ and $\nabla f(\mathbf{w}_t)$ to compute the learning-rate (deterministic computation over all training samples). This constitutes a major computational advantage of ALI-G over the usual Polyak step-size.

We emphasize that in Theorem 1, our careful analysis explicitly shows the dependency of the convergence result on the interpolation tolerance $\varepsilon$. It is reassuring to note that convergence is exact when the interpolation property is exactly satisfied ($\varepsilon = 0$).

In the supplementary material, we also establish convergence rates of $\mathcal{O}(1/T)$ for smooth convex functions, and $\mathcal{O}(\exp(-\alpha T/8\beta))$ for $\alpha$-strongly convex and $\beta$-smooth functions. Similar results can be proved when using a maximal learning-rate $\eta$: the convergence speed then remains unchanged provided that $\eta$ is large enough, and it is lowered when $\eta$ is small. We refer the interested reader to the supplementary for the formal results and their proofs.

### 3.2   A MAXIMAL LEARNING-RATE HELPS WITH NON-CONVEXITY

The Polyak step-size may fail to converge when the objective is non-convex, as figure 2 illustrates: in this (non-convex) setting, gradient descent with Polyak step-size oscillates between two symmetrical points because its step-size is too large (details in the supplementary).

Indeed, we empirically find that non-convexity usually leads to overestimation of the Polyak step-size. Intuitively, using a maximal learning-rate allows ALI-G to behave like constant step-size SGD in non-convex regions where the Polyak step-size would be over-estimated, and to automatically use the Polyak step-size once reaching a convex basin of convergence.

Importantly, using a maximal learning-rate can be seen as a very natural extension of SGD when using a non-negative loss function:

**Proposition 1.** *[Proximal Interpretation] Suppose that $\Omega = \mathbb{R}^p$ and let $\delta = 0$. We consider the update performed by SGD: $\mathbf{w}_{t+1}^{SGD} = \mathbf{w}_t - \eta_t \nabla \ell_{z_t}(\mathbf{w}_t)$; and the update performed by ALI-G: $\mathbf{w}_{t+1}^{ALI\text{-}G} = \mathbf{w}_t - \gamma_t \nabla \ell_{z_t}(\mathbf{w}_t)$, where $\gamma_t = \min\left\{\frac{\ell_{z_t}(\mathbf{w}_t)}{\|\nabla \ell_{z_t}(\mathbf{w}_t)\|^2 + \delta}, \eta\right\}$. Then we have:*

$$\mathbf{w}_{t+1}^{SGD} = \operatorname*{argmin}_{\mathbf{w} \in \mathbb{R}^p} \left\{ \frac{1}{2\eta_t} \|\mathbf{w} - \mathbf{w}_t\|^2 + \ell_{z_t}(\mathbf{w}_t) + \nabla \ell_{z_t}(\mathbf{w}_t)^\top (\mathbf{w} - \mathbf{w}_t) \right\}, \tag{4}$$

$$\mathbf{w}_{t+1}^{ALI\text{-}G} = \operatorname*{argmin}_{\mathbf{w} \in \mathbb{R}^p} \left\{ \frac{1}{2\eta} \|\mathbf{w} - \mathbf{w}_t\|^2 + \max\left\{ \ell_{z_t}(\mathbf{w}_t) + \nabla \ell_{z_t}(\mathbf{w}_t)^\top (\mathbf{w} - \mathbf{w}_t), 0 \right\} \right\}. \tag{5}$$

In other words, at each iteration, ALI-G solves a proximal problem in closed form in a similar way to SGD. In both cases, the loss function $\ell_{z_t}$ is locally approximated by a first-order Taylor expansion at $\mathbf{w}_t$. The difference is that ALI-G also exploits the fact that $\ell_{z_t}$ is non-negative. This allows ALI-G to use a constant value for $\eta$ in the interpolation setting, while the learning-rate $\eta_t$ of SGD needs to be manually decayed.

It is currently an open question whether ALI-G can be proved to converge in the stochastic non-convex setting given a sufficiently small yet constant maximal learning-rate $\eta$.

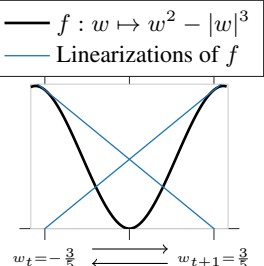

Figure 2: *A simple example where the Polyak step-size oscillates due to non-convexity. On this problem, ALI-G converges whenever its maximal learning-rate $\eta$ is lower than 10.*

## 4 RELATED WORK

**Interpolation in Deep Learning.** As mentioned in the introduction, recent works have successfully exploited the interpolation assumption to prove convergence of SGD in the context of deep learning (Ma et al., 2018, Vaswani et al., 2019a, Zhou et al., 2019). Such works are complementary to ours in the sense that they provide a convergence analysis of an existing algorithm for deep learning.

**Adaptive Gradient Methods.** Similarly to ALI-G, adaptive gradient methods also rely on tuning a single hyper-parameter, thereby providing a more pragmatic alternative to SGD that needs a specification of the full learning-rate schedule. While the most popular ones are Adagrad (Duchi et al., 2011), RMSPROP (Tieleman & Hinton, 2012), Adam (Kingma & Ba, 2015) and AMSGrad (Reddi et al., 2018), there have been many other variants (Zeiler, 2012, Orabona & Pál, 2015, Défossez & Bach, 2017, Levy, 2017, Mukkamala & Hein, 2017, Zheng & Kwok, 2017, Bernstein et al., 2018, Chen & Gu, 2018, Shazeer & Stern, 2018, Zaheer et al., 2018, Chen et al., 2019, Loshchilov & Hutter, 2019, Luo et al., 2019). However, as pointed out in Wilson et al. (2017), adaptive gradient methods tend to give poor generalization in supervised learning. In our experiments, the results provided by ALI-G are significantly better than those obtained by the most popular adaptive gradient methods.

**Adaptive Learning-Rate Algorithms.** Vaswani et al. (2019b) show that one can use line search in a stochastic setting for interpolating models while guaranteeing convergence. However, in contrast to our work, the resulting algorithm requires more than one hyper-parameter (up to four), and the line-search is not computed in closed form. Less closely related methods have proposed adaptive learning-rates without using the minimum for the computation of the learning rate (Schaul et al., 2013, Tan et al., 2016, Zhang et al., 2017, Baydin et al., 2018, Wu et al., 2018, Li & Orabona, 2019), but they have not demonstrated competitiveness against SGD with a well-tuned hand-designed schedule.

$L_4$ **Algorithm.** The $L_4$ algorithm (Rolinek & Martius, 2018) also uses a modified version of the Polyak step-size. However, the $L_4$ algorithm computes an online estimate of $f_\star$ rather than relying on a fixed value. This requires three hyper-parameters, which are in practice sensitive to noise and crucial for empirical convergence of the method. In addition, $L_4$ does not come with convergence guarantees. In contrast, by utilizing the interpolation property and a maximal learning-rate, our method is able to (i) provide reliable and accurate minimization with only a single hyper-parameter, and (ii) offer guarantees of convergence in the stochastic convex setting.

**Frank-Wolfe Methods.**   The proximal interpretation in Proposition 1 allows us to draw additional parallels to existing methods. In particular, the formula of the learning-rate $\gamma_t$ may remind the reader of the Frank-Wolfe algorithm (Frank & Wolfe, 1956) in some of its variants (Locatello et al., 2017), or other dual methods (Lacoste-Julien et al., 2013, Shalev-Shwartz & Zhang, 2016). This is because such methods solve in closed form the dual of problem (5), and problems in the form of (5) naturally appear in dual coordinate ascent methods (Shalev-Shwartz & Zhang, 2016).

When no regularization is used, ALI-G and Deep Frank-Wolfe (DFW) (Berrada et al., 2019) are procedurally identical algorithms. This is because in such a setting, one iteration of DFW also amounts to solving (5) in closed-form – more generally, DFW is designed to train deep neural networks by solving proximal linear support vector machine problems approximately. However, we point out the two fundamental advantages of ALI-G over DFW: (i) ALI-G can handle arbitrary (lower-bounded) loss functions, while DFW can only use convex piece-wise linear loss functions; and (ii) as seen previously, ALI-G provides convergence guarantees in the convex setting.

**SGD with Polyak's Learning-Rate.**   Oberman & Prazeres (2019) extend the Polyak step-size to rely on a stochastic estimation of the gradient $\nabla \ell_{z_t}(\mathbf{w}_t)$ only, instead of the expensive deterministic gradient $\nabla f(\mathbf{w}_t)$. However, they still require to evaluate $f(\mathbf{w}_t)$, the objective function over the entire training data set, in order to compute its learning-rate, which makes the method impractical. In addition, since they do not do exploit the interpolation setting nor the fact that regularization can be expressed as a constraint, they also require the optimal objective function value $f_\star$ in advance.

APROX **Algorithm.**   Asi & Duchi (2019) have recently introduced the APROX algorithm, a family of proximal stochastic optimization algorithms for convex problems. Notably, the APROX "truncated model" version is similar to ALI-G. However, there are four clear advantages of our work over (Asi & Duchi, 2019) in the interpolation setting, in particular for training neural networks. First, our work is the first to empirically demonstrate the applicability and usefulness of the algorithm on varied modern deep learning tasks – most of our experiments use several orders of magnitude more data and model parameters than the small-scale convex problems of (Asi & Duchi, 2019). Second, our analysis and insights allow us to make more aggressive choices of learning rate than (Asi & Duchi, 2019). Indeed, Asi & Duchi (2019) assume that the maximal learning-rate is exponentially decaying, even in the interpolating convex setting. In contrast, by avoiding the need for an exponential decay, the learning-rate of ALI-G requires only one hyper-parameters instead of two for APROX. Third, our analysis takes into account the interpolation tolerance $\varepsilon \geq 0$ rather than unrealistically assuming the perfect case $\varepsilon = 0$ (that would require infinite weights when using the cross-entropy loss for instance). Fourth, our analysis proves fast convergence in function space rather than iterate space.

## 5   EXPERIMENTS

We empirically compare ALI-G to the optimization algorithms most commonly used in deep learning. Our experiments span a variety of architectures and tasks: (i) learning a differentiable neural computer; (ii) training wide residual networks on SVHN; (iii) training a Bi-LSTM on the Stanford Natural Language Inference data set; and (iv) training wide residual networks and densely connected networks on the CIFAR data sets. Note that the tasks of training wide residual networks on SVHN and CIFAR-100 are part of the DeepOBS benchmark Schneider et al. (2019), which aims at standardizing baselines for deep learning optimizers. In particular, these tasks are among the most difficult ones of the benchmark because the SGD baseline benefits from a manual schedule for the learning rate. Despite this, ALI-G obtains competitive performance with SGD. In addition, ALI-G is the best performing method with a single hyper-parameter on the difficult tasks of Bi-LSTM on SNLI and ResNet variants on CIFAR.

The code to reproduce our results will be made publicly available. In the Tensorflow (Abadi et al., 2015) experiment, we use the official and publicly available implementation of $L_4$[1]. In the PyTorch (Paszke et al., 2017) experiments, we use our implementation of $L_4$, which we unit-test against the official Tensorflow implementation. In addition, we employ the official implementation of DFW[2] and

---

[1]`https://github.com/martius-lab/l4-optimizer`
[2]`https://github.com/oval-group/dfw`

we re-use their code for the experiments on SNLI and CIFAR. All experiments are performed either on a 12-core CPU (differentiable neural computer) or on a single GPU (SVHN, SNLI, CIFAR).

## 5.1 DIFFERENTIABLE NEURAL COMPUTERS

**Setting.** The Differentiable Neural Computer (DNC) (Graves et al., 2016) is a recurrent neural network that aims at performing computing tasks by learning from examples rather than by executing an explicit program. In this case, the DNC learns to repeatedly copy a fixed size string given as input. Although this learning task is relatively simple, the complex architecture of the DNC makes it an interesting benchmark problem for optimization algorithms.

**Methods.** We use the official and publicly available implementation of DNC[3]. We vary the initial learning rate as powers of ten between $10^{-4}$ and $10^4$ for each method except for $L_4$Adam and $L_4$Mom. For $L_4$Adam and $L_4$Mom, since the main hyper-parameter $\alpha$ is designed to lie in $(0, 1)$, we vary it between $0.05$ and $0.095$ with a step of $0.1$. The gradient norm is clipped for all methods except for ALI-G, $L_4$Adam and $L_4$Mom (as recommended by Rolinek & Martius (2018)).

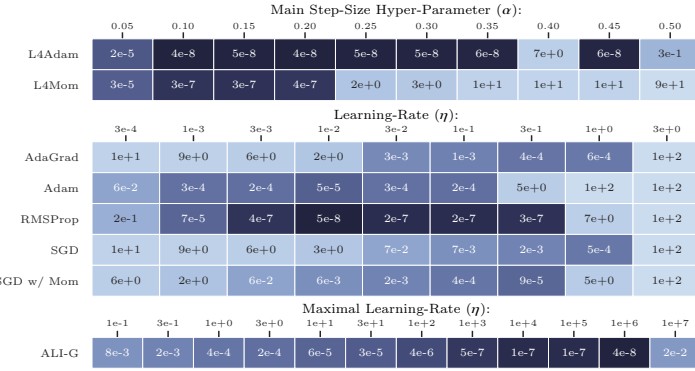

| Main Step-Size Hyper-Parameter ($\alpha$): | 0.05 | 0.10 | 0.15 | 0.20 | 0.25 | 0.30 | 0.35 | 0.40 | 0.45 | 0.50 |
|---|---|---|---|---|---|---|---|---|---|---|
| L4Adam | 2e-5 | 4e-8 | 5e-8 | 4e-8 | 5e-8 | 5e-8 | 6e-8 | 7e+0 | 6e-8 | 3e-1 |
| L4Mom | 3e-5 | 3e-7 | 3e-7 | 4e-7 | 2e+0 | 3e+0 | 1e+1 | 1e+1 | 1e+1 | 9e+1 |

| Learning-Rate ($\eta$): | 3e-4 | 1e-3 | 3e-3 | 1e-2 | 3e-2 | 1e-1 | 3e-1 | 1e+0 | 3e+0 |
|---|---|---|---|---|---|---|---|---|---|
| AdaGrad | 1e+1 | 9e+0 | 6e+0 | 2e+0 | 3e-3 | 1e-3 | 4e-4 | 6e-4 | 1e+2 |
| Adam | 6e-2 | 3e-4 | 2e-4 | 5e-5 | 3e-4 | 2e-4 | 5e+0 | 1e+2 | 1e+2 |
| RMSProp | 2e-1 | 7e-5 | 4e-7 | 5e-8 | 2e-7 | 2e-7 | 3e-7 | 7e+0 | 1e+2 |
| SGD | 1e+1 | 9e+0 | 6e+0 | 3e+0 | 7e-2 | 7e-3 | 2e-3 | 5e-4 | 1e+2 |
| SGD w/ Mom | 6e+0 | 2e+0 | 6e-2 | 6e-3 | 2e-3 | 4e-4 | 9e-5 | 5e+0 | 1e+2 |

| Maximal Learning-Rate ($\eta$): | 1e-1 | 3e-1 | 1e+0 | 3e+0 | 1e+1 | 3e+1 | 1e+2 | 1e+3 | 1e+4 | 1e+5 | 1e+6 | 1e+7 |
|---|---|---|---|---|---|---|---|---|---|---|---|---|
| ALI-G | 8e-3 | 2e-3 | 4e-4 | 2e-4 | 6e-5 | 3e-5 | 4e-6 | 5e-7 | 1e-7 | 1e-7 | 4e-8 | 2e-2 |

Figure 3: *Final objective function when training a Differentiable Neural Computer for $10k$ steps (lower is better). The intensity of each cell is log-proportional to the value of the objective function (darker is better). ALI-G obtains good performance for a very large range of $\eta$ ($10^{-1} \leq \eta \leq 10^6$).*

**Results.** We present the results in Figure 3. ALI-G provides accurate optimization for any $\eta$ within $[10^{-1}, 10^6]$, and is among the best performing methods by reaching an objective function of $4.10^{-8}$. On this task, RMSProp, $L_4$Adam and $L_4$Mom also provide accurate and robust optimization. In contrast to ALI-G and the $L_4$ methods, the most commonly used algorithms such as SGD, SGD with momentum and Adam are very sensitive to their main learning-rate hyper-parameter. Note that the difference between well-performing methods is not significant here because these reach the numerical precision limit of single-precision float numbers.

## 5.2 WIDE RESIDUAL NETWORKS ON SVHN

**Setting.** The SVHN data set contains 73k training samples, 26k testing samples and 531k additional easier samples. From the 73k difficult training examples, we select 6k samples for validation; we use all remaining (both difficult and easy) examples for training, for a total of 598k samples. We train a wide residual network 16-4 following Zagoruyko & Komodakis (2016).

**Method.** For SGD, we use the manual schedule for the learning rate of Zagoruyko & Komodakis (2016). For $L_4$Adam and $L_4$Mom, we cross-validate the main learning-rate hyper-parameter $\alpha$ to be in $\{0.0015, 0.015, 0.15\}$ ($0.15$ is the value recommended by Rolinek & Martius (2018)). For other methods, the learning rate hyper-parameter is tuned as a power of 10. The $\mathcal{L}_2$ regularization is cross-validated in $\{0.0001, 0.0005\}$ for all methods but ALI-G. For ALI-G, the regularization is expressed as a constraint on the $\mathcal{L}_2$-norm of the parameters, and its maximal value is set to 50. SGD,

---

[3]https://github.com/deepmind/dnc

ALI-G and BPGrad use a Nesterov momentum of 0.9. All methods use a dropout rate of 0.4 and a fixed budget of 160 epochs, following (Zagoruyko & Komodakis, 2016).

**Results.** The results are presented in Table 1. On this relatively easy task, most methods achieve about 98% test accuracy. Despite the cross-validation, $L_4$Mom does not converge on this task. Even though SGD benefits from a hand-designed schedule, ALI-G and other adaptive methods obtain close performance to it.

| Adagrad | Adam | AMSGrad | BPGrad | DFW | $L_4$Adam | $L_4$Mom | ALI-G | SGD | SGD$^\dagger$ |
|---------|------|---------|--------|-----|-----------|----------|-------|-----|------|
| 98.0 | 97.9 | 97.9 | 98.1 | 98.1 | **98.2** | 19.6 | 98.1 | 98.3 | 98.4 |

Table 1: *Test Accuracy (%) on SVHN. In red, SGD benefits from a hand-designed schedule for its learning-rate. In black, adaptive methods, including ALI-G, have a single hyper-parameter for their learning-rate. $SGD^\dagger$ refers to the performance reported by Zagoruyko & Komodakis (2016).*

## 5.3 BI-LSTM ON SNLI

**Setting.** We train a Bi-LSTM of 47M parameters on the Stanford Natural Language Inference (SNLI) data set (Bowman et al., 2015). The SNLI data set consists in 570k pairs of sentences, with each pair labeled as entailment, neutral or contradiction. This large scale data set is commonly used as a pre-training corpus for transfer learning to many other natural language tasks where labeled data is scarcer (Conneau et al., 2017) – much like ImageNet is used for pre-training in computer vision. We follow the protocol of Berrada et al. (2019); we also re-use their code and results for the baselines.

**Method.** For $L_4$Adam and $L_4$Mom, the main hyper-parameter $\alpha$ is cross-validated in $\{0.015, 0.15\}$ – compared to the recommended value of 0.15, this helped convergence and considerably improved performance. The SGD algorithm benefits from a hand-designed schedule, where the learning-rate is decreased by 5 when the validation accuracy does not improve. Other methods use adaptive learning-rates and do not require such schedule. The value of the main hyper-parameter $\eta$ is cross-validated as a power of ten for the ALI-G algorithm and for previously reported adaptive methods. Following the implementation by Conneau et al. (2017), no $\mathcal{L}_2$ regularization is used. The algorithms are evaluated with the Cross-Entropy (CE) loss and the multi-class hinge loss (SVM), except for DFW which is designed for SVMs only. For all optimization algorithms, the model is trained for 20 epochs, following (Conneau et al., 2017).

| Loss | Adagrad$^*$ | Adam$^*$ | AMSGrad$^*$ | BPGrad$^*$ | DFW$^*$ | $L_4$Adam | $L_4$Mom | ALI-G$^\infty$ | ALI-G | SGD$^*$ | SGD$^\dagger$ |
|------|---------|------|---------|--------|-----|-----------|----------|--------|-------|------|------|
| CE | 83.8 | 84.5 | 84.2 | 83.6 | - | 83.3 | 83.7 | 84.6 | **84.8** | 84.7 | 84.5 |
| SVM | 84.6 | 85.0 | 85.1 | 84.2 | **85.2** | 82.5 | 83.2 | 84.7 | **85.2** | 85.2 | - |

Table 2: *Test Accuracy (%) on SNLI. In red, SGD benefits from a hand-designed schedule for its learning-rate. In black, adaptive methods have a single hyper-parameter for their learning-rate. In blue, ALI-G$^\infty$ does not have any hyper-parameter for its learning-rate. With an SVM loss, DFW and ALI-G are procedurally identical algorithms – but in contrast to DFW, ALI-G can also employ the CE loss. Methods in the format $X^*$ re-use results from Berrada et al. (2019). $SGD^\dagger$ is the result from Conneau et al. (2017).*

**Results.** We present the results in Table 2. ALI-G$^\infty$ is the only method that requires no hyper-parameter for its learning-rate. Despite this, and the fact that SGD employs a learning-rate schedule that has been hand designed for good validation performance, ALI-G$^\infty$ is still able to obtain results that are competitive with SGD. Moreover, ALI-G, which requires a single hyper-parameter for the learning-rate, outperforms all other methods for both the SVM and the CE loss functions.

## 5.4 WIDE RESIDUAL NETWORKS AND DENSELY CONNECTED NETWORKS ON CIFAR

**Setting.** We follow the methodology of Berrada et al. (2019); we also re-use their code and we reproduce their results. We test two architectures: a Wide Residual Network (WRN) 40-4 (Zagoruyko & Komodakis, 2016) and a bottleneck DenseNet (DN) 40-40 (Huang et al., 2017). We use 45k samples for training and 5k for validation. The images are centered and normalized per channel. We apply standard data augmentation with random horizontal flipping and random crops. AMSGrad was selected in Berrada et al. (2019) because it was the best adaptive method on similar tasks, outperforming in particular Adam and Adagrad. In addition to the baselines from Berrada et al. (2019), we also provide the performance of $L_4$Adam, $L_4$Mom, AdamW (Loshchilov & Hutter, 2019) and Yogi (Zaheer et al., 2018).

**Method.** All optimization methods employ the cross-entropy loss, except for the DFW algorithm, which is designed to use an SVM loss. For DN and WRN respectively, SGD uses the manual learning rate schedules from Huang et al. (2017) and Zagoruyko & Komodakis (2016). Following Berrada et al. (2019), the batch-size is cross-validated in $\{64, 128, 256\}$ for the DN architecture, and $\{128, 256, 512\}$ for the WRN architecture. For $L_4$Adam and $L_4$Mom, the learning-rate hyper-parameter $\alpha$ is cross-validated in $\{0.015, 0.15\}$. For AMSGrad, AdamW, Yogi, DFW and ALI-G, the learning-rate hyper-parameter $\eta$ is cross-validated as a power of 10 (in practice $\eta \in \{0.1, 1\}$ for ALI-G). SGD, DFW and ALI-G use a Nesterov momentum of 0.9. Following Berrada et al. (2019), for all methods but ALI-G and AdamW, the $\mathcal{L}_2$ regularization is cross-validated in $\{0.0001, 0.0005\}$ on the WRN architecture, and is set to 0.0001 for the DN architecture. For AdamW, the weight-decay is cross-validated as a power of 10. For ALI-G, $\mathcal{L}_2$ regularization is expressed as a constraint on the norm on the vector of parameters; its maximal value is set to 100 for the WRN models, 80 for DN on CIFAR-10 and 75 for DN on CIFAR-100. For all optimization algorithms, the WRN model is trained for 200 epochs and the DN model for 300 epochs, following respectively (Zagoruyko & Komodakis, 2016) and (Huang et al., 2017).

**Results.** We present the results in Table 3. In this setting again, ALI-G obtains competitive performance with manually decayed SGD. ALI-G largely outperforms AMSGrad, AdamW and Yogi. In addition, it significantly bridges the gap between DFW and SGD on CIFAR-10 with the WRN model, and on CIFAR-100 with the DN one.

| Data Set | Architecture | AMSGrad | AdamW | Yogi | DFW | $L_4$Adam | $L_4$Mom | ALI-G | SGD | SGD[†] |
|---|---|---|---|---|---|---|---|---|---|---|
| CIFAR-10 | WRN | 90.8 | 92.1 | 91.2 | 94.2 | 90.5 | 91.6 | **95.2** | 95.3 | 95.4 |
| | DN | 91.7 | 92.6 | 92.1 | 94.6 | 90.8 | 91.9 | **95.0** | 95.1 | - |
| CIFAR-100 | WRN | 68.7 | 69.6 | 68.7 | **76.0** | 61.7 | 61.4 | 75.8 | 77.8 | 78.8 |
| | DN | 69.4 | 69.5 | 69.6 | 73.2 | 60.5 | 62.6 | **76.3** | 76.3 | - |

Table 3: *Test Accuracy (%) on the CIFAR data sets. In red, SGD benefits from a hand-designed schedule for its learning-rate. In black, adaptive methods, including ALI-G, have a single hyper-parameter for their learning-rate. SGD[†] refers to the result from Zagoruyko & Komodakis (2016). Each reported result is an average over three independent runs; the standard deviations are reported in Appendix (they are at most 0.3 for ALI-G and SGD).*

## 5.5 TRAINING PERFORMANCE

The experiments have so far focused on testing accuracy (except for the DNC task), because that is the main metric driving practitioners' choice of optimization algorithm. In this section, we empirically assess the performance of ALI-G and its competitors in terms of training objective. In order to have comparable objective functions, the $\mathcal{L}_2$ regularization is deactivated. We do not use dropout. The learning-rate is selected as a power of ten for best final objective value, and the batch-size is set to its default value. All methods use a fix budget of 160 epochs for WRN-SVHN, 200 epochs for WRN-CIFAR and 300 epochs for DN-CIFAR, following (Zagoruyko & Komodakis, 2016) and (Huang et al., 2017). The $L_4$ methods diverge on CIFAR-100 in this setting. For clarity, we only display the

performance of SGD, Adam, Adagrad and ALI-G (DFW does not support the cross-entropy loss). Here SGD uses a constant learning-rate to emphasize the need for adaptivity. Therefore all methods use one hyper-parameter for their learning-rate. As can be seen, ALI-G provides better training performance than the baseline algorithms on all tasks.

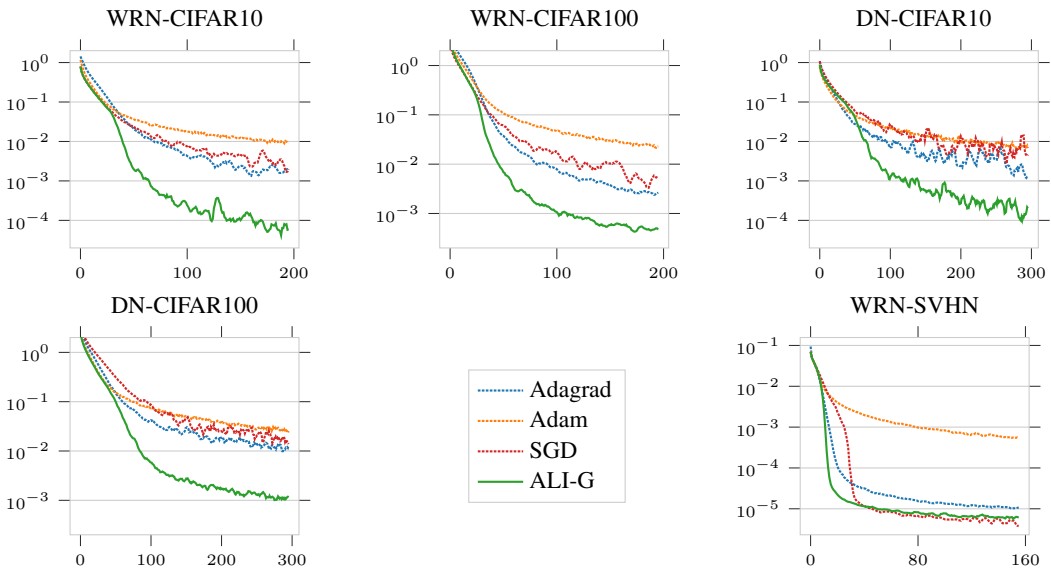

Figure 4: *Objective function over the epochs on CIFAR-10, CIFAR-100 and SVHN (smoothed with a moving average over 5 epochs). On SVHN, ALI-G obtains similar performance to its competitors and converges faster. On CIFAR-10 and CIFAR-100, which are more difficult tasks, ALI-G yields an objective function that is an order of magnitude better than the baselines.*

## 6    DISCUSSION

We hope that the ALI-G algorithm is a helpful step towards efficient and reliable training of deep neural networks. ALI-G is readily applicable to a broad range of applications in deep learning where the model can interpolate the data. When that is not the case however, it would be interesting to design new algorithms that adapt the minimum $f_\star$ online while requiring few hyper-parameters. This could be achieved for instance by building upon the works of Nedić & Bertsekas (2001b) and Rolinek & Martius (2018).

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

## A   LOCAL INTERPRETATION OF THE POLYAK STEP-SIZE

**Proposition 2.** *Suppose that the problem is unconstrained:* $\Omega = \mathbb{R}^p$. *Let* $\mathbf{w}_{t+1} = \mathbf{w}_t - \frac{f(\mathbf{w}_t) - f_\star}{\|\nabla f(\mathbf{w}_t)\|^2} \nabla f(\mathbf{w}_t)$. *Then* $\mathbf{w}_{t+1}$ *verifies:*

$$\mathbf{w}_{t+1} = \operatorname*{argmin}_{\mathbf{w} \in \mathbb{R}^p} \|\mathbf{w} - \mathbf{w}_t\| \text{ subject to: } f(\mathbf{w}_t) + \nabla f(\mathbf{w}_t)^\top (\mathbf{w} - \mathbf{w}_t) = f_\star, \tag{6}$$

*where we remind that* $f_\star$ *is the minimum of* $f$, *and* $\mathbf{w} \mapsto f(\mathbf{w}_t) + \nabla f(\mathbf{w}_t)^\top (\mathbf{w} - \mathbf{w}_t)$ *is the linearization of* $f$ *at* $\mathbf{w}_t$. *In other words,* $\mathbf{w}_{t+1}$ *is the closest point to* $\mathbf{w}_t$ *that lies on the hyper-plane* $f(\mathbf{w}_t) + \nabla f(\mathbf{w}_t)^\top (\mathbf{w} - \mathbf{w}_t) = f_\star$.

**Proof:** See Appendix D.1 ∎

## B   CONVERGENCE RESULTS

Before we detail our convergence results, we introduce the notions of uniform lower bound and $\varepsilon$-interpolation.

### B.1   NOTATION

Intuitively, a uniform lower bound on the problem $(\mathcal{P})$ is a lower bound on all loss functions $\ell_z$ on their unconstrained domain $\mathbb{R}^p$. We formalize this below:

**Definition 1** (Uniform Lower Bound). *We say that* $\underline{\mathrm{B}}$ *is a uniform lower bound on* $(\mathcal{P})$ *if:*

$$\underline{\mathrm{B}} \leq \inf_{z \in \mathcal{Z}} \inf_{\mathbf{w} \in \mathbb{R}^p} \ell_z(\mathbf{w}). \tag{7}$$

Note that in the main paper, we have used the special case $\underline{\mathrm{B}} = 0$ as the uniform lower bound. The definition above makes $\underline{\mathrm{B}}$ a useful statistic to analyze the behavior of each loss function $\ell_z$ around $\mathbf{w}_\star$, in a uniform way (that is, independently of $z$). The quality of a uniform lower bound $\underline{\mathrm{B}}$ can be quantified by the notion of $\varepsilon$-interpolation:

**Definition 2** ($\varepsilon$-Interpolation). *Let* $\underline{\mathrm{B}}$ *be a uniform lower bound on* $(\mathcal{P})$, $\mathbf{w}_\star$ *be a solution of* $(\mathcal{P})$ *and* $\varepsilon \geq 0$ *be a non-negative number. Then we say that* $\mathbf{w}_\star$ *is an* $\varepsilon$-*interpolation for* $((\mathcal{P}), \underline{\mathrm{B}})$ *if:*

$$\forall z \in \mathcal{Z}, \ \ell_z(\mathbf{w}_\star) - \underline{\mathrm{B}} \leq \varepsilon. \tag{8}$$

By taking the expectation over equation (8), we can see that if $\mathbf{w}_\star$ is an $\varepsilon$-interpolation for $((\mathcal{P}), \underline{\mathrm{B}})$, then we immediately have: $f_\star - \varepsilon \leq \underline{\mathrm{B}} \leq f_\star$. In other words, $\underline{\mathrm{B}}$ is also an approximation of $f_\star$ by below, and its quality is quantified by $\varepsilon$. We further note that $f_\star$ does not satisfy the definition of a uniform lower bound in the general case. However when $f_\star$ actually is a uniform lower bound, for any solution $\mathbf{w}_\star$ of $(\mathcal{P})$, $\mathbf{w}_\star$ is a $\varepsilon = 0$-interpolation for $((\mathcal{P}), f_\star)$.

In the general case where $\underline{\mathrm{B}}$ may be different from $0$, the ALI-G step-size can be defined as:

$$\gamma_t = \min \left\{ \frac{\ell_{z_t}(\mathbf{w}_t) - \underline{\mathrm{B}}}{\|\nabla \ell_{z_t}(\mathbf{w}_t)\|^2 + \delta}, \eta \right\} \tag{9}$$

We now turn to our convergence results, which give convergence rates in three settings: convex and Lipschitz functions, convex and smooth function, and strongly convex and smooth functions. In each setting, we analyze three regimes which complement each other: no (infinite) maximal learning-rate, large maximal learning-rate and small maximal learning-rate.

### B.2   LIPSCHITZ CONVEX FUNCTIONS

**Theorem 2.** *[Convex and Lipschitz] We assume that* $\mathcal{X}$ *is a convex set, and that for every* $z \in \mathcal{Z}$, $\ell_z$ *is convex and* $C$-*Lipschitz. Let* $\underline{\mathrm{B}}$ *be a uniform lower bound on* $(\mathcal{P})$ *and* $\mathbf{w}_\star$ *be a solution of* $(\mathcal{P})$. *Further suppose that* $\mathbf{w}_\star$ *is an* $\varepsilon$-*interpolation for* $((\mathcal{P}), \underline{\mathrm{B}})$. *Then ALI-G$^\infty$ applied to* $f$ *satisfies:*

$$f \left( \frac{1}{T+1} \sum_{t=0}^{T} \mathbf{w}_t \right) - f_\star \leq \varepsilon \sqrt{\left( \frac{C^2}{\delta} + 1 \right)} + \frac{\|\mathbf{w}_0 - \mathbf{w}_\star\| \sqrt{C^2 + \delta}}{\sqrt{T+1}}. \tag{10}$$

**Proof :** See Appendix D.3. ∎

**Theorem 3.** *We assume that $\mathcal{X}$ is a convex set, and that for every $z \in \mathcal{Z}$, $\ell_z$ is convex and $C$-Lipschitz. Let $\mathbf{w}_\star$ be an $\varepsilon$-interpolation for $((\mathcal{P})$, $\underline{\mathrm{B}})$. We further assume that $\eta > \frac{\varepsilon}{\delta}$. Then if we apply ALI-G with a maximal learning-rate of $\eta$ to $f$, we have:*

$$f\left(\frac{1}{T+1}\sum_{t=0}^{T}\mathbf{w}_t\right) - f_\star \leq \frac{\|\mathbf{w}_0 - \mathbf{w}_\star\|^2}{(\eta - \frac{\varepsilon}{\delta})(T+1)} + \frac{\varepsilon^2}{\delta(\eta - \frac{\varepsilon}{\delta})} + \sqrt{\frac{(C^2 + \delta)\|\mathbf{w}_0 - \mathbf{w}_\star\|^2}{T+1}} + \varepsilon\sqrt{\frac{C^2}{\delta} + 1}.$$
(11)

**Proof :** See Appendix D.4. ∎

We note that for very large values of $\eta$ ($\eta \to \infty$), Theorem 3 gives the exact same result as Theorem 2. However when $\eta$ is small, the convergence error of Theorem 3 is large. This is corrected in the following result which is informative in the regime where $\eta$ is small:

**Theorem 4.** *We assume that $\mathcal{X}$ is a convex set, and that for every $z \in \mathcal{Z}$, $\ell_z$ is convex and $C$-Lipschitz. Let $\mathbf{w}_\star$ be an $\varepsilon$-interpolation for $((\mathcal{P})$, $\underline{\mathrm{B}})$. Then if we apply ALI-G with a maximal learning-rate of $\eta$ to $f$, we have:*

$$f\left(\frac{1}{T+1}\sum_{t=0}^{T}\mathbf{w}_t\right) - f_\star \leq \frac{\|\mathbf{w}_0 - \mathbf{w}_\star\|^2}{\eta(T+1)} + \varepsilon + \sqrt{\frac{(C^2 + \delta)\|\mathbf{w}_0 - \mathbf{w}_\star\|^2}{T+1}} + \eta\varepsilon\sqrt{C^2 + \delta}. \quad (12)$$

**Proof :** See Appendix D.5. ∎

### B.3 SMOOTH CONVEX FUNCTIONS

We now tackle the convex and $\beta$-smooth case. Our proof techniques naturally produce the separation $\eta \geq \frac{1}{2\beta}$ and $\eta \leq \frac{1}{2\beta}$. Whenever $\eta \geq \frac{1}{2\beta}$, the convergence result is exactly the same as when $\eta \to \infty$. When $\eta \leq \frac{1}{2\beta}$, the speed of convergence is limited by the value of $\eta$.

**Theorem 5.** *[Convex and Smooth] We assume that $\mathcal{X}$ is a convex set, and that for every $z \in \mathcal{Z}$, $\ell_z$ is convex and $\beta$-smooth. Let $\underline{\mathrm{B}}$ be a uniform lower bound on $(\mathcal{P})$ and $\mathbf{w}_\star$ be a solution of $(\mathcal{P})$. Further suppose that $\mathbf{w}_\star$ is an $\varepsilon$-interpolation for $((\mathcal{P})$, $\underline{\mathrm{B}})$, and that $\delta > 2\beta\varepsilon$. Then ALI-G$^\infty$ applied to $f$ satisfies:*

$$f\left(\frac{1}{T+1}\sum_{t=0}^{T}\mathbf{w}_t\right) - f_\star \leq \frac{\delta}{\beta(1 - \frac{2\beta\varepsilon}{\delta})} + \frac{2\beta}{1 - \frac{2\beta\varepsilon}{\delta}}\frac{\|\mathbf{w}_0 - \mathbf{w}_\star\|^2}{T+1}. \quad (13)$$

**Proof :** See Appendix D.6. ∎

**Theorem 6.** *We assume that $\mathcal{X}$ is a convex set, and that for every $z \in \mathcal{Z}$, $\ell_z$ is convex and $\beta$-smooth. Let $\mathbf{w}_\star$ be an $\varepsilon$-interpolation for $((\mathcal{P})$, $\underline{\mathrm{B}})$, and suppose that $\delta > 2\beta\varepsilon$. Further assume that $\eta \geq \frac{1}{2\beta}$. Then if we apply ALI-G with a maximal learning-rate of $\eta$ to $f$, we have:*

$$f\left(\frac{1}{T+1}\sum_{t=0}^{T}\mathbf{w}_t\right) - f_\star \leq \frac{\delta}{\beta(1 - \frac{2\beta\varepsilon}{\delta})} + \frac{2\beta}{1 - \frac{2\beta\varepsilon}{\delta}}\frac{\|\mathbf{w}_0 - \mathbf{w}_\star\|^2}{T+1}. \quad (14)$$

**Proof :** See Appendix D.7. ∎

**Theorem 7.** *We assume that $\mathcal{X}$ is a convex set, and that for every $z \in \mathcal{Z}$, $\ell_z$ is convex and $\beta$-smooth. Let $\mathbf{w}_\star$ be an $\varepsilon$-interpolation for $((\mathcal{P})$, $\underline{\mathrm{B}})$, and suppose that $\delta > 2\beta\varepsilon$. Further assume that $\eta \leq \frac{1}{2\beta}$. Then if we apply ALI-G with a maximal learning-rate of $\eta$ to $f$, we have:*

$$f\left(\frac{1}{T+1}\sum_{t=0}^{T}\mathbf{w}_t\right) - f_\star \leq \frac{\|\mathbf{w}_0 - \mathbf{w}_\star\|^2}{\eta(T+1)} + \frac{\delta}{2\beta} + \varepsilon. \quad (15)$$

**Proof :** See Appendix D.8. ∎

### B.4 Smooth and Strongly Convex Functions

Finally, we consider the $\alpha$-strongly convex and $\beta$-smooth case. Again, our proof yields a natural separation between $\eta \geq \frac{1}{2\beta}$ and $\eta \leq \frac{1}{2\beta}$. In a similar way to the $\beta$-smooth case, when $\eta \geq \frac{1}{2\beta}$, Theorem 9 gives the exact same result as $\eta \to \infty$. And when $\eta \leq \frac{1}{2\beta}$, the rate of convergence given by Theorem 10 is limited by the value of $\eta$.

**Theorem 8.** *[Strongly Convex and Smooth] We assume that $\mathcal{X}$ is a convex set, and that for every $z \in \mathcal{Z}$, $\ell_z$ is $\alpha$-strongly convex and $\beta$-smooth. Let $\underline{\mathtt{B}}$ be a uniform lower bound on ($\mathcal{P}$) and $\mathbf{w}_\star$ be a solution of ($\mathcal{P}$). Further suppose that $\mathbf{w}_\star$ is an $\varepsilon$-interpolation for (($\mathcal{P}$), $\underline{\mathtt{B}}$), and that $\delta > 2\beta\varepsilon$. Then ALI-G$^\infty$ applied to $f$ satisfies:*

$$f(\mathbf{w}_{T+1}) - f_\star \leq \beta \exp\left(-\frac{\alpha T}{8\beta}\right) \|\mathbf{w}_0 - \mathbf{w}_\star\|^2 + \frac{2\delta}{\alpha} + \left(10\frac{\beta}{\alpha} + 4\frac{\beta^2}{\alpha^2}\right)\varepsilon. \tag{16}$$

*In other words, $f$ approximately converges to $f_\star$ at a rate of $\mathcal{O}(\exp(-\alpha T/8\beta))$.*

**Proof :** See Appendix D.9. ∎

**Theorem 9.** *We assume that $\mathcal{X}$ is a convex set, and that for every $z \in \mathcal{Z}$, $\ell_z$ is $\alpha$-strongly convex and $\beta$-smooth. Let $\mathbf{w}_\star$ be an $\varepsilon$-interpolation for (($\mathcal{P}$), $\underline{\mathtt{B}}$), and suppose that $\delta > 2\beta\varepsilon$. Further assume that $\eta \geq \frac{1}{2\beta}$. Then if we apply ALI-G with a maximal learning-rate of $\eta$ to $f$, we have:*

$$f(\mathbf{w}_{T+1}) - f_\star \leq \beta \exp\left(-\frac{\alpha T}{8\beta}\right) \|\mathbf{w}_0 - \mathbf{w}_\star\|^2 + \frac{2\delta}{\alpha} + \left(10\frac{\beta}{\alpha} + 4\frac{\beta^2}{\alpha^2}\right)\varepsilon. \tag{17}$$

**Proof :** See Appendix D.10. ∎

**Theorem 10.** *We assume that $\mathcal{X}$ is a convex set, and that for every $z \in \mathcal{Z}$, $\ell_z$ is $\alpha$-strongly convex and $\beta$-smooth. Let $\mathbf{w}_\star$ be an $\varepsilon$-interpolation for (($\mathcal{P}$), $\underline{\mathtt{B}}$), and suppose that $\delta > 2\beta\varepsilon$. Further assume that $\eta \leq \frac{1}{2\beta}$. Then if we apply ALI-G with a maximal learning-rate of $\eta$ to $f$, we have:*

$$f(\mathbf{w}_{T+1}) - f_\star \leq \beta \exp\left(-\frac{\alpha \eta T}{4}\right) \|\mathbf{w}_0 - \mathbf{w}_\star\|^2 + \frac{2\delta}{\alpha} + \frac{14\varepsilon\beta}{\alpha}. \tag{18}$$

**Proof :** See Appendix D.11. ∎

## C  On the Need for a Maximal Learning-Rate for Non-Convexity

The Restricted Secant Inequality (RSI) is a milder assumption than convexity. It can be defined as follows:

**Definition 3.** *Let $f : \mathbb{R}^p \to \mathbb{R}$ be a lower-bounded differentiable function achieving its minimum at $\mathbf{w}_\star$. We say that $f$ satisfies the RSI if there exists $\alpha > 0$ such that:*

$$\forall \mathbf{w} \in \mathbb{R}^p, \ \nabla f(\mathbf{w})^\top (\mathbf{w} - \mathbf{w}_\star) \geq \alpha \|\mathbf{w} - \mathbf{w}_\star\|^2. \tag{19}$$

The RSI is sometimes used to prove convergence of optimization algorithms without assuming convexity (Vaswani et al., 2019b).

As we prove below, the Polyak step-size may not convergence under the RSI assumption, even in a non-stochastic setting with the exact minimum known.

We introduce the function $f : w \in \left[\frac{-3}{5}; \frac{3}{5}\right] \mapsto w^2 - |w|^3$. We restrict our domain of study to $\left[\frac{-3}{5}; \frac{3}{5}\right]$ for simplicity purposes – an extension to $\mathbb{R}$ can easily be constructed by extending $f$. We will first show that $f$ fulfills the RSI assumption, and then that it oscillates between two points for a well chosen initialization.

Let us show that $f$ satisfies the RSI with $\alpha = \frac{1}{5}$. First we note that $f$ achieves its minimum at $w_\star = 0$, and that $f(w_\star) = 0$. In addition, we introduce the sign function $\sigma(w)$, which is equal to 1 if $w \geq 0$,

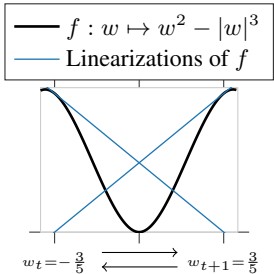

Figure 5: *Illustration of the function $f$, which satisfies the RSI. When starting at $w = -3/5$, gradient descent with the Polyak step-size oscillates between $w = -3/5$ and $w = 3/5$.*

and $-1$ otherwise. Now let $w \in [\frac{-3}{5}; \frac{3}{5}]$. Then we have that:

$$
\begin{aligned}
\nabla f(w)(w - w_\star) - \frac{1}{5}(w - w_\star)^2, &= (2w - 3\sigma(w)w^2)(w - 0) - \frac{1}{5}(w - 0)^2, \\
&= \frac{9}{5}w^2 - 3\sigma(w)w^3, \\
&= 3w^2(\frac{3}{5} - \sigma(w)w), \\
&\geq 0.
\end{aligned}
\tag{20}
$$

Now let us show that if we apply gradient descent with a Polyak step-size to $f$, with starting point $w_0 = \frac{-3}{5}$, we obtain $w_1 = \frac{3}{5}$. This will prove oscillation of the iterates by symmetry of the problem. Let $w_0 = \frac{-3}{5}$. Then we have $f(w_0) = \frac{9}{25} - \frac{27}{125} = \frac{18}{125}$. Furthermore, $\nabla f(w_0) = 2(\frac{-3}{5}) + 3(\frac{9}{25}) = \frac{-3}{25}$. Therefore:

$$
\begin{aligned}
w_1 &= w_0 - \frac{f(w_0) - 0}{(\nabla f(w_0))^2}\nabla f(w_0), \\
&= w_0 - \frac{f(w_0)}{\nabla f(w_0)}, \\
&= \frac{-3}{5} + \frac{\frac{18}{125}}{\frac{3}{25}}, \\
&= \frac{-3}{5} + \frac{6}{5}, \\
&= \frac{3}{5}.
\end{aligned}
\tag{21}
$$

## D  PROOFS

### D.1  PROPOSITION 2

**Proposition 2.** *Suppose that the problem is unconstrained: $\Omega = \mathbb{R}^p$. Let $\mathbf{w}_{t+1} = \mathbf{w}_t - \frac{f(\mathbf{w}_t) - f_\star}{\|\nabla f(\mathbf{w}_t)\|^2}\nabla f(\mathbf{w}_t)$. Then $\mathbf{w}_{t+1}$ verifies:*

$$
\mathbf{w}_{t+1} = \underset{\mathbf{w} \in \mathbb{R}^p}{\operatorname{argmin}} \|\mathbf{w} - \mathbf{w}_t\| \text{ subject to: } f(\mathbf{w}_t) + \nabla f(\mathbf{w}_t)^\top(\mathbf{w} - \mathbf{w}_t) = f_\star,
\tag{6}
$$

*where we remind that $f_\star$ is the minimum of $f$, and $\mathbf{w} \mapsto f(\mathbf{w}_t) + \nabla f(\mathbf{w}_t)^\top(\mathbf{w} - \mathbf{w}_t)$ is the linearization of $f$ at $\mathbf{w}_t$. In other words, $\mathbf{w}_{t+1}$ is the closest point to $\mathbf{w}_t$ that lies on the hyper-plane $f(\mathbf{w}_t) + \nabla f(\mathbf{w}_t)^\top(\mathbf{w} - \mathbf{w}_t) = f_\star$.*

**Proof:** First we show that $\mathbf{w}_{t+1}$ satisfies the linear equality constraint:

$$\begin{aligned}
f(\mathbf{w}_t) &+ \nabla f(\mathbf{w}_t)^\top (\mathbf{w}_{t+1} - \mathbf{w}_t) \\
&= f(\mathbf{w}_t) + \nabla f(\mathbf{w}_t)^\top \left( -\frac{f(\mathbf{w}_t) - f_\star}{\|\nabla f(\mathbf{w}_t)\|^2} \nabla f(\mathbf{w}_t) \right), \\
&= f(\mathbf{w}_t) - f(\mathbf{w}_t) + f_\star, \\
&= f_\star.
\end{aligned} \tag{22}$$

Now let us show that it has a minimal distance to $\mathbf{w}_t$.

We take $\hat{\mathbf{w}} \in \mathbb{R}^p$ a solution of the linear equality constraint, and we will show that $\|\mathbf{w}_{t+1} - \mathbf{w}_t\| \leq \|\hat{\mathbf{w}} - \mathbf{w}_t\|$. By definition, we have that $\hat{\mathbf{w}}$ satisfies:

$$f(\mathbf{w}_t) + \nabla f(\mathbf{w}_t)^\top (\hat{\mathbf{w}} - \mathbf{w}_t) = f_\star. \tag{23}$$

Now we can write:

$$\begin{aligned}
\|\mathbf{w}_{t+1} - \mathbf{w}_t\| &= \|\frac{f(\mathbf{w}_t) - f_\star}{\|\nabla f(\mathbf{w}_t)\|^2} \nabla f(\mathbf{w}_t)\|, \\
&= \frac{f(\mathbf{w}_t) - f_\star}{\|\nabla f(\mathbf{w}_t)\|}, \\
&= \frac{|\nabla f(\mathbf{w}_t)^\top (\hat{\mathbf{w}} - \mathbf{w}_t)|}{\|\nabla f(\mathbf{w}_t)\|}, \\
&\leq \frac{\|\nabla f(\mathbf{w}_t)\| \|\hat{\mathbf{w}} - \mathbf{w}_t\|}{\|\nabla f(\mathbf{w}_t)\|}, \quad \text{(Cauchy-Schwarz)} \\
&= \|\hat{\mathbf{w}} - \mathbf{w}_t\|.
\end{aligned} \tag{24}$$

∎

## D.2 THEOREM 1

**Proposition 1.** *[Proximal Interpretation] Suppose that $\Omega = \mathbb{R}^p$ and let $\delta = 0$. We consider the update performed by SGD: $\mathbf{w}_{t+1}^{SGD} = \mathbf{w}_t - \eta_t \nabla \ell_{z_t}(\mathbf{w}_t)$; and the update performed by ALI-G: $\mathbf{w}_{t+1}^{ALI\text{-}G} = \mathbf{w}_t - \gamma_t \nabla \ell_{z_t}(\mathbf{w}_t)$, where $\gamma_t = \min\left\{ \frac{\ell_{z_t}(\mathbf{w}_t)}{\|\nabla \ell_{z_t}(\mathbf{w}_t)\|^2 + \delta}, \eta \right\}$. Then we have:*

$$\mathbf{w}_{t+1}^{SGD} = \underset{\mathbf{w} \in \mathbb{R}^p}{\operatorname{argmin}} \left\{ \frac{1}{2\eta_t} \|\mathbf{w} - \mathbf{w}_t\|^2 + \ell_{z_t}(\mathbf{w}_t) + \nabla \ell_{z_t}(\mathbf{w}_t)^\top (\mathbf{w} - \mathbf{w}_t) \right\}, \tag{4}$$

$$\mathbf{w}_{t+1}^{ALI\text{-}G} = \underset{\mathbf{w} \in \mathbb{R}^p}{\operatorname{argmin}} \left\{ \frac{1}{2\eta} \|\mathbf{w} - \mathbf{w}_t\|^2 + \max\left\{ \ell_{z_t}(\mathbf{w}_t) + \nabla \ell_{z_t}(\mathbf{w}_t)^\top (\mathbf{w} - \mathbf{w}_t), 0 \right\} \right\}. \tag{5}$$

**Proof :** We tackle the slightly more general case where $\underline{B}$ may be different from zero. In order to make the notation simpler, we use $\boldsymbol{d}_t \triangleq \nabla \ell_{z_t}(\mathbf{w}_t)$ and $l_t \triangleq \ell_{z_t}(\mathbf{w}_t) - \underline{B}$.
First, let us consider $\boldsymbol{d}_t = \mathbf{0}$.

Then we choose $\gamma_t = 0$ and it is clear that $\mathbf{w}_{t+1} = \mathbf{w}_t - \eta \gamma_t \boldsymbol{d}_t = \mathbf{w}_t$ is the optimal solution of problem (5).

We now assume $\boldsymbol{d}_t \neq \mathbf{0}$.

We can successively re-write the proximal problem (5) as :

$$\min_{\mathbf{w} \in \mathbb{R}^p} \left\{ \frac{1}{2\eta} \|\mathbf{w} - \mathbf{w}_t\|^2 + \max\left\{ \ell_{z_t}(\mathbf{w}_t) + \nabla \ell_{z_t}(\mathbf{w}_t)^\top (\mathbf{w} - \mathbf{w}_t), \underline{B} \right\} \right\},$$

$$\min_{\mathbf{w} \in \mathbb{R}^p} \left\{ \frac{1}{2\eta} \|\mathbf{w} - \mathbf{w}_t\|^2 + \max\left\{ \ell_{z_t}(\mathbf{w}_t) - \underline{B} + \nabla \ell_{z_t}(\mathbf{w}_t)^\top (\mathbf{w} - \mathbf{w}_t), 0 \right\} \right\},$$

$$\min_{\mathbf{w} \in \mathbb{R}^p} \left\{ \frac{1}{2\eta} \|\mathbf{w} - \mathbf{w}_t\|^2 + \max\left\{ l_t + \boldsymbol{d}_t^\top (\mathbf{w} - \mathbf{w}_t), 0 \right\} \right\},$$

$$\min_{\mathbf{w} \in \mathbb{R}^p, \upsilon} \left\{ \frac{1}{2\eta} \|\mathbf{w} - \mathbf{w}_t\|^2 + \upsilon \right\} \text{ subject to: } \upsilon \geq 0, \ \upsilon \geq l_t + \boldsymbol{d}_t^\top (\mathbf{w} - \mathbf{w}_t)$$

$$\min_{\mathbf{w}\in\mathbb{R}^p, \upsilon} \sup_{\mu,\nu\geq 0} \left\{ \frac{1}{2\eta}\|\mathbf{w} - \mathbf{w}_t\|^2 + \upsilon - \mu\upsilon - \nu(\upsilon - l_t - \boldsymbol{d}_t^\top(\mathbf{w} - \mathbf{w}_t)) \right\}$$

$$\sup_{\mu,\nu\geq 0} \min_{\mathbf{w}\in\mathbb{R}^p, \upsilon} \left\{ \frac{1}{2\eta}\|\mathbf{w} - \mathbf{w}_t\|^2 + \upsilon - \mu\upsilon - \nu(\upsilon - l_t - \boldsymbol{d}_t^\top(\mathbf{w} - \mathbf{w}_t)) \right\} \quad \text{(strong duality)} \tag{25}$$

The inner problem is now smooth in $\mathbf{w}$ and $\upsilon$. We write its KKT conditions:

$$\frac{\partial \cdot}{\partial \upsilon} = 0: \quad 1 - \mu - \nu = 0 \tag{26}$$

$$\frac{\partial \cdot}{\partial \mathbf{w}} = 0: \quad \frac{1}{\eta}(\mathbf{w} - \mathbf{w}_t) + \nu\boldsymbol{d}_t = \mathbf{0} \tag{27}$$

We plug in these results and obtain:

$$\sup_{\mu,\nu\geq 0} \left\{ \frac{1}{2\eta}\|\eta\nu\boldsymbol{d}_t\|^2 + \nu(l_t + \boldsymbol{d}_t^\top(-\eta\nu\boldsymbol{d}_t)) \right\}$$

$$\text{st:} \quad \mu + \nu = 1$$

$$\sup_{\nu\in[0,1]} \left\{ \frac{\eta}{2}\nu^2\|\boldsymbol{d}_t\|^2 + \nu l_t - \eta\nu^2\|\boldsymbol{d}_t^\top\|^2 \right\}$$

$$\sup_{\nu\in[0,1]} \left\{ -\frac{\eta}{2}\nu^2\|\boldsymbol{d}_t\|^2 + \nu l_t \right\} \tag{28}$$

This is a one-dimensional quadratic problem in $\nu$. It can be solved in closed-form by finding the global maximum of the quadratic objective, and projecting the solution on $[0, 1]$. We have:

$$\frac{\partial \cdot}{\partial \nu} = 0: -\eta\nu\|\boldsymbol{d}_t\|^2 + l_t = 0 \tag{29}$$

Since $\boldsymbol{d}_t \neq \mathbf{0}$ and $\eta \neq 0$, this gives the optimal solution:

$$\nu = \min\left\{ \max\left\{ \frac{l_t}{\eta\|\boldsymbol{d}_t\|^2}, 0 \right\}, 1 \right\} = \min\left\{ \frac{l_t}{\eta\|\boldsymbol{d}_t\|^2}, 1 \right\}, \tag{30}$$

since $l_t, \eta, \|\boldsymbol{d}_t\|^2 \geq 0$.

Plugging this back in the KKT conditions, we obtain that the solution $\mathbf{w}_{t+1}$ of the primal problem can be written as:

$$\begin{aligned} \mathbf{w}_{t+1} &= \mathbf{w}_t - \eta\nu\boldsymbol{d}_t, \\ &= \mathbf{w}_t - \eta\min\left\{ \frac{l_t}{\eta\|\boldsymbol{d}_t\|^2}, 1 \right\}\boldsymbol{d}_t, \\ &= \mathbf{w}_t - \eta\min\left\{ \frac{\ell_{z_t}(\mathbf{w}_t) - \underline{\mathrm{B}}}{\eta\|\nabla\ell_{z_t}(\mathbf{w}_t)\|^2}, 1 \right\}\nabla\ell_{z_t}(\mathbf{w}_t), \\ &= \mathbf{w}_t - \min\left\{ \frac{\ell_{z_t}(\mathbf{w}_t) - \underline{\mathrm{B}}}{\|\nabla\ell_{z_t}(\mathbf{w}_t)\|^2}, \eta \right\}\nabla\ell_{z_t}(\mathbf{w}_t). \end{aligned} \tag{31}$$

∎

## D.3 THEOREM 2

**Theorem 2.** *[Convex and Lipschitz] We assume that $\mathcal{X}$ is a convex set, and that for every $z \in \mathcal{Z}$, $\ell_z$ is convex and $C$-Lipschitz. Let $\underline{\mathrm{B}}$ be a uniform lower bound on $(\mathcal{P})$ and $\mathbf{w}_\star$ be a solution of $(\mathcal{P})$. Further suppose that $\mathbf{w}_\star$ is an $\varepsilon$-interpolation for $((\mathcal{P}), \underline{\mathrm{B}})$. Then ALI-G$^\infty$ applied to $f$ satisfies:*

$$f\left( \frac{1}{T+1}\sum_{t=0}^{T}\mathbf{w}_t \right) - f_\star \leq \varepsilon\sqrt{\left( \frac{C^2}{\delta} + 1 \right)} + \frac{\|\mathbf{w}_0 - \mathbf{w}_\star\|\sqrt{C^2 + \delta}}{\sqrt{T+1}}. \tag{10}$$

**Proof :**

We consider the update at time $t$, which we condition on the draw of $z_t \in \mathcal{Z}$:

$$\|\mathbf{w}_{t+1} - \mathbf{w}_\star\|^2$$

$$= \|\Pi_\Omega(\mathbf{w}_t - \gamma_t \nabla \ell_{z_t}(\mathbf{w}_t)) - \mathbf{w}_\star\|^2$$

$$\leq \|\mathbf{w}_t - \gamma_t \nabla \ell_{z_t}(\mathbf{w}_t) - \mathbf{w}_\star\|^2 \quad (\Pi_\Omega \text{ projection})$$

$$= \|\mathbf{w}_t - \mathbf{w}_\star\|^2 - 2\gamma_t \nabla \ell_{z_t}(\mathbf{w}_t)^\top (\mathbf{w}_t - \mathbf{w}_\star) + \gamma_t^2 \|\nabla \ell_{z_t}(\mathbf{w}_t)\|^2$$

$$= \|\mathbf{w}_t - \mathbf{w}_\star\|^2 - 2\gamma_t \nabla \ell_{z_t}(\mathbf{w}_t)^\top (\mathbf{w}_t - \mathbf{w}_\star) + \gamma_t \frac{\ell_{z_t}(\mathbf{w}_t) - \underline{\mathrm{B}}}{\|\nabla \ell_{z_t}(\mathbf{w}_t)\|^2 + \delta} \|\nabla \ell_{z_t}(\mathbf{w}_t)\|^2$$

$$\text{(definition of } \gamma_t)$$

$$\leq \|\mathbf{w}_t - \mathbf{w}_\star\|^2 - 2\gamma_t \nabla \ell_{z_t}(\mathbf{w}_t)^\top (\mathbf{w}_t - \mathbf{w}_\star) + \gamma_t \frac{\ell_{z_t}(\mathbf{w}_t) - \underline{\mathrm{B}}}{\|\nabla \ell_{z_t}(\mathbf{w}_t)\|^2} \|\nabla \ell_{z_t}(\mathbf{w}_t)\|^2$$

$$\text{(because } \ell_{z_t}(\mathbf{w}_t) - \underline{\mathrm{B}} \geq 0 \text{ and } \delta \geq 0)$$

$$\leq \|\mathbf{w}_t - \mathbf{w}_\star\|^2 - 2\gamma_t(\ell_{z_t}(\mathbf{w}_t) - \ell_{z_t}(\mathbf{w}_\star)) + \gamma_t(\ell_{z_t}(\mathbf{w}_t) - \underline{\mathrm{B}}) \quad (\text{convexity of } \ell_{z_t})$$

$$= \|\mathbf{w}_t - \mathbf{w}_\star\|^2 - \gamma_t\Big((\ell_{z_t}(\mathbf{w}_t) - \ell_{z_t}(\mathbf{w}_\star)) - (\ell_{z_t}(\mathbf{w}_\star) - \underline{\mathrm{B}})\Big)$$

$$= \|\mathbf{w}_t - \mathbf{w}_\star\|^2 - \frac{1}{\|\nabla \ell_{z_t}(\mathbf{w}_t)\|^2 + \delta}\Big((\ell_{z_t}(\mathbf{w}_t) - \underline{\mathrm{B}})(\ell_{z_t}(\mathbf{w}_t) - \ell_{z_t}(\mathbf{w}_\star))$$

$$- (\ell_{z_t}(\mathbf{w}_t) - \underline{\mathrm{B}})(\ell_{z_t}(\mathbf{w}_\star) - \underline{\mathrm{B}})\Big) \quad (\text{definition of } \gamma_t)$$

$$= \|\mathbf{w}_t - \mathbf{w}_\star\|^2 - \frac{1}{\|\nabla \ell_{z_t}(\mathbf{w}_t)\|^2 + \delta}\Big((\ell_{z_t}(\mathbf{w}_t) - \ell_{z_t}(\mathbf{w}_\star))(\ell_{z_t}(\mathbf{w}_t) - \ell_{z_t}(\mathbf{w}_\star))$$

$$+ (\ell_{z_t}(\mathbf{w}_\star) - \underline{\mathrm{B}})(\ell_{z_t}(\mathbf{w}_t) - \ell_{z_t}(\mathbf{w}_\star)) - (\ell_{z_t}(\mathbf{w}_t) - \ell_{z_t}(\mathbf{w}_\star))(\ell_{z_t}(\mathbf{w}_\star) - \underline{\mathrm{B}})$$

$$- (\ell_{z_t}(\mathbf{w}_\star) - \underline{\mathrm{B}})(\ell_{z_t}(\mathbf{w}_\star) - \underline{\mathrm{B}})\Big)$$

$$(\text{we use twice } \ell_{z_t}(\mathbf{w}_t) - \underline{\mathrm{B}} = \ell_{z_t}(\mathbf{w}_t) - \ell_{z_t}(\mathbf{w}_\star) + \ell_{z_t}(\mathbf{w}_\star) - \underline{\mathrm{B}})$$

$$= \|\mathbf{w}_t - \mathbf{w}_\star\|^2 - \frac{1}{\|\nabla \ell_{z_t}(\mathbf{w}_t)\|^2 + \delta}\Big((\ell_{z_t}(\mathbf{w}_t) - \ell_{z_t}(\mathbf{w}_\star))^2 - (\ell_{z_t}(\mathbf{w}_\star) - \underline{\mathrm{B}})^2\Big)$$

$$(\text{middle terms cancel out})$$

$$= \|\mathbf{w}_t - \mathbf{w}_\star\|^2 - \frac{(\ell_{z_t}(\mathbf{w}_t) - \ell_{z_t}(\mathbf{w}_\star))^2}{\|\nabla \ell_{z_t}(\mathbf{w}_t)\|^2 + \delta} + \frac{(\ell_{z_t}(\mathbf{w}_\star) - \underline{\mathrm{B}})^2}{\|\nabla \ell_{z_t}(\mathbf{w}_t)\|^2 + \delta}$$

$$\leq \|\mathbf{w}_t - \mathbf{w}_\star\|^2 - \frac{(\ell_{z_t}(\mathbf{w}_t) - \ell_{z_t}(\mathbf{w}_\star))^2}{C^2 + \delta} + \frac{(\ell_{z_t}(\mathbf{w}_\star) - \underline{\mathrm{B}})^2}{\delta} \quad (0 \leq \|\nabla \ell_{z_t}(\mathbf{w}_t)\|^2 \leq C)$$

$$\leq \|\mathbf{w}_t - \mathbf{w}_\star\|^2 - \frac{(\ell_{z_t}(\mathbf{w}_t) - \ell_{z_t}(\mathbf{w}_\star))^2}{C^2 + \delta} + \frac{\varepsilon^2}{\delta} \quad (\text{definition of } \varepsilon) \tag{32}$$

We re-write this inequality as:

$$(\ell_{z_t}(\mathbf{w}_t) - \ell_{z_t}(\mathbf{w}_\star))^2 \leq \varepsilon^2 \left(\frac{C^2}{\delta} + 1\right) + (C^2 + \delta)\left(\|\mathbf{w}_t - \mathbf{w}_\star\|^2 - \|\mathbf{w}_{t+1} - \mathbf{w}_\star\|^2\right) \tag{33}$$

We can now use the Cauchy-Schwarz inequality to bound the sum over the iterations:

$$(T+1)\sum_{t=0}^T (\ell_{z_t}(\mathbf{w}_t) - \ell_{z_t}(\mathbf{w}_\star))^2 \geq \left(\sum_{t=0}^T \ell_{z_t}(\mathbf{w}_t) - \ell_{z_t}(\mathbf{w}_\star)\right)^2 \tag{34}$$

Therefore we can write:

$$\left(\sum_{t=0}^T \ell_{z_t}(\mathbf{w}_t) - \ell_{z_t}(\mathbf{w}_\star)\right)^2$$

$$\leq (T+1)\sum_{t=0}^T (\ell_{z_t}(\mathbf{w}_t) - \ell_{z_t}(\mathbf{w}_\star))^2 \tag{35}$$

$$\leq (T+1)^2 \varepsilon^2 \left(\frac{C^2}{\delta} + 1\right) + (T+1)(C^2 + \delta)\left(\|\mathbf{w}_0 - \mathbf{w}_\star\|^2 - \|\mathbf{w}_{T+1} - \mathbf{w}_\star\|^2\right),$$

$$\leq (T+1)^2 \varepsilon^2 \left(\frac{C^2}{\delta} + 1\right) + (T+1)(C^2 + \delta)\|\mathbf{w}_0 - \mathbf{w}_\star\|^2,$$

which yields:

$$\sum_{t=0}^{T} \ell_{z_t}(\mathbf{w}_t) - \ell_{z_t}(\mathbf{w}_\star) \leq \sqrt{(T+1)^2\, \varepsilon^2 \left( \frac{C^2}{\delta} + 1 \right) + (T+1)\left( C^2 + \delta \right) \|\mathbf{w}_0 - \mathbf{w}_\star\|^2}, \tag{36}$$

$$\leq (T+1)\, \varepsilon \sqrt{\frac{C^2}{\delta} + 1} + \sqrt{(T+1)\left( C^2 + \delta \right) \|\mathbf{w}_0 - \mathbf{w}_\star\|^2},$$

We can now take the expectation over the $z_t$:

$$\sum_{t=0}^{T} f(\mathbf{w}_t) - f_\star \leq (T+1)\, \varepsilon \sqrt{\frac{C^2}{\delta} + 1} + \sqrt{(T+1)\left( C^2 + \delta \right) \|\mathbf{w}_0 - \mathbf{w}_\star\|^2}. \tag{37}$$

Dividing by $T+1$ and exploiting convexity of $f$, we finally get:

$$f\left( \frac{1}{T+1} \sum_{t=0}^{T} \mathbf{w}_t \right) - f_\star \leq \frac{1}{T+1} \sum_{t=0}^{T} f(\mathbf{w}_t) - f_\star \quad \text{(convexity of } f\text{)}$$

$$\leq \varepsilon \sqrt{\frac{C^2}{\delta} + 1} + \sqrt{\frac{\left( C^2 + \delta \right) \|\mathbf{w}_0 - \mathbf{w}_\star\|^2}{T+1}}. \tag{38}$$

∎

### D.4 THEOREM 3

**Theorem 3.** *We assume that $\mathcal{X}$ is a convex set, and that for every $z \in \mathcal{Z}$, $\ell_z$ is convex and $C$-Lipschitz. Let $\mathbf{w}_\star$ be an $\varepsilon$-interpolation for $((\mathcal{P}), \underline{\mathrm{B}})$. We further assume that $\eta > \frac{\varepsilon}{\delta}$. Then if we apply ALI-G with a maximal learning-rate of $\eta$ to $f$, we have:*

$$f\left( \frac{1}{T+1} \sum_{t=0}^{T} \mathbf{w}_t \right) - f_\star \leq \frac{\|\mathbf{w}_0 - \mathbf{w}_\star\|^2}{(\eta - \frac{\varepsilon}{\delta})(T+1)} + \frac{\varepsilon^2}{\delta(\eta - \frac{\varepsilon}{\delta})} + \sqrt{\frac{(C^2 + \delta)\|\mathbf{w}_0 - \mathbf{w}_\star\|^2}{T+1}} + \varepsilon \sqrt{\frac{C^2}{\delta} + 1}. \tag{11}$$

**Proof :**

We consider the update at time $t$, which we condition on the draw of $z_t \in \mathcal{Z}$:

$$\|\mathbf{w}_{t+1} - \mathbf{w}_\star\|^2$$

$$= \|\Pi_\Omega(\mathbf{w}_t - \gamma_t \nabla \ell_{z_t}(\mathbf{w}_t)) - \mathbf{w}_\star\|^2$$

$$\leq \|\mathbf{w}_t - \gamma_t \nabla \ell_{z_t}(\mathbf{w}_t) - \mathbf{w}_\star\|^2 \quad (\Pi_\Omega \text{ projection})$$

$$= \|\mathbf{w}_t - \mathbf{w}_\star\|^2 - 2\gamma_t \nabla \ell_{z_t}(\mathbf{w}_t)^\top (\mathbf{w}_t - \mathbf{w}_\star) + \gamma_t^2 \|\nabla \ell_{z_t}(\mathbf{w}_t)\|^2$$

$$\leq \|\mathbf{w}_t - \mathbf{w}_\star\|^2 - 2\gamma_t \nabla \ell_{z_t}(\mathbf{w}_t)^\top (\mathbf{w}_t - \mathbf{w}_\star) + \gamma_t \frac{\ell_{z_t}(\mathbf{w}_t) - \underline{\mathrm{B}}}{\|\nabla \ell_{z_t}(\mathbf{w}_t)\|^2 + \delta} \|\nabla \ell_{z_t}(\mathbf{w}_t)\|^2$$

$$\text{(because } \gamma_t \leq \frac{\ell_{z_t}(\mathbf{w}_t) - \underline{\mathrm{B}}}{\|\nabla \ell_{z_t}(\mathbf{w}_t)\|^2 + \delta})$$

$$\leq \|\mathbf{w}_t - \mathbf{w}_\star\|^2 - 2\gamma_t \nabla \ell_{z_t}(\mathbf{w}_t)^\top (\mathbf{w}_t - \mathbf{w}_\star) + \gamma_t \frac{\ell_{z_t}(\mathbf{w}_t) - \underline{\mathrm{B}}}{\|\nabla \ell_{z_t}(\mathbf{w}_t)\|^2} \|\nabla \ell_{z_t}(\mathbf{w}_t)\|^2$$

$$\text{(because } \ell_{z_t}(\mathbf{w}_t) - \underline{\mathrm{B}} \geq 0 \text{ and } \delta \geq 0)$$

$$\leq \|\mathbf{w}_t - \mathbf{w}_\star\|^2 - 2\gamma_t(\ell_{z_t}(\mathbf{w}_t) - \ell_{z_t}(\mathbf{w}_\star)) + \gamma_t(\ell_{z_t}(\mathbf{w}_t) - \underline{\mathrm{B}}) \quad \text{(convexity of } \ell_{z_t})$$

$$= \|\mathbf{w}_t - \mathbf{w}_\star\|^2 - 2\gamma_t(\ell_{z_t}(\mathbf{w}_t) - \ell_{z_t}(\mathbf{w}_\star)) + \gamma_t(\ell_{z_t}(\mathbf{w}_t) - \ell_{z_t}(\mathbf{w}_\star)) + \gamma_t(\ell_{z_t}(\mathbf{w}_\star) - \underline{\mathrm{B}})$$

$$= \|\mathbf{w}_t - \mathbf{w}_\star\|^2 - \gamma_t(\ell_{z_t}(\mathbf{w}_t) - \ell_{z_t}(\mathbf{w}_\star)) + \gamma_t(\ell_{z_t}(\mathbf{w}_\star) - \underline{\mathrm{B}}) \tag{39}$$

We now consider different cases, according to the value that $\gamma_t$ takes: $\gamma_t = \frac{\ell_{z_t}(\mathbf{w}_t) - \underline{\mathrm{B}}}{\|\nabla \ell_{z_t}(\mathbf{w}_t)\|^2 + \delta}$ or $\gamma_t = \eta$.

First, suppose that $\gamma_t = \frac{\ell_{z_t}(\mathbf{w}_t) - \underline{B}}{\|\nabla \ell_{z_t}(\mathbf{w}_t)\|^2 + \delta}$. Then we can follow the proof of Theorem 2 to obtain:

$$\|\mathbf{w}_{t+1} - \mathbf{w}_\star\|^2 \leq \|\mathbf{w}_t - \mathbf{w}_\star\|^2 - \frac{(\ell_{z_t}(\mathbf{w}_t) - \ell_{z_t}(\mathbf{w}_\star))^2}{C^2 + \delta} + \frac{\varepsilon^2}{\delta}. \tag{40}$$

Now suppose $\gamma_t = \eta$ and $\ell_{z_t}(\mathbf{w}_t) - \ell_{z_t}(\mathbf{w}_\star) \leq 0$. We can use $\gamma_t \leq \frac{\ell_{z_t}(\mathbf{w}_t) - \underline{B}}{\|\nabla \ell_{z_t}(\mathbf{w}_t)\|^2 + \delta}$ to write:

$$\begin{aligned}
\|\mathbf{w}_{t+1} - \mathbf{w}_\star\|^2 &\leq \|\mathbf{w}_t - \mathbf{w}_\star\|^2 - \gamma_t(\ell_{z_t}(\mathbf{w}_t) - \ell_{z_t}(\mathbf{w}_\star)) + \gamma_t(\ell_{z_t}(\mathbf{w}_\star) - \underline{B}), \\
&\leq \|\mathbf{w}_t - \mathbf{w}_\star\|^2 - \frac{\ell_{z_t}(\mathbf{w}_t) - \underline{B}}{\|\nabla \ell_{z_t}(\mathbf{w}_t)\|^2 + \delta}(\ell_{z_t}(\mathbf{w}_t) - \ell_{z_t}(\mathbf{w}_\star)) \\
&\quad + \frac{\ell_{z_t}(\mathbf{w}_t) - \underline{B}}{\|\nabla \ell_{z_t}(\mathbf{w}_t)\|^2 + \delta}(\ell_{z_t}(\mathbf{w}_\star) - \underline{B}),
\end{aligned} \tag{41}$$

where the last inequality has used $\gamma_t \leq \frac{\ell_{z_t}(\mathbf{w}_t) - \underline{B}}{\|\nabla \ell_{z_t}(\mathbf{w}_t)\|^2 + \delta}$, $\ell_{z_t}(\mathbf{w}_t) - \ell_{z_t}(\mathbf{w}_\star) \leq 0$ and $\ell_{z_t}(\mathbf{w}_\star) - \underline{B} \geq 0$. Therefore we are exactly in the same situation as the first case (where we used $\gamma_t = \frac{\ell_{z_t}(\mathbf{w}_t) - \underline{B}}{\|\nabla \ell_{z_t}(\mathbf{w}_t)\|^2 + \delta}$), and thus we have again:

$$\|\mathbf{w}_{t+1} - \mathbf{w}_\star\|^2 \leq \|\mathbf{w}_t - \mathbf{w}_\star\|^2 - \frac{(\ell_{z_t}(\mathbf{w}_t) - \ell_{z_t}(\mathbf{w}_\star))^2}{C^2 + \delta} + \frac{\varepsilon^2}{\delta}. \tag{42}$$

Now suppose that $\gamma_t = \eta$ and $\ell_{z_t}(\mathbf{w}_t) - \ell_{z_t}(\mathbf{w}_\star) \geq 0$. The inequality (39) gives:

$$\begin{aligned}
\|\mathbf{w}_{t+1} &- \mathbf{w}_\star\|^2 \\
&\leq \|\mathbf{w}_t - \mathbf{w}_\star\|^2 - \gamma_t(\ell_{z_t}(\mathbf{w}_t) - \ell_{z_t}(\mathbf{w}_\star)) + \gamma_t(\ell_{z_t}(\mathbf{w}_\star) - \underline{B}), \\
&= \|\mathbf{w}_t - \mathbf{w}_\star\|^2 - \eta(\ell_{z_t}(\mathbf{w}_t) - \ell_{z_t}(\mathbf{w}_\star)) + \gamma_t(\ell_{z_t}(\mathbf{w}_\star) - \underline{B}), \quad (\gamma_t = \eta) \\
&\leq \|\mathbf{w}_t - \mathbf{w}_\star\|^2 - \eta(\ell_{z_t}(\mathbf{w}_t) - \ell_{z_t}(\mathbf{w}_\star)) + \gamma_t \varepsilon. \quad (\text{definition of } \varepsilon, \gamma_t \geq 0) \\
&\leq \|\mathbf{w}_t - \mathbf{w}_\star\|^2 - \eta(\ell_{z_t}(\mathbf{w}_t) - \ell_{z_t}(\mathbf{w}_\star)) + \varepsilon \frac{\ell_{z_t}(\mathbf{w}_t) - \underline{B}}{\|\nabla \ell_{z_t}(\mathbf{w}_t)\|^2 + \delta}, \\
&\quad (\text{because } \gamma_t \leq \frac{\ell_{z_t}(\mathbf{w}_t) - \underline{B}}{\|\nabla \ell_{z_t}(\mathbf{w}_t)\|^2 + \delta}, \varepsilon \geq 0) \\
&\leq \|\mathbf{w}_t - \mathbf{w}_\star\|^2 - \eta(\ell_{z_t}(\mathbf{w}_t) - \ell_{z_t}(\mathbf{w}_\star)) + \varepsilon \frac{\ell_{z_t}(\mathbf{w}_t) - \underline{B}}{\delta}, \quad (\|\nabla \ell_{z_t}(\mathbf{w}_t)\|^2 \geq 0) \\
&= \|\mathbf{w}_t - \mathbf{w}_\star\|^2 - \eta(\ell_{z_t}(\mathbf{w}_t) - \ell_{z_t}(\mathbf{w}_\star)) + \varepsilon \frac{\ell_{z_t}(\mathbf{w}_t) - \ell_{z_t}(\mathbf{w}_\star) + \ell_{z_t}(\mathbf{w}_\star) - \underline{B}}{\delta}, \\
&\leq \|\mathbf{w}_t - \mathbf{w}_\star\|^2 - \eta(\ell_{z_t}(\mathbf{w}_t) - \ell_{z_t}(\mathbf{w}_\star)) + \varepsilon \frac{\ell_{z_t}(\mathbf{w}_t) - \ell_{z_t}(\mathbf{w}_\star) + \varepsilon}{\delta}, \\
&\quad (\text{because } \ell_{z_t}(\mathbf{w}_\star) - \underline{B} \leq \varepsilon) \\
&= \|\mathbf{w}_t - \mathbf{w}_\star\|^2 - \left(\eta - \frac{\varepsilon}{\delta}\right)(\ell_{z_t}(\mathbf{w}_t) - \ell_{z_t}(\mathbf{w}_\star)) + \frac{\varepsilon^2}{\delta}.
\end{aligned} \tag{43}$$

We now introduce $\mathcal{I}_T$ and $\mathcal{J}_T$ as follows:

$$\begin{aligned}
\mathcal{I}_T &\triangleq \{t \in \{0, ..., T\} : \gamma_t = \eta \text{ and } \ell_{z_t}(\mathbf{w}_t) - \ell_{z_t}(\mathbf{w}_\star) \geq 0\} \\
\mathcal{J}_T &\triangleq \{0, ..., T\} \setminus \mathcal{I}_T
\end{aligned} \tag{44}$$

Then, by combining inequalities (40), (42) and (43), and using a telescopic sum, we obtain:

$$\begin{aligned}
\|\mathbf{w}_{T+1} - \mathbf{w}_\star\|^2 &\leq \|\mathbf{w}_0 - \mathbf{w}_\star\|^2 + \sum_{t \in \mathcal{J}_T} \left(-\frac{(\ell_{z_t}(\mathbf{w}_t) - \ell_{z_t}(\mathbf{w}_\star))^2}{C^2 + \delta} + \frac{\varepsilon^2}{\delta}\right) \\
&\quad + \sum_{t \in \mathcal{I}_T} \left(-\left(\eta - \frac{\varepsilon}{\delta}\right)(\ell_{z_t}(\mathbf{w}_t) - \ell_{z_t}(\mathbf{w}_\star)) + \frac{\varepsilon^2}{\delta}\right)
\end{aligned} \tag{45}$$

Using $\|\mathbf{w}_{T+1} - \mathbf{w}_\star\|^2 \geq 0$, we obtain:

$$\frac{1}{C^2 + \delta} \sum_{t \in \mathcal{J}_T} (\ell_{z_t}(\mathbf{w}_t) - \ell_{z_t}(\mathbf{w}_\star))^2 + \left(\eta - \frac{\varepsilon}{\delta}\right) \sum_{t \in \mathcal{I}_T} (\ell_{z_t}(\mathbf{w}_t) - \ell_{z_t}(\mathbf{w}_\star)) \leq \|\mathbf{w}_0 - \mathbf{w}_\star\|^2 + (T+1)\frac{\varepsilon^2}{\delta} \tag{46}$$

In particular, the inequality (46) gives that:

$$\left(\eta - \frac{\varepsilon}{\delta}\right) \sum_{t \in \mathcal{I}_T} (\ell_{z_t}(\mathbf{w}_t) - \ell_{z_t}(\mathbf{w}_\star)) \leq \|\mathbf{w}_0 - \mathbf{w}_\star\|^2 + (T+1)\frac{\varepsilon^2}{\delta}. \tag{47}$$

Furthermore, for every $t \in \mathcal{I}_T$, we have $(\ell_{z_t}(\mathbf{w}_t) - \ell_{z_t}(\mathbf{w}_\star)) \geq 0$, which yields $\left(\eta - \frac{\varepsilon}{\delta}\right) \sum_{t \in \mathcal{I}_T} (\ell_{z_t}(\mathbf{w}_t) - \ell_{z_t}(\mathbf{w}_\star)) \geq 0$ since $\eta > \frac{\epsilon}{\delta}$. Thus the inequality (46) also gives:

$$\frac{1}{C^2 + \delta} \sum_{t \in \mathcal{J}_T} (\ell_{z_t}(\mathbf{w}_t) - \ell_{z_t}(\mathbf{w}_\star))^2 \leq \|\mathbf{w}_0 - \mathbf{w}_\star\|^2 + (T+1)\frac{\varepsilon^2}{\delta}. \tag{48}$$

Using the Cauchy-Schwarz inequality, we can further write:

$$\left(\sum_{t \in \mathcal{J}_T} \ell_{z_t}(\mathbf{w}_t) - \ell_{z_t}(\mathbf{w}_\star)\right)^2 \leq |\mathcal{J}_T| \sum_{t \in \mathcal{J}_T} (\ell_{z_t}(\mathbf{w}_t) - \ell_{z_t}(\mathbf{w}_\star))^2. \tag{49}$$

Therefore we have:

$$\begin{aligned}
\sum_{t \in \mathcal{J}_T} \ell_{z_t}(\mathbf{w}_t) - \ell_{z_t}(\mathbf{w}_\star) &\leq \sqrt{|\mathcal{J}_T| \sum_{t \in \mathcal{J}_T} (\ell_{z_t}(\mathbf{w}_t) - \ell_{z_t}(\mathbf{w}_\star))^2}, \\
&\leq \sqrt{|\mathcal{J}_T|(C^2 + \delta)\left(\|\mathbf{w}_0 - \mathbf{w}_\star\|^2 + (T+1)\frac{\varepsilon^2}{\delta}\right)}.
\end{aligned} \tag{50}$$

We can now put together inequalities (47) and (49) by writing:

$$\begin{aligned}
\sum_{t=0}^{T} & \ell_{z_t}(\mathbf{w}_t) - \ell_{z_t}(\mathbf{w}_\star) \\
&= \sum_{t \in \mathcal{I}_T} \ell_{z_t}(\mathbf{w}_t) - \ell_{z_t}(\mathbf{w}_\star) + \sum_{t \in \mathcal{J}_T} \ell_{z_t}(\mathbf{w}_t) - \ell_{z_t}(\mathbf{w}_\star) \\
&\leq \frac{1}{\eta - \frac{\varepsilon}{\delta}} \left(\|\mathbf{w}_0 - \mathbf{w}_\star\|^2 + (T+1)\frac{\varepsilon^2}{\delta}\right) + \sqrt{|\mathcal{J}_T|(C^2 + \delta)\left(\|\mathbf{w}_0 - \mathbf{w}_\star\|^2 + (T+1)\frac{\varepsilon^2}{\delta}\right)} \\
&\leq \frac{1}{\eta - \frac{\varepsilon}{\delta}} \left(\|\mathbf{w}_0 - \mathbf{w}_\star\|^2 + (T+1)\frac{\varepsilon^2}{\delta}\right) + \sqrt{(T+1)(C^2 + \delta)\left(\|\mathbf{w}_0 - \mathbf{w}_\star\|^2 + (T+1)\frac{\varepsilon^2}{\delta}\right)}
\end{aligned} \tag{51}$$

Dividing by $T+1$ and taking the expectation, we obtain:

$$\begin{aligned}
f\left(\frac{1}{T+1}\sum_{t=0}^{T} \mathbf{w}_t\right) - f_\star &\leq \frac{1}{T+1}\sum_{t=0}^{T} f(\mathbf{w}_t) - f_\star, \quad (f \text{ is convex}) \\
&\leq \frac{\|\mathbf{w}_0 - \mathbf{w}_\star\|^2}{(\eta - \frac{\varepsilon}{\delta})(T+1)} + \frac{\varepsilon^2}{\delta(\eta - \frac{\varepsilon}{\delta})} + \sqrt{(C^2 + \delta)\left(\frac{\|\mathbf{w}_0 - \mathbf{w}_\star\|^2}{T+1} + \frac{\varepsilon^2}{\delta}\right)}, \\
&\leq \frac{\|\mathbf{w}_0 - \mathbf{w}_\star\|^2}{(\eta - \frac{\varepsilon}{\delta})(T+1)} + \frac{\varepsilon^2}{\delta(\eta - \frac{\varepsilon}{\delta})} + \sqrt{\frac{(C^2 + \delta)\|\mathbf{w}_0 - \mathbf{w}_\star\|^2}{T+1}} \\
&\quad + \varepsilon\sqrt{\frac{C^2}{\delta} + 1}.
\end{aligned} \tag{52}$$

$\blacksquare$

### D.5 THEOREM 4

**Theorem 4.** *We assume that $\mathcal{X}$ is a convex set, and that for every $z \in \mathcal{Z}$, $\ell_z$ is convex and $C$-Lipschitz. Let $\mathbf{w}_\star$ be an $\varepsilon$-interpolation for (($\mathcal{P}$), $\underline{\mathrm{B}}$). Then if we apply ALI-G with a maximal learning-rate of $\eta$ to $f$, we have:*

$$f\left(\frac{1}{T+1}\sum_{t=0}^{T}\mathbf{w}_t\right) - f_\star \leq \frac{\|\mathbf{w}_0 - \mathbf{w}_\star\|^2}{\eta(T+1)} + \varepsilon + \sqrt{\frac{(C^2+\delta)\|\mathbf{w}_0 - \mathbf{w}_\star\|^2}{T+1}} + \eta\varepsilon\sqrt{C^2+\delta}. \quad (12)$$

**Proof :**

We consider the update at time $t$, which we condition on the draw of $z_t \in \mathcal{Z}$. We re-use the inequality (39) from the proof of Theorem 3:

$$\|\mathbf{w}_{t+1} - \mathbf{w}_\star\|^2 \leq \|\mathbf{w}_t - \mathbf{w}_\star\|^2 - \gamma_t(\ell_{z_t}(\mathbf{w}_t) - \ell_{z_t}(\mathbf{w}_\star)) + \gamma_t(\ell_{z_t}(\mathbf{w}_\star) - \underline{\mathrm{B}}) \qquad (53)$$

We consider again different cases, according to the value of $\gamma_t$ and the sign of $\ell_{z_t}(\mathbf{w}_t) - \ell_{z_t}(\mathbf{w}_\star)$.

Suppose that $\ell_{z_t}(\mathbf{w}_t) - \ell_{z_t}(\mathbf{w}_\star) \leq 0$. Then the inequality (53) gives:

$$\begin{aligned}
\|\mathbf{w}_{t+1} &- \mathbf{w}_\star\|^2 \\
&\leq \|\mathbf{w}_t - \mathbf{w}_\star\|^2 - \gamma_t(\ell_{z_t}(\mathbf{w}_t) - \ell_{z_t}(\mathbf{w}_\star)) + \gamma_t(\ell_{z_t}(\mathbf{w}_\star) - \underline{\mathrm{B}}), \\
&\leq \|\mathbf{w}_t - \mathbf{w}_\star\|^2 - \eta(\ell_{z_t}(\mathbf{w}_t) - \ell_{z_t}(\mathbf{w}_\star)) + \gamma_t(\ell_{z_t}(\mathbf{w}_\star) - \underline{\mathrm{B}}), \\
&\quad \text{(because } \gamma_t \leq \eta,\, \ell_{z_t}(\mathbf{w}_t) - \ell_{z_t}(\mathbf{w}_\star) \leq 0) \\
&\leq \|\mathbf{w}_t - \mathbf{w}_\star\|^2 - \eta(\ell_{z_t}(\mathbf{w}_t) - \ell_{z_t}(\mathbf{w}_\star)) + \gamma_t\varepsilon, \quad \text{(definition of } \varepsilon,\, \gamma_t \geq 0) \\
&\leq \|\mathbf{w}_t - \mathbf{w}_\star\|^2 - \eta(\ell_{z_t}(\mathbf{w}_t) - \ell_{z_t}(\mathbf{w}_\star)) + \eta\varepsilon, \quad (\gamma_t \leq \eta,\, \varepsilon \geq 0)
\end{aligned} \qquad (54)$$

Now suppose $\ell_{z_t}(\mathbf{w}_t) - \ell_{z_t}(\mathbf{w}_\star) \geq 0$ and $\gamma_t = \eta$. Then the inequality (53) gives:

$$\begin{aligned}
\|\mathbf{w}_{t+1} - \mathbf{w}_\star\|^2 &\leq \|\mathbf{w}_t - \mathbf{w}_\star\|^2 - \gamma_t(\ell_{z_t}(\mathbf{w}_t) - \ell_{z_t}(\mathbf{w}_\star)) + \gamma_t(\ell_{z_t}(\mathbf{w}_\star) - \underline{\mathrm{B}}), \\
&= \|\mathbf{w}_t - \mathbf{w}_\star\|^2 - \eta(\ell_{z_t}(\mathbf{w}_t) - \ell_{z_t}(\mathbf{w}_\star)) + \eta(\ell_{z_t}(\mathbf{w}_\star) - \underline{\mathrm{B}}), \quad (\gamma_t = \eta) \\
&\leq \|\mathbf{w}_t - \mathbf{w}_\star\|^2 - \eta(\ell_{z_t}(\mathbf{w}_t) - \ell_{z_t}(\mathbf{w}_\star)) + \eta\varepsilon, \quad \text{(definition of } \varepsilon,\, \eta \geq 0)
\end{aligned} \qquad (55)$$

Finally, suppose that $\ell_{z_t}(\mathbf{w}_t) - \ell_{z_t}(\mathbf{w}_\star) \geq 0$ and $\gamma_t = \frac{\ell_{z_t}(\mathbf{w}_t) - \underline{\mathrm{B}}}{\|\nabla\ell_{z_t}(\mathbf{w}_t)\|^2+\delta}$. Then the inequality (53) gives:

$$\begin{aligned}
\|\mathbf{w}_{t+1} &- \mathbf{w}_\star\|^2 \\
&\leq \|\mathbf{w}_t - \mathbf{w}_\star\|^2 - \gamma_t(\ell_{z_t}(\mathbf{w}_t) - \ell_{z_t}(\mathbf{w}_\star)) + \gamma_t(\ell_{z_t}(\mathbf{w}_\star) - \underline{\mathrm{B}}), \\
&\leq \|\mathbf{w}_t - \mathbf{w}_\star\|^2 - \gamma_t(\ell_{z_t}(\mathbf{w}_t) - \ell_{z_t}(\mathbf{w}_\star)) + \eta(\ell_{z_t}(\mathbf{w}_\star) - \underline{\mathrm{B}}), \quad (\gamma_t \leq \eta,\, \ell_{z_t}(\mathbf{w}_\star) - \underline{\mathrm{B}} \geq 0) \\
&\leq \|\mathbf{w}_t - \mathbf{w}_\star\|^2 - \gamma_t(\ell_{z_t}(\mathbf{w}_t) - \ell_{z_t}(\mathbf{w}_\star)) + \eta\varepsilon, \quad \text{(definition of } \varepsilon,\, \eta \geq 0) \\
&= \|\mathbf{w}_t - \mathbf{w}_\star\|^2 - \frac{\ell_{z_t}(\mathbf{w}_t) - \underline{\mathrm{B}}}{\|\nabla\ell_{z_t}(\mathbf{w}_t)\|^2+\delta}(\ell_{z_t}(\mathbf{w}_t) - \ell_{z_t}(\mathbf{w}_\star)) + \eta\varepsilon, \quad (\gamma_t = \frac{\ell_{z_t}(\mathbf{w}_t) - \underline{\mathrm{B}}}{\|\nabla\ell_{z_t}(\mathbf{w}_t)\|^2+\delta}) \\
&\leq \|\mathbf{w}_t - \mathbf{w}_\star\|^2 - \frac{(\ell_{z_t}(\mathbf{w}_t) - \ell_{z_t}(\mathbf{w}_\star))^2}{\|\nabla\ell_{z_t}(\mathbf{w}_t)\|^2+\delta} + \eta\varepsilon, \quad (\ell_{z_t}(\mathbf{w}_t) - \underline{\mathrm{B}} \geq \ell_{z_t}(\mathbf{w}_t) - \ell_{z_t}(\mathbf{w}_\star) \geq 0) \\
&\leq \|\mathbf{w}_t - \mathbf{w}_\star\|^2 - \frac{(\ell_{z_t}(\mathbf{w}_t) - \ell_{z_t}(\mathbf{w}_\star))^2}{C^2+\delta} + \eta\varepsilon, \quad (\|\nabla\ell_{z_t}(\mathbf{w}_t)\|^2 \leq C^2)
\end{aligned}$$
$$(56)$$

We now introduce $\mathcal{I}_T$ and $\mathcal{J}_T$ as follows:

$$\begin{aligned}
\mathcal{J}_T &\triangleq \left\{ t \in \{0, ..., T\} : \gamma_t = \frac{\ell_{z_t}(\mathbf{w}_t) - \underline{\mathrm{B}}}{\|\nabla\ell_{z_t}(\mathbf{w}_t)\|^2+\delta} \text{ and } \ell_{z_t}(\mathbf{w}_t) - \ell_{z_t}(\mathbf{w}_\star) \geq 0 \right\} \\
\mathcal{I}_T &\triangleq \{0, ..., T\} \setminus \mathcal{I}_T
\end{aligned} \qquad (57)$$

Then, by combining inequalities (54), (55) and (56), and using a telescopic sum, we obtain:

$$
\begin{aligned}
\|\mathbf{w}_{T+1} - \mathbf{w}_\star\|^2 \leq \|\mathbf{w}_0 - \mathbf{w}_\star\|^2 &+ \sum_{t \in \mathcal{J}_T} \left( -\frac{(\ell_{z_t}(\mathbf{w}_t) - \ell_{z_t}(\mathbf{w}_\star))^2}{C^2 + \delta} + \eta\varepsilon \right) \\
&+ \sum_{t \in \mathcal{I}_T} \left( -\eta(\ell_{z_t}(\mathbf{w}_t) - \ell_{z_t}(\mathbf{w}_\star)) + \eta\varepsilon \right)
\end{aligned}
\tag{58}
$$

Using $\|\mathbf{w}_{T+1} - \mathbf{w}_\star\|^2 \geq 0$, we obtain:

$$
\frac{1}{C^2 + \delta} \sum_{t \in \mathcal{J}_T} (\ell_{z_t}(\mathbf{w}_t) - \ell_{z_t}(\mathbf{w}_\star))^2 + \eta \sum_{t \in \mathcal{I}_T} (\ell_{z_t}(\mathbf{w}_t) - \ell_{z_t}(\mathbf{w}_\star)) \leq \|\mathbf{w}_0 - \mathbf{w}_\star\|^2 + (T+1)\eta\varepsilon \tag{59}
$$

We now take the expectation and obtain:

$$
\frac{1}{C^2 + \delta} \sum_{t \in \mathcal{J}_T} \mathbb{E}\left[ (\ell_{z_t}(\mathbf{w}_t) - \ell_{z_t}(\mathbf{w}_\star))^2 \right] + \eta \sum_{t \in \mathcal{I}_T} (f(\mathbf{w}_t) - f_\star) \leq \|\mathbf{w}_0 - \mathbf{w}_\star\|^2 + (T+1)\eta\varepsilon \tag{60}
$$

Since $\mathbb{E}[U]^2 \leq \mathbb{E}[U^2]$ for any real-valued random variable, we can write:

$$
\frac{1}{C^2 + \delta} \sum_{t \in \mathcal{J}_T} (f(\mathbf{w}_t) - f_\star)^2 + \eta \sum_{t \in \mathcal{I}_T} (f(\mathbf{w}_t) - f_\star) \leq \|\mathbf{w}_0 - \mathbf{w}_\star\|^2 + (T+1)\eta\varepsilon \tag{61}
$$

Since each $f(\mathbf{w}_t) - f_\star \geq 0$, the inequality (61) gives that:

$$
\eta \sum_{t \in \mathcal{I}_T} (f(\mathbf{w}_t) - f_\star) \leq \|\mathbf{w}_0 - \mathbf{w}_\star\|^2 + (T+1)\eta\varepsilon, \tag{62}
$$

and:

$$
\frac{1}{C^2 + \delta} \sum_{t \in \mathcal{J}_T} (f(\mathbf{w}_t) - f_\star)^2 \leq \|\mathbf{w}_0 - \mathbf{w}_\star\|^2 + (T+1)\eta\varepsilon. \tag{63}
$$

Using the Cauchy-Schwarz inequality, we can further write:

$$
\left( \sum_{t \in \mathcal{J}_T} f(\mathbf{w}_t) - f_\star \right)^2 \leq |\mathcal{J}_T| \sum_{t \in \mathcal{J}_T} (f(\mathbf{w}_t) - f_\star)^2. \tag{64}
$$

Therefore we have:

$$
\begin{aligned}
\sum_{t \in \mathcal{J}_T} f(\mathbf{w}_t) - f_\star &\leq \sqrt{|\mathcal{J}_T| \sum_{t \in \mathcal{J}_T} (f(\mathbf{w}_t) - f_\star)^2}, \\
&\leq \sqrt{|\mathcal{J}_T|(C^2 + \delta)\left(\|\mathbf{w}_0 - \mathbf{w}_\star\|^2 + (T+1)\eta\varepsilon\right)}.
\end{aligned}
\tag{65}
$$

We can now put together inequalities (62) and (65) by writing:

$$
\begin{aligned}
\sum_{t=0}^{T} f(\mathbf{w}_t) - f_\star \\
= \sum_{t \in \mathcal{I}_T} f(\mathbf{w}_t) - f_\star &+ \sum_{t \in \mathcal{J}_T} f(\mathbf{w}_t) - f_\star \\
\leq \frac{1}{\eta} \left( \|\mathbf{w}_0 - \mathbf{w}_\star\|^2 + (T+1)\eta\varepsilon \right) &+ \sqrt{|\mathcal{J}_T|(C^2 + \delta)\left(\|\mathbf{w}_0 - \mathbf{w}_\star\|^2 + (T+1)\eta\varepsilon\right)} \\
\leq \frac{1}{\eta} \left( \|\mathbf{w}_0 - \mathbf{w}_\star\|^2 + (T+1)\eta\varepsilon \right) &+ \sqrt{(T+1)(C^2 + \delta)\left(\|\mathbf{w}_0 - \mathbf{w}_\star\|^2 + (T+1)\eta\varepsilon\right)}
\end{aligned}
\tag{66}
$$

Dividing by $T + 1$, we obtain:

$$f\left(\frac{1}{T+1}\sum_{t=0}^{T}\mathbf{w}_t\right) - f_\star \leq \frac{1}{T+1}\sum_{t=0}^{T} f(\mathbf{w}_t) - f_\star, \quad (f \text{ is convex})$$

$$\leq \frac{\|\mathbf{w}_0 - \mathbf{w}_\star\|^2}{\eta(T+1)} + \varepsilon + \sqrt{(C^2 + \delta)\left(\frac{\|\mathbf{w}_0 - \mathbf{w}_\star\|^2}{T+1} + \eta\varepsilon\right)}, \qquad (67)$$

$$\leq \frac{\|\mathbf{w}_0 - \mathbf{w}_\star\|^2}{\eta(T+1)} + \varepsilon + \sqrt{\frac{(C^2 + \delta)\|\mathbf{w}_0 - \mathbf{w}_\star\|^2}{T+1}} + \eta\varepsilon\sqrt{C^2 + \delta}. \qquad \blacksquare$$

### D.6 THEOREM 5

**Lemma 1.** *Let $z \in \mathcal{Z}$. Assume that $\ell_z$ is convex, $\beta$-smooth and is lower-bounded on $\mathbb{R}^p$ by $\underline{\mathrm{B}} \in \mathbb{R}$. Then we have:*

$$\forall\, \mathbf{w} \in \mathbb{R}^p,\ \ell_z(\mathbf{w}) - \underline{\mathrm{B}} \geq \frac{1}{2\beta}\|\nabla\ell_z(\mathbf{w})\|^2 \qquad (68)$$

**Proof:** Let $\mathbf{w} \in \mathbb{R}^p$ and suppose that $\ell_z$ reaches its infimum at $\underline{\mathbf{w}} \in (\mathbb{R} \cup \{-\infty, +\infty\})^p$.

First, let us consider the case $\underline{\mathbf{w}} \in \mathbb{R}^p$. Then by Lemma 3.5 of Bubeck (2015), we have:

$$\ell_z(\underline{\mathbf{w}}) - \ell_z(\mathbf{w}) \leq \nabla\ell_z(\underline{\mathbf{w}})^\top(\underline{\mathbf{w}} - \mathbf{w}) - \frac{1}{2\beta}\|\nabla\ell_z(\underline{\mathbf{w}}) - \nabla\ell_z(\mathbf{w})\|^2,$$

$$= -\frac{1}{2\beta}\|\nabla\ell_z(\mathbf{w})\|^2 \quad (\nabla\ell_z(\underline{\mathbf{w}}) = \mathbf{0}). \qquad (69)$$

Therefore we can write:

$$\ell_z(\mathbf{w}) - \underline{\mathrm{B}} \geq \ell_z(\mathbf{w}) - \ell_z(\underline{\mathbf{w}}) \geq \frac{1}{2\beta}\|\nabla\ell_z(\mathbf{w})\|^2. \qquad (70)$$

Now let us assume that $\underline{\mathbf{w}} \notin \mathbb{R}^p$. Then we can construct a sequence $(\underline{\mathbf{w}}_k)_{k\in\mathbb{N}} \in (\mathbb{R}^p)^{\mathbb{N}}$ that converges to $\underline{\mathbf{w}}$. Since $\ell_z$ and $\nabla\ell_z$ are continuous functions (they are respectively convex and smooth), we have:

$$\lim_{k\to\infty}\ell_z(\underline{\mathbf{w}}_k) = \inf\ell_z,$$
$$\lim_{k\to\infty}\nabla\ell_z(\underline{\mathbf{w}}_k) = \mathbf{0}. \qquad (71)$$

Therefore the previous case gives the wanted result by using $\underline{\mathbf{w}}_k$ in place of $\underline{\mathbf{w}}$ and then taking the limit $k \to \infty$. $\blacksquare$

**Lemma 2.** *Let $z \in \mathcal{Z}$. Assume that $\ell_z$ is convex, $\beta$-smooth and is lower-bounded on $\mathbb{R}^p$ by $\underline{\mathrm{B}} \in \mathbb{R}$. Then we have:*

$$\forall\, \mathbf{w} \in \mathbb{R}^p,\ \frac{\ell_z(\mathbf{w}) - \underline{\mathrm{B}}}{\|\nabla\ell_z(\mathbf{w})\|^2 + \delta} \geq \frac{1}{2\beta} - \frac{\delta}{4\beta^2(\ell_z(\mathbf{w}) - \underline{\mathrm{B}}))} \qquad (72)$$

**Proof:**

Let $\mathbf{w} \in \mathbb{R}^p$. We apply Lemma 1 and we write successively:

$$\frac{\ell_z(\mathbf{w}) - \underline{\mathrm{B}}}{\|\nabla\ell_z(\mathbf{w})\|^2 + \delta} \geq \frac{\ell_z(\mathbf{w}) - \underline{\mathrm{B}}}{2\beta(\ell_z(\mathbf{w}) - \underline{\mathrm{B}}) + \delta}, \quad (\text{Lemma 1})$$

$$= \frac{\ell_z(\mathbf{w}) - \underline{\mathrm{B}} + \frac{\delta}{2\beta} - \frac{\delta}{2\beta}}{2\beta(\ell_z(\mathbf{w}) - \underline{\mathrm{B}} + \frac{\delta}{2\beta})},$$

$$= \frac{1}{2\beta} - \frac{\frac{\delta}{2\beta}}{2\beta(\ell_z(\mathbf{w}) - \underline{\mathrm{B}} + \frac{\delta}{2\beta})}, \qquad (73)$$

$$\geq \frac{1}{2\beta} - \frac{\delta}{4\beta^2(\ell_z(\mathbf{w}) - \underline{\mathrm{B}})}. \quad (\delta \geq 0)$$

$\blacksquare$

**Theorem 5.** *[Convex and Smooth] We assume that $\mathcal{X}$ is a convex set, and that for every $z \in \mathcal{Z}$, $\ell_z$ is convex and $\beta$-smooth. Let $\underline{\mathrm{B}}$ be a uniform lower bound on $(\mathcal{P})$ and $\mathbf{w}_\star$ be a solution of $(\mathcal{P})$. Further suppose that $\mathbf{w}_\star$ is an $\varepsilon$-interpolation for $((\mathcal{P}),\underline{\mathrm{B}})$, and that $\delta > 2\beta\varepsilon$. Then ALI-G$^\infty$ applied to $f$ satisfies:*

$$f\left(\frac{1}{T+1}\sum_{t=0}^{T}\mathbf{w}_t\right) - f_\star \leq \frac{\delta}{\beta(1 - \frac{2\beta\varepsilon}{\delta})} + \frac{2\beta}{1 - \frac{2\beta\varepsilon}{\delta}}\frac{\|\mathbf{w}_0 - \mathbf{w}_\star\|^2}{T+1}. \tag{13}$$

**Proof:**

We consider the update at time $t$, which we condition on the draw of $z_t \in \mathcal{Z}$:

$$\|\mathbf{w}_{t+1} - \mathbf{w}_\star\|^2$$
$$= \|\Pi_\Omega(\mathbf{w}_t - \gamma_t \nabla\ell_{z_t}(\mathbf{w}_t)) - \mathbf{w}_\star\|^2$$
$$\leq \|\mathbf{w}_t - \gamma_t \nabla\ell_{z_t}(\mathbf{w}_t) - \mathbf{w}_\star\|^2 \quad (\Pi_\Omega \text{ projection})$$
$$= \|\mathbf{w}_t - \mathbf{w}_\star\|^2 - 2\gamma_t \nabla\ell_{z_t}(\mathbf{w}_t)^\top(\mathbf{w}_t - \mathbf{w}_\star) + \gamma_t^2\|\nabla\ell_{z_t}(\mathbf{w}_t)\|^2$$
$$= \|\mathbf{w}_t - \mathbf{w}_\star\|^2 - 2\gamma_t \nabla\ell_{z_t}(\mathbf{w}_t)^\top(\mathbf{w}_t - \mathbf{w}_\star) + \gamma_t \frac{\ell_{z_t}(\mathbf{w}_t) - \underline{\mathrm{B}}}{\|\nabla\ell_{z_t}(\mathbf{w}_t)\|^2 + \delta}\|\nabla\ell_{z_t}(\mathbf{w}_t)\|^2$$

$$\qquad (\text{definition of } \gamma_t)$$

$$\leq \|\mathbf{w}_t - \mathbf{w}_\star\|^2 - 2\gamma_t \nabla\ell_{z_t}(\mathbf{w}_t)^\top(\mathbf{w}_t - \mathbf{w}_\star) + \gamma_t \frac{\ell_{z_t}(\mathbf{w}_t) - \underline{\mathrm{B}}}{\|\nabla\ell_{z_t}(\mathbf{w}_t)\|^2}\|\nabla\ell_{z_t}(\mathbf{w}_t)\|^2$$

$$\qquad (\text{because } \ell_{z_t}(\mathbf{w}_t) - \underline{\mathrm{B}} \geq 0 \text{ and } \delta \geq 0)$$

$$\leq \|\mathbf{w}_t - \mathbf{w}_\star\|^2 - 2\gamma_t(\ell_{z_t}(\mathbf{w}_t) - \ell_{z_t}(\mathbf{w}_\star)) + \gamma_t(\ell_{z_t}(\mathbf{w}_t) - \underline{\mathrm{B}}) \quad (\text{convexity of } \ell_{z_t})$$

$$= \|\mathbf{w}_t - \mathbf{w}_\star\|^2 - 2\gamma_t(\ell_{z_t}(\mathbf{w}_t) - \ell_{z_t}(\mathbf{w}_\star)) + \gamma_t(\ell_{z_t}(\mathbf{w}_t) - \ell_{z_t}(\mathbf{w}_\star)) + \gamma_t(\ell_{z_t}(\mathbf{w}_\star) - \underline{\mathrm{B}})$$

$$= \|\mathbf{w}_t - \mathbf{w}_\star\|^2 - \gamma_t(\ell_{z_t}(\mathbf{w}_t) - \ell_{z_t}(\mathbf{w}_\star)) + \gamma_t(\ell_{z_t}(\mathbf{w}_\star) - \underline{\mathrm{B}}) \tag{74}$$

We now lower bound $\gamma_t(\ell_{z_t}(\mathbf{w}_t) - \ell_{z_t}(\mathbf{w}_\star))$ and upper bound $\gamma_t(\ell_{z_t}(\mathbf{w}_\star) - \underline{\mathrm{B}})$ individually.

We begin with $\gamma_t(\ell_{z_t}(\mathbf{w}_t) - \ell_{z_t}(\mathbf{w}_\star))$, for which we distinguish two cases according to its sign:

Suppose $(\ell_{z_t}(\mathbf{w}_t) - \ell_{z_t}(\mathbf{w}_\star)) \geq 0$. Then we can write:

$$\gamma_t(\ell_{z_t}(\mathbf{w}_t) - \ell_{z_t}(\mathbf{w}_\star))$$
$$= \frac{\ell_{z_t}(\mathbf{w}_t) - \underline{\mathrm{B}}}{\|\nabla\ell_{z_t}(\mathbf{w}_t)\|^2 + \delta}(\ell_{z_t}(\mathbf{w}_t) - \ell_{z_t}(\mathbf{w}_\star)), \quad (\text{definition of } \gamma_t)$$
$$\geq \left(\frac{1}{2\beta} - \frac{\delta}{4\beta^2(\ell_z(\mathbf{w}_t) - \underline{\mathrm{B}})}\right)(\ell_{z_t}(\mathbf{w}_t) - \ell_{z_t}(\mathbf{w}_\star)) \quad (\text{Lemma 2, } (\ell_{z_t}(\mathbf{w}_t) - \ell_{z_t}(\mathbf{w}_\star)) \geq 0)$$
$$= \frac{1}{2\beta}(\ell_{z_t}(\mathbf{w}_t) - \ell_{z_t}(\mathbf{w}_\star)) - \frac{\delta}{4\beta^2}\frac{\ell_{z_t}(\mathbf{w}_t) - \ell_{z_t}(\mathbf{w}_\star)}{(\ell_{z_t}(\mathbf{w}_t) - \underline{\mathrm{B}})}$$
$$\geq \frac{1}{2\beta}(\ell_{z_t}(\mathbf{w}_t) - \ell_{z_t}(\mathbf{w}_\star)) - \frac{\delta}{4\beta^2} \quad (\ell_{z_t}(\mathbf{w}_\star) \geq \underline{\mathrm{B}}, (\ell_{z_t}(\mathbf{w}_t) - \ell_{z_t}(\mathbf{w}_\star)) \geq 0)$$

$$\tag{75}$$

Now suppose $(\ell_{z_t}(\mathbf{w}_t) - \ell_{z_t}(\mathbf{w}_\star)) \leq 0$, and let us show that the same result holds. We have:

$$
\begin{aligned}
\gamma_t &= \frac{\ell_{z_t}(\mathbf{w}_t) - \underline{\mathbf{B}}}{\|\nabla\ell_{z_t}(\mathbf{w}_t)\|^2 + \delta} \\
&\leq \frac{\ell_{z_t}(\mathbf{w}_\star) - \underline{\mathbf{B}}}{\|\nabla\ell_{z_t}(\mathbf{w}_t)\|^2 + \delta} \quad ((\ell_{z_t}(\mathbf{w}_t) - \ell_{z_t}(\mathbf{w}_\star)) \leq 0) \\
&\leq \frac{\varepsilon}{\|\nabla\ell_{z_t}(\mathbf{w}_t)\|^2 + \delta} \quad \text{(definition of } \varepsilon) \\
&\leq \frac{\varepsilon}{\delta} \quad (\|\nabla\ell_{z_t}(\mathbf{w}_t)\| \geq 0) \\
&\leq \frac{1}{2\beta} \quad (\delta \geq 2\beta\varepsilon)
\end{aligned}
\tag{76}
$$

Therefore we have:

$$
\begin{aligned}
&\gamma_t(\ell_{z_t}(\mathbf{w}_t) - \ell_{z_t}(\mathbf{w}_\star)) \\
&\geq \frac{1}{2\beta}(\ell_{z_t}(\mathbf{w}_t) - \ell_{z_t}(\mathbf{w}_\star)) \quad ((\ell_{z_t}(\mathbf{w}_t) - \ell_{z_t}(\mathbf{w}_\star)) \leq 0) \\
&\geq \frac{1}{2\beta}(\ell_{z_t}(\mathbf{w}_t) - \ell_{z_t}(\mathbf{w}_\star)) - \frac{\delta}{4\beta^2} \quad (\delta \geq 0)
\end{aligned}
\tag{77}
$$

In conclusion, regardless of the sign, it always holds true that:

$$
\gamma_t(\ell_{z_t}(\mathbf{w}_t) - \ell_{z_t}(\mathbf{w}_\star)) \geq \frac{1}{2\beta}(\ell_{z_t}(\mathbf{w}_t) - \ell_{z_t}(\mathbf{w}_\star)) - \frac{\delta}{4\beta^2}
\tag{78}
$$

We now upper bound $\gamma_t(\ell_{z_t}(\mathbf{w}_\star) - \underline{\mathbf{B}})$:

$$
\begin{aligned}
\gamma_t(\ell_{z_t}(\mathbf{w}_\star) - \underline{\mathbf{B}}) &= \frac{(\ell_{z_t}(\mathbf{w}_t) - \underline{\mathbf{B}})(\ell_{z_t}(\mathbf{w}_\star) - \underline{\mathbf{B}})}{\|\nabla\ell_{z_t}(\mathbf{w}_t)\|^2 + \delta}, \quad \text{(definition of } \gamma_t) \\
&\leq \frac{(\ell_{z_t}(\mathbf{w}_t) - \underline{\mathbf{B}})(\ell_{z_t}(\mathbf{w}_\star) - \underline{\mathbf{B}})}{\delta}, \quad (\|\nabla\ell_{z_t}(\mathbf{w}_t)\| \geq 0) \\
&\leq \frac{(\ell_{z_t}(\mathbf{w}_t) - \ell_{z_t}(\mathbf{w}_\star) + \varepsilon)\varepsilon}{\delta}, \quad \text{(definition of } \varepsilon \text{ twice)} \\
&= \frac{\varepsilon}{\delta}((\ell_{z_t}(\mathbf{w}_t) - \ell_{z_t}(\mathbf{w}_\star)) + \frac{\varepsilon^2}{\delta}.
\end{aligned}
\tag{79}
$$

Putting inequalities (74), (78) and (79) together, we obtain:

$$
\|\mathbf{w}_{t+1} - \mathbf{w}_\star\|^2 \leq \|\mathbf{w}_t - \mathbf{w}_\star\|^2 - \frac{1}{2\beta}(\ell_{z_t}(\mathbf{w}_t) - \ell_{z_t}(\mathbf{w}_\star)) + \frac{\delta}{4\beta^2} + \frac{\varepsilon}{\delta}((\ell_{z_t}(\mathbf{w}_t) - \ell_{z_t}(\mathbf{w}_\star)) + \frac{\varepsilon^2}{\delta}. \tag{80}
$$

Therefore we have:

$$
\left(\frac{1}{2\beta} - \frac{\varepsilon}{\delta}\right)(\ell_{z_t}(\mathbf{w}_t) - \ell_{z_t}(\mathbf{w}_\star)) - \left(\frac{\delta}{4\beta^2} + \frac{\varepsilon^2}{\delta}\right) \leq \|\mathbf{w}_t - \mathbf{w}_\star\|^2 - \|\mathbf{w}_{t+1} - \mathbf{w}_\star\|^2. \tag{81}
$$

By summing over $t$ and taking the expectation over the $z_t$, we obtain:

$$
\frac{\delta - 2\beta\varepsilon}{2\beta\delta}\sum_{t=0}^{T}\left(f(\mathbf{w}_t) - f(\mathbf{w}_\star) - \frac{\delta^2 + 4\beta^2\varepsilon^2}{4\beta^2\delta}\right) \leq \|\mathbf{w}_0 - \mathbf{w}_\star\|^2 - \mathbb{E}\left[\|\mathbf{w}_{T+1} - \mathbf{w}_\star\|^2\right] \leq \|\mathbf{w}_0 - \mathbf{w}_\star\|^2.
\tag{82}
$$

By assumption, we have that $\delta - 2\beta\varepsilon > 0$. Dividing by $T + 1$ and using the convexity of $f$, we finally obtain:

$$
\begin{aligned}
f\left(\frac{1}{T+1}\sum_{t=0}^{T}\mathbf{w}_t\right) - f_\star &\leq \frac{1}{T+1}\sum_{t=0}^{T}f(\mathbf{w}_t) - f_\star \quad \text{(convexity of } f\text{)}, \\
&= \frac{2\beta\delta}{\delta - 2\beta\varepsilon}\frac{\delta^2 + 4\beta^2\varepsilon^2}{4\beta^2\delta} + \frac{2\beta\delta}{\delta - 2\beta\varepsilon}\frac{\|\mathbf{w}_0 - \mathbf{w}_\star\|^2}{T+1}, \\
&= \frac{\delta^2 + 4\beta^2\varepsilon^2}{2\beta(\delta - 2\beta\varepsilon)} + \frac{2\beta\delta}{\delta - 2\beta\varepsilon}\frac{\|\mathbf{w}_0 - \mathbf{w}_\star\|^2}{T+1}, \\
&\leq \frac{\delta^2}{\beta(\delta - 2\beta\varepsilon)} + \frac{2\beta\delta}{\delta - 2\beta\varepsilon}\frac{\|\mathbf{w}_0 - \mathbf{w}_\star\|^2}{T+1}, \quad (\delta - 2\beta\varepsilon \geq 0) \\
&= \frac{\delta}{\beta(1 - \frac{2\beta\varepsilon}{\delta})} + \frac{2\beta}{1 - \frac{2\beta\varepsilon}{\delta}}\frac{\|\mathbf{w}_0 - \mathbf{w}_\star\|^2}{T+1}.
\end{aligned}
\tag{83}
$$

$\blacksquare$

## D.7 THEOREM 6

**Theorem 6.** *We assume that $\mathcal{X}$ is a convex set, and that for every $z \in \mathcal{Z}$, $\ell_z$ is convex and $\beta$-smooth. Let $\mathbf{w}_\star$ be an $\varepsilon$-interpolation for $((\mathcal{P}),\ \underline{\mathrm{B}})$, and suppose that $\delta > 2\beta\varepsilon$. Further assume that $\eta \geq \frac{1}{2\beta}$. Then if we apply ALI-G with a maximal learning-rate of $\eta$ to $f$, we have:*

$$
f\left(\frac{1}{T+1}\sum_{t=0}^{T}\mathbf{w}_t\right) - f_\star \leq \frac{\delta}{\beta(1 - \frac{2\beta\varepsilon}{\delta})} + \frac{2\beta}{1 - \frac{2\beta\varepsilon}{\delta}}\frac{\|\mathbf{w}_0 - \mathbf{w}_\star\|^2}{T+1}.
\tag{14}
$$

**Proof :**

By using the fact that $\gamma_t \leq \frac{\ell_{z_t}(\mathbf{w}_t) - \underline{\mathrm{B}}}{\|\nabla\ell_{z_t}(\mathbf{w}_t)\|^2 + \delta}$ rather than $\gamma_t = \frac{\ell_{z_t}(\mathbf{w}_t) - \underline{\mathrm{B}}}{\|\nabla\ell_{z_t}(\mathbf{w}_t)\|^2 + \delta}$, we can see that the inequality (74) is still valid:

$$
\|\mathbf{w}_{t+1} - \mathbf{w}_\star\|^2 \leq \|\mathbf{w}_t - \mathbf{w}_\star\|^2 - \gamma_t(\ell_{z_t}(\mathbf{w}_t) - \ell_{z_t}(\mathbf{w}_\star)) + \gamma_t(\ell_{z_t}(\mathbf{w}_\star) - \underline{\mathrm{B}})
\tag{84}
$$

As previously, we lower bound $\gamma_t(\ell_{z_t}(\mathbf{w}_t) - \ell_{z_t}(\mathbf{w}_\star))$ and upper bound $\gamma_t(\ell_{z_t}(\mathbf{w}_\star) - \underline{\mathrm{B}})$ individually.

We begin with $\gamma_t(\ell_{z_t}(\mathbf{w}_t) - \ell_{z_t}(\mathbf{w}_\star))$. We remark that either $\gamma_t = \frac{\ell_{z_t}(\mathbf{w}_t) - \underline{\mathrm{B}}}{\|\nabla\ell_{z_t}(\mathbf{w}_t)\|^2 + \delta}$ or $\gamma_t = \eta$.

Suppose $\gamma_t = \frac{\ell_{z_t}(\mathbf{w}_t) - \underline{\mathrm{B}}}{\|\nabla\ell_{z_t}(\mathbf{w}_t)\|^2 + \delta}$ and $\ell_{z_t}(\mathbf{w}_t) - \ell_{z_t}(\mathbf{w}_\star) \geq 0$. Then we are in the same condition as in Theorem 5, and thus the inequality (78) holds true:

$$
\gamma_t(\ell_{z_t}(\mathbf{w}_t) - \ell_{z_t}(\mathbf{w}_\star)) \geq \frac{1}{2\beta}(\ell_{z_t}(\mathbf{w}_t) - \ell_{z_t}(\mathbf{w}_\star)) - \frac{\delta}{4\beta^2}.
\tag{85}
$$

Now suppose $\gamma_t = \eta$ and $\ell_{z_t}(\mathbf{w}_t) - \ell_{z_t}(\mathbf{w}_\star) \geq 0$. Then we have:

$$
\begin{aligned}
\gamma_t(\ell_{z_t}(\mathbf{w}_t) - \ell_{z_t}(\mathbf{w}_\star)) &= \eta(\ell_{z_t}(\mathbf{w}_t) - \ell_{z_t}(\mathbf{w}_\star)) \\
&\geq \eta(\ell_{z_t}(\mathbf{w}_t) - \ell_{z_t}(\mathbf{w}_\star)) - \frac{\delta}{4\beta^2} \\
&\geq \frac{1}{2\beta}(\ell_{z_t}(\mathbf{w}_t) - \ell_{z_t}(\mathbf{w}_\star)) - \frac{\delta}{4\beta^2} \\
&\qquad \text{(because } \eta \geq \frac{1}{2\beta}, \ \ell_{z_t}(\mathbf{w}_t) - \ell_{z_t}(\mathbf{w}_\star) \geq 0).
\end{aligned}
\tag{86}
$$

Now suppose $\ell_{z_t}(\mathbf{w}_t) - \ell_{z_t}(\mathbf{w}_\star) \leq 0$. By using $\gamma_t \leq \frac{\ell_{z_t}(\mathbf{w}_t) - \underline{\mathbf{B}}}{\|\nabla \ell_{z_t}(\mathbf{w}_t)\|^2 + \delta}$ instead of $\gamma_t = \frac{\ell_{z_t}(\mathbf{w}_t) - \underline{\mathbf{B}}}{\|\nabla \ell_{z_t}(\mathbf{w}_t)\|^2 + \delta}$, we can see that the inequality (76) is still valid, which gives:

$$\gamma_t \leq \frac{1}{2\beta} \tag{87}$$

We now use $\ell_{z_t}(\mathbf{w}_t) - \ell_{z_t}(\mathbf{w}_\star) \leq 0$ to write:

$$
\begin{aligned}
\gamma_t \left( \ell_{z_t}(\mathbf{w}_t) - \ell_{z_t}(\mathbf{w}_\star) \right) &\geq \frac{1}{2\beta} (\ell_{z_t}(\mathbf{w}_t) - \ell_{z_t}(\mathbf{w}_\star)) \quad (\ell_{z_t}(\mathbf{w}_t) - \ell_{z_t}(\mathbf{w}_\star) \leq 0) \\
&\geq \frac{1}{2\beta} (\ell_{z_t}(\mathbf{w}_t) - \ell_{z_t}(\mathbf{w}_\star)) - \frac{\delta}{4\beta^2}
\end{aligned}
\tag{88}
$$

In conclusion, in all cases, it holds true that:

$$\gamma_t(\ell_{z_t}(\mathbf{w}_t) - \ell_{z_t}(\mathbf{w}_\star)) \geq \frac{1}{2\beta}(\ell_{z_t}(\mathbf{w}_t) - \ell_{z_t}(\mathbf{w}_\star)) - \frac{\delta}{4\beta^2} \tag{89}$$

By using $\gamma_t \leq \frac{\ell_{z_t}(\mathbf{w}_t) - \underline{\mathbf{B}}}{\|\nabla \ell_{z_t}(\mathbf{w}_t)\|^2 + \delta}$, we can remark that the inequality (79) holds true and gives:

$$\gamma_t(\ell_{z_t}(\mathbf{w}_\star) - \underline{\mathbf{B}}) \leq \frac{\varepsilon}{\delta}((\ell_{z_t}(\mathbf{w}_t) - \ell_{z_t}(\mathbf{w}_\star)) + \frac{\varepsilon^2}{\delta}. \tag{90}$$

We now put together inequalities (84), (89) and (90):

$$
\begin{aligned}
\|\mathbf{w}_{t+1} - \mathbf{w}_\star\|^2 &\leq \|\mathbf{w}_t - \mathbf{w}_\star\|^2 - \frac{1}{2\beta}(\ell_{z_t}(\mathbf{w}_t) - \ell_{z_t}(\mathbf{w}_\star)) + \frac{\delta}{4\beta^2} + \frac{\varepsilon}{\delta}((\ell_{z_t}(\mathbf{w}_t) - \ell_{z_t}(\mathbf{w}_\star)) + \frac{\varepsilon^2}{\delta}, \\
&= \|\mathbf{w}_t - \mathbf{w}_\star\|^2 - \left(\frac{1}{2\beta} - \frac{\varepsilon}{\delta}\right)(\ell_{z_t}(\mathbf{w}_t) - \ell_{z_t}(\mathbf{w}_\star)) + \frac{\delta}{4\beta^2} + \frac{\varepsilon^2}{\delta}.
\end{aligned}
\tag{91}
$$

This is exactly the same result as in the inequality (81) from the proof of Theorem 5. Therefore the rest of the proof of Theorem 5 follows and we obtain the desired result. ∎

## D.8 THEOREM 7

**Theorem 7.** *We assume that $\mathcal{X}$ is a convex set, and that for every $z \in \mathcal{Z}$, $\ell_z$ is convex and $\beta$-smooth. Let $\mathbf{w}_\star$ be an $\varepsilon$-interpolation for $((\mathcal{P}),\ \underline{\mathbf{B}})$, and suppose that $\delta > 2\beta\varepsilon$. Further assume that $\eta \leq \frac{1}{2\beta}$. Then if we apply ALI-G with a maximal learning-rate of $\eta$ to $f$, we have:*

$$f\left(\frac{1}{T+1} \sum_{t=0}^{T} \mathbf{w}_t\right) - f_\star \leq \frac{\|\mathbf{w}_0 - \mathbf{w}_\star\|^2}{\eta(T+1)} + \frac{\delta}{2\beta} + \varepsilon. \tag{15}$$

**Proof:**

By using the fact that $\gamma_t \leq \frac{\ell_{z_t}(\mathbf{w}_t) - \underline{\mathbf{B}}}{\|\nabla \ell_{z_t}(\mathbf{w}_t)\|^2 + \delta}$ rather than $\gamma_t = \frac{\ell_{z_t}(\mathbf{w}_t) - \underline{\mathbf{B}}}{\|\nabla \ell_{z_t}(\mathbf{w}_t)\|^2 + \delta}$, we can see that the inequality (74) is still valid:

$$\|\mathbf{w}_{t+1} - \mathbf{w}_\star\|^2 \leq \|\mathbf{w}_t - \mathbf{w}_\star\|^2 - \gamma_t(\ell_{z_t}(\mathbf{w}_t) - \ell_{z_t}(\mathbf{w}_\star)) + \gamma_t(\ell_{z_t}(\mathbf{w}_\star) - \underline{\mathbf{B}}) \tag{92}$$

As previously, we lower bound $\gamma_t(\ell_{z_t}(\mathbf{w}_t) - \ell_{z_t}(\mathbf{w}_\star))$ and upper bound $\gamma_t(\ell_{z_t}(\mathbf{w}_\star) - \underline{\mathbf{B}})$ individually.

We begin with $\gamma_t(\ell_{z_t}(\mathbf{w}_t) - \ell_{z_t}(\mathbf{w}_\star))$. We remark that either $\gamma_t = \frac{\ell_{z_t}(\mathbf{w}_t) - \underline{\mathbf{B}}}{\|\nabla \ell_{z_t}(\mathbf{w}_t)\|^2 + \delta}$ or $\gamma_t = \eta$.

Suppose $\gamma_t = \frac{\ell_{z_t}(\mathbf{w}_t) - \underline{\mathbf{B}}}{\|\nabla\ell_{z_t}(\mathbf{w}_t)\|^2 + \delta}$ and $\ell_{z_t}(\mathbf{w}_t) - \ell_{z_t}(\mathbf{w}_\star) \geq 0$. First we write:

$$
\begin{aligned}
\gamma_t &= \frac{\ell_{z_t}(\mathbf{w}_t) - \underline{\mathbf{B}}}{\|\ell_{z_t}(\mathbf{w}_t)\|^2 + \delta} \\
&= \frac{\ell_{z_t}(\mathbf{w}_t) - \underline{\mathbf{B}} + \frac{\delta}{2\beta}}{\|\ell_{z_t}(\mathbf{w}_t)\|^2 + \delta} - \frac{\frac{\delta}{2\beta}}{\|\ell_{z_t}(\mathbf{w}_t)\|^2 + \delta} \\
&\geq \frac{\frac{\|\ell_{z_t}(\mathbf{w}_t)\|^2}{2\beta} + \frac{\delta}{2\beta}}{\|\ell_{z_t}(\mathbf{w}_t)\|^2 + \delta} - \frac{\delta}{2\beta}\frac{1}{\|\ell_{z_t}(\mathbf{w}_t)\|^2 + \delta} \quad \text{(Lemma 1)} \\
&= \frac{1}{2\beta} - \frac{\delta}{2\beta}\frac{1}{\|\ell_{z_t}(\mathbf{w}_t)\|^2 + \delta} \\
&\geq \eta - \frac{\delta}{2\beta}\frac{1}{\|\ell_{z_t}(\mathbf{w}_t)\|^2 + \delta} \quad (\eta \leq \frac{1}{2\beta})
\end{aligned}
\tag{93}
$$

Since $\ell_{z_t}(\mathbf{w}_t) - \ell_{z_t}(\mathbf{w}_\star) \geq 0$, this yields:

$$
\begin{aligned}
\gamma_t(\ell_{z_t}(\mathbf{w}_t) - \ell_{z_t}(\mathbf{w}_\star)) &\geq \left(\eta - \frac{\delta}{2\beta}\frac{1}{\|\ell_{z_t}(\mathbf{w}_t)\|^2 + \delta}\right)(\ell_{z_t}(\mathbf{w}_t) - \ell_{z_t}(\mathbf{w}_\star)) \\
&= \eta(\ell_{z_t}(\mathbf{w}_t) - \ell_{z_t}(\mathbf{w}_\star)) - \frac{\delta}{2\beta}\frac{\ell_{z_t}(\mathbf{w}_t) - \ell_{z_t}(\mathbf{w}_\star)}{\|\ell_{z_t}(\mathbf{w}_t)\|^2 + \delta} \\
&\geq \eta(\ell_{z_t}(\mathbf{w}_t) - \ell_{z_t}(\mathbf{w}_\star)) - \frac{\delta}{2\beta}\frac{\ell_{z_t}(\mathbf{w}_t) - \underline{\mathbf{B}}}{\|\ell_{z_t}(\mathbf{w}_t)\|^2 + \delta} \quad (\ell_{z_t}(\mathbf{w}_\star) \geq \underline{\mathbf{B}})
\end{aligned}
\tag{94}
$$

We now notice that since $\gamma_t = \frac{\ell_{z_t}(\mathbf{w}_t) - \underline{\mathbf{B}}}{\|\nabla\ell_{z_t}(\mathbf{w}_t)\|^2 + \delta}$, and $\gamma_t \leq \eta$, then necessarily $\frac{\ell_{z_t}(\mathbf{w}_t) - \underline{\mathbf{B}}}{\|\nabla\ell_{z_t}(\mathbf{w}_t)\|^2 + \delta} \leq \eta$. This gives:

$$
\gamma_t(\ell_{z_t}(\mathbf{w}_t) - \ell_{z_t}(\mathbf{w}_\star)) \geq \eta(\ell_{z_t}(\mathbf{w}_t) - \ell_{z_t}(\mathbf{w}_\star)) - \frac{\eta\delta}{2\beta}
\tag{95}
$$

Now suppose $\gamma_t = \eta$ and $\ell_{z_t}(\mathbf{w}_t) - \ell_{z_t}(\mathbf{w}_\star) \geq 0$. Then we have:

$$
\begin{aligned}
\gamma_t(\ell_{z_t}(\mathbf{w}_t) - \ell_{z_t}(\mathbf{w}_\star)) &= \eta(\ell_{z_t}(\mathbf{w}_t) - \ell_{z_t}(\mathbf{w}_\star)) \\
&\geq \eta(\ell_{z_t}(\mathbf{w}_t) - \ell_{z_t}(\mathbf{w}_\star)) - \frac{\eta\delta}{2\beta}.
\end{aligned}
\tag{96}
$$

Now suppose $\ell_{z_t}(\mathbf{w}_t) - \ell_{z_t}(\mathbf{w}_\star) \leq 0$. Since $\gamma_t \leq \eta$ by definition, we have that:

$$
\begin{aligned}
\gamma_t(\ell_{z_t}(\mathbf{w}_t) - \ell_{z_t}(\mathbf{w}_\star)) &\geq \eta(\ell_{z_t}(\mathbf{w}_t) - \ell_{z_t}(\mathbf{w}_\star)) \quad (\ell_{z_t}(\mathbf{w}_t) - \ell_{z_t}(\mathbf{w}_\star) \leq 0) \\
&\geq \eta(\ell_{z_t}(\mathbf{w}_t) - \ell_{z_t}(\mathbf{w}_\star)) - \frac{\eta\delta}{2\beta}.
\end{aligned}
\tag{97}
$$

In conclusion, in all cases, it holds true that:

$$
\gamma_t(\ell_{z_t}(\mathbf{w}_t) - \ell_{z_t}(\mathbf{w}_\star)) \geq \eta(\ell_{z_t}(\mathbf{w}_t) - \ell_{z_t}(\mathbf{w}_\star)) - \frac{\eta\delta}{2\beta}
\tag{98}
$$

We upper bound $\gamma_t(\ell_{z_t}(\mathbf{w}_\star) - \underline{\mathbf{B}})$ as follows:

$$
\begin{aligned}
\gamma_t(\ell_{z_t}(\mathbf{w}_\star) - \underline{\mathbf{B}}) &\leq \eta(\ell_{z_t}(\mathbf{w}_\star) - \underline{\mathbf{B}}) \quad (\ell_{z_t}(\mathbf{w}_\star) \geq \underline{\mathbf{B}}) \\
&\leq \eta\varepsilon \quad \text{(definition of } \varepsilon)
\end{aligned}
\tag{99}
$$

We combine inequalities (92), (98) and (99) and obtain:

$$
\|\mathbf{w}_{t+1} - \mathbf{w}_\star\|^2 \leq \|\mathbf{w}_t - \mathbf{w}_\star\|^2 - \eta(\ell_{z_t}(\mathbf{w}_t) - \ell_{z_t}(\mathbf{w}_\star)) + \frac{\eta\delta}{2\beta} + \eta\varepsilon.
\tag{100}
$$

By taking the expectation and using a telescopic sum, we obtain:

$$
0 \leq \|\mathbf{w}_{T+1} - \mathbf{w}_\star\|^2 \leq \|\mathbf{w}_0 - \mathbf{w}_\star\|^2 - \sum_{t=0}^{T}\left(\eta(f(\mathbf{w}_t) - f_\star) + \frac{\eta\delta}{2\beta} + \eta\varepsilon\right).
\tag{101}
$$

Re-arranging and using the convexity of $f$, we finally obtain:

$$f\left(\frac{1}{T+1}\sum_{t=0}^{T}\mathbf{w}_t\right) \leq \frac{\|\mathbf{w}_0 - \mathbf{w}_\star\|^2}{\eta(T+1)} + \frac{\delta}{2\beta} + \varepsilon. \tag{102}$$

∎

### D.9 THEOREM 8

**Lemma 3.** *For any $a, b \in \mathbb{R}^p$, we have that:*

$$\|a\|^2 + \|b\|^2 \geq \frac{1}{2}\|a - b\|^2 \tag{103}$$

**Proof :** This is a simple application of the parallelogram law, but we give the proof here for completeness.

$$\begin{aligned}
\|a\|^2 + \|b\|^2 - \frac{1}{2}\|a - b\|^2 &= \|a\|^2 + \|b\|^2 - \frac{1}{2}\|a\|^2 - \frac{1}{2}\|b\|^2 + a^\top b \\
&= \frac{1}{2}\|a\|^2 + \frac{1}{2}\|b\|^2 + a^\top b \\
&= \frac{1}{2}\|a + b\|^2 \\
&\geq 0
\end{aligned}$$

∎

**Lemma 4.** *Let $z \in \mathcal{Z}$. Assume that $\ell_z$ is $\alpha$-strongly convex and is lower-bounded on $\mathbb{R}^p$ by $\underline{\mathrm{B}} \in \mathbb{R}$ such that $\inf \ell_z - \underline{\mathrm{B}} \leq \varepsilon$. In addition, suppose that $\delta \geq 2\alpha\varepsilon$. Then we have:*

$$\forall\, \mathbf{w} \in \mathbb{R}^p, \quad \frac{\ell_z(\mathbf{w}) - \underline{\mathrm{B}}}{\|\nabla\ell_z(\mathbf{w})\|^2 + \delta} \leq \frac{1}{2\alpha}. \tag{104}$$

**Proof :**

Let $\mathbf{w} \in \mathbb{R}^p$ and suppose that $\ell_z$ reaches its minimum at $\underline{\mathbf{w}} \in \mathbb{R}^p$ (this minimum exists because of strong convexity). By definition of strong convexity, we have that:

$$\forall\, \hat{\mathbf{w}} \in \mathbb{R}^p, \ \ell_z(\hat{\mathbf{w}}) \geq \ell_z(\mathbf{w}) + \nabla\ell_z(\mathbf{w})^\top(\hat{\mathbf{w}} - \mathbf{w}) + \frac{\alpha}{2}\|\hat{\mathbf{w}} - \mathbf{w}\|^2 \tag{105}$$

We minimize the right hand-side over $\hat{\mathbf{w}}$, which gives:

$$\forall\hat{\mathbf{w}} \in \mathbb{R}^p, \ \ell_z(\hat{\mathbf{w}}) \geq \ell_z(\mathbf{w}) + \nabla\ell_z(\mathbf{w})^\top(\hat{\mathbf{w}} - \mathbf{w}) + \frac{\alpha}{2}\|\hat{\mathbf{w}} - \mathbf{w}\|^2 \geq \ell_z(\mathbf{w}) - \frac{1}{2\alpha}\|\nabla\ell_z(\mathbf{w})\|^2 \tag{106}$$

Thus by choosing $\hat{\mathbf{w}} = \underline{\mathbf{w}}$ and re-ordering, we obtain the following result (a.k.a. the Polyak-Lojasiewicz inequality):

$$\ell_z(\mathbf{w}) - \ell_z(\underline{\mathbf{w}}) \leq \frac{1}{2\alpha}\|\nabla\ell_z(\mathbf{w})\|^2 \tag{107}$$

Therefore we can write:

$$\frac{\ell_z(\mathbf{w}) - \underline{\mathrm{B}}}{\|\nabla\ell_z(\mathbf{w})\|^2 + \delta} \leq \frac{\ell_z(\mathbf{w}) - \ell_z(\underline{\mathbf{w}}) + \varepsilon}{\|\nabla\ell_z(\mathbf{w})\|^2 + \delta} \leq \frac{\frac{1}{2\alpha}\|\nabla\ell_z(\mathbf{w})\|^2 + \varepsilon}{\|\nabla\ell_z(\mathbf{w})\|^2 + \delta}. \tag{108}$$

We introduce the function $\psi : x \in \mathbb{R}^+ \mapsto \dfrac{\frac{1}{2\alpha}x + \varepsilon}{x + \delta}$, and we compute its derivative:

$$\begin{aligned}
\psi'(x) &= \frac{\frac{1}{2\alpha}(x + \delta) - \frac{1}{2\alpha}x - \varepsilon}{(x + \delta)^2}, \\
&= \frac{\frac{\delta}{2\alpha} - \varepsilon}{(x + \delta)^2} \geq 0. \quad (\delta \geq 2\alpha\varepsilon)
\end{aligned} \tag{109}$$

Therefore $\psi$ is monotonically increasing. As a result, we have:

$$\forall\, x \in \mathbb{R}^+, \ \psi(x) \leq \lim_{x \to \infty} \psi(x) = \frac{1}{2\alpha}. \tag{110}$$

Therefore we have that:

$$\frac{\frac{1}{2\alpha}\|\nabla \ell_z(\mathbf{w})\|^2 + \varepsilon}{\|\nabla \ell_z(\mathbf{w})\|^2 + \delta} = \psi\left(\|\nabla \ell_z(\mathbf{w})\|^2\right) \le \frac{1}{2\alpha}, \tag{111}$$

which concludes the proof. ∎

**Theorem 8.** *[Strongly Convex and Smooth] We assume that $\mathcal{X}$ is a convex set, and that for every $z \in \mathcal{Z}$, $\ell_z$ is $\alpha$-strongly convex and $\beta$-smooth. Let $\underline{\mathbf{B}}$ be a uniform lower bound on $(\mathcal{P})$ and $\mathbf{w}_\star$ be a solution of $(\mathcal{P})$. Further suppose that $\mathbf{w}_\star$ is an $\varepsilon$-interpolation for $((\mathcal{P}), \underline{\mathbf{B}})$, and that $\delta > 2\beta\varepsilon$. Then ALI-G$^\infty$ applied to $f$ satisfies:*

$$f(\mathbf{w}_{T+1}) - f_\star \le \beta \exp\left(-\frac{\alpha T}{8\beta}\right) \|\mathbf{w}_0 - \mathbf{w}_\star\|^2 + \frac{2\delta}{\alpha} + \left(10\frac{\beta}{\alpha} + 4\frac{\beta^2}{\alpha^2}\right)\varepsilon. \tag{16}$$

*In other words, $f$ approximately converges to $f_\star$ at a rate of $\mathcal{O}(\exp(-\alpha T/8\beta))$.*

**Proof:**

We condition the update on $z_t$ drawn at random. The beginning of the proof is identical to that of Theorem 5 (and in particular requires $\delta > 2\beta\varepsilon$). In addition, we remark that $\delta > 2\beta\varepsilon \ge 2\alpha\varepsilon$, because it always holds true that $\beta \ge \alpha$. Combining inequalities (74) and (78), we obtain:

$$\|\mathbf{w}_{t+1} - \mathbf{w}_\star\|^2 \le \|\mathbf{w}_t - \mathbf{w}_\star\|^2 - \frac{1}{2\beta}(\ell_{z_t}(\mathbf{w}_t) - \ell_{z_t}(\mathbf{w}_\star)) + \frac{\delta}{4\beta^2} + \gamma_t(\ell_{z_t}(\mathbf{w}_\star) - \underline{\mathbf{B}}),$$

$$\le \|\mathbf{w}_t - \mathbf{w}_\star\|^2 - \frac{1}{2\beta}(\ell_{z_t}(\mathbf{w}_t) - \ell_{z_t}(\mathbf{w}_\star)) + \frac{\delta}{4\beta^2} + \gamma_t\varepsilon, \quad \text{(definition of } \varepsilon)$$

$$\le \|\mathbf{w}_t - \mathbf{w}_\star\|^2 - \frac{1}{2\beta}(\ell_{z_t}(\mathbf{w}_t) - \ell_{z_t}(\mathbf{w}_\star)) + \frac{\delta}{4\beta^2} + \frac{\varepsilon}{2\alpha}. \quad \text{(Lemma 4)} \tag{112}$$

Let $\underline{\mathbf{w}}^{(z_t)}$ be the minimizer of $\ell_{z_t}$ on its unconstrained domain $\mathbb{R}^p$ (its existence is guaranteed by the strong convexity property). Then we exploit strong convexity to lower bound the progress made:

$$\ell_{z_t}(\mathbf{w}_t) - \ell_{z_t}(\mathbf{w}_\star) = \ell_{z_t}(\mathbf{w}_t) - \ell_{z_t}(\underline{\mathbf{w}}^{(z_t)}) + \ell_{z_t}(\mathbf{w}_\star) - \ell_{z_t}(\underline{\mathbf{w}}^{(z_t)}) - 2(\ell_{z_t}(\mathbf{w}_\star) - \ell_{z_t}(\underline{\mathbf{w}}^{(z_t)}))$$

$$\ge \ell_{z_t}(\mathbf{w}_t) - \ell_{z_t}(\underline{\mathbf{w}}^{(z_t)}) + \ell_{z_t}(\mathbf{w}_\star) - \ell_{z_t}(\underline{\mathbf{w}}^{(z_t)}) - 2(\ell_{z_t}(\mathbf{w}_\star) - \underline{\mathbf{B}})$$

$$\text{(because } \ell_{z_t}(\underline{\mathbf{w}}^{(z_t)}) \ge \underline{\mathbf{B}})$$

$$\ge \ell_{z_t}(\mathbf{w}_t) - \ell_{z_t}(\underline{\mathbf{w}}^{(z_t)}) + \ell_{z_t}(\mathbf{w}_\star) - \ell_{z_t}(\underline{\mathbf{w}}^{(z_t)}) - 2\varepsilon \quad \text{(definition of } \varepsilon)$$

$$\ge \frac{\alpha}{2}\|\mathbf{w}_t - \underline{\mathbf{w}}^{(z_t)}\|^2 + \frac{\alpha}{2}\|\mathbf{w}_\star - \underline{\mathbf{w}}^{(z_t)}\|^2 - 2\varepsilon \quad (\alpha\text{-strong convexity})$$

$$\ge \frac{\alpha}{4}\|\mathbf{w}_t - \mathbf{w}_\star\|^2 - 2\varepsilon \quad \text{(Lemma 3)} \tag{113}$$

We now combine inequalities (112) and (113):

$$\|\mathbf{w}_{t+1} - \mathbf{w}_\star\|^2 \le \left(1 - \frac{\alpha}{8\beta}\right)\|\mathbf{w}_t - \mathbf{w}_\star\|^2 + \frac{\varepsilon}{\beta} + \frac{\delta}{4\beta^2} + \frac{\varepsilon}{2\alpha} \tag{114}$$

We use a trivial induction over $t$ and write:

$$\|\mathbf{w}_{t+1} - \mathbf{w}_\star\|^2 \le \left(1 - \frac{\alpha}{8\beta}\right)\|\mathbf{w}_t - \mathbf{w}_\star\|^2 + \frac{\varepsilon}{\beta} + \frac{\delta}{4\beta^2} + \frac{\varepsilon}{2\alpha},$$

$$\le \left(1 - \frac{\alpha}{8\beta}\right)^t \|\mathbf{w}_0 - \mathbf{w}_\star\|^2 + \sum_{k=0}^t \left(1 - \frac{\alpha}{8\beta}\right)^{t-k}\left(\frac{\varepsilon}{\beta} + \frac{\delta}{4\beta^2} + \frac{\varepsilon}{2\alpha}\right),$$

$$\le \left(1 - \frac{\alpha}{8\beta}\right)^t \|\mathbf{w}_0 - \mathbf{w}_\star\|^2 + \sum_{k=0}^{\infty} \left(1 - \frac{\alpha}{8\beta}\right)^{k}\left(\frac{\varepsilon}{\beta} + \frac{\delta}{4\beta^2} + \frac{\varepsilon}{2\alpha}\right),$$

$$= \left(1 - \frac{\alpha}{8\beta}\right)^t \|\mathbf{w}_0 - \mathbf{w}_\star\|^2 + \frac{1}{\frac{\alpha}{8\beta}}\left(\frac{\varepsilon}{\beta} + \frac{\delta}{4\beta^2} + \frac{\varepsilon}{2\alpha}\right),$$

$$= \left(1 - \frac{\alpha}{8\beta}\right)^t \|\mathbf{w}_0 - \mathbf{w}_\star\|^2 + \frac{8\beta}{\alpha}\left(\frac{\varepsilon}{\beta} + \frac{\delta}{4\beta^2} + \frac{\varepsilon}{2\alpha}\right). \tag{115}$$

In particular, we remark that the right hand-side of the equation is independent on $z_t$.

Given an arbitrary $\mathbf{w} \in \mathbb{R}^p$, we now wish to relate the distance $\|\mathbf{w} - \mathbf{w}_\star\|^2$ to the function values $f(\mathbf{w}) - f(\mathbf{w}_\star)$.

Since each $\ell_z$ is $\alpha$-strongly convex and $\beta$-smooth, so is $f = \mathbb{E}_z[\ell_z]$. We introduce $\underline{\mathbf{w}}$ the minimizer of $f$ on its unconstrained domain $\mathbb{R}^p$. Then we can write that for any $\mathbf{w} \in \mathbb{R}^p$:

$$
\begin{aligned}
f(\mathbf{w}) - f(\mathbf{w}_\star) &\leq f(\mathbf{w}) - f(\underline{\mathbf{w}}), \quad (f(\underline{\mathbf{w}}) \leq f(\mathbf{w}_\star)) \\
&\leq \nabla f(\underline{\mathbf{w}})^\top (\mathbf{w} - \underline{\mathbf{w}}) + \frac{\beta}{2}\|\mathbf{w} - \underline{\mathbf{w}}\|^2, \quad (f \text{ is } \beta\text{-smooth}) \\
&= \frac{\beta}{2}\|\mathbf{w} - \underline{\mathbf{w}}\|^2, \quad (\nabla f(\underline{\mathbf{w}}) = \mathbf{0}) \\
&\leq \beta(\|\mathbf{w} - \mathbf{w}_\star\|^2 + \|\mathbf{w}_\star - \underline{\mathbf{w}}\|^2), \quad (\text{Lemma 3}) \\
&\leq \beta\|\mathbf{w} - \mathbf{w}_\star\|^2 + \frac{2\beta}{\alpha}\left(f(\mathbf{w}_\star) - f(\underline{\mathbf{w}})\right), \quad (f \text{ is } \alpha\text{-strongly convex}) \\
&\leq \beta\|\mathbf{w} - \mathbf{w}_\star\|^2 + \frac{2\beta}{\alpha}\left(f(\mathbf{w}_\star) - \underline{\mathrm{B}}\right), \quad (\underline{\mathrm{B}} \leq f(\underline{\mathbf{w}}) \text{ by definition}) \\
&\leq \beta\|\mathbf{w} - \mathbf{w}_\star\|^2 + 2\frac{\beta\varepsilon}{\alpha}, \quad (\text{definition of } \varepsilon)
\end{aligned}
\tag{116}
$$

We combine the results to obtain the final result:

$$
\begin{aligned}
f(\mathbf{w}_{t+1}) - f(\mathbf{w}_\star) &\leq \beta\|\mathbf{w}_{t+1} - \mathbf{w}_\star\|^2 + 2\frac{\beta\varepsilon}{\alpha}, \\
&\leq \beta\left(\left(1 - \frac{\alpha}{8\beta}\right)^t \|\mathbf{w}_0 - \mathbf{w}_\star\|^2 + \frac{8\beta}{\alpha}\left(\frac{\varepsilon}{\beta} + \frac{\delta}{4\beta^2} + \frac{\varepsilon}{2\alpha}\right)\right) + 2\frac{\beta\varepsilon}{\alpha}, \\
&= \beta\left(1 - \frac{\alpha}{8\beta}\right)^t \|\mathbf{w}_0 - \mathbf{w}_\star\|^2 + \frac{8\beta}{\alpha}\left(\varepsilon + \frac{\delta}{4\beta} + \frac{\varepsilon\beta}{2\alpha}\right) + 2\frac{\beta\varepsilon}{\alpha}, \\
&= \beta\left(1 - \frac{\alpha}{8\beta}\right)^t \|\mathbf{w}_0 - \mathbf{w}_\star\|^2 + \frac{2\delta}{\alpha} + \left(10\frac{\beta}{\alpha} + 4\frac{\beta^2}{\alpha^2}\right)\varepsilon, \\
&\leq \beta\exp\left(-\frac{\alpha t}{8\beta}\right)\|\mathbf{w}_0 - \mathbf{w}_\star\|^2 + \frac{2\delta}{\alpha} + \left(10\frac{\beta}{\alpha} + 4\frac{\beta^2}{\alpha^2}\right)\varepsilon.
\end{aligned}
\tag{117}
$$

∎

## D.10 Theorem 9

**Theorem 9.** *We assume that $\mathcal{X}$ is a convex set, and that for every $z \in \mathcal{Z}$, $\ell_z$ is $\alpha$-strongly convex and $\beta$-smooth. Let $\mathbf{w}_\star$ be an $\varepsilon$-interpolation for $((\mathcal{P}), \underline{\mathrm{B}})$, and suppose that $\delta > 2\beta\varepsilon$. Further assume that $\eta \geq \frac{1}{2\beta}$. Then if we apply ALI-G with a maximal learning-rate of $\eta$ to $f$, we have:*

$$f(\mathbf{w}_{T+1}) - f_\star \leq \beta\exp\left(-\frac{\alpha T}{8\beta}\right)\|\mathbf{w}_0 - \mathbf{w}_\star\|^2 + \frac{2\delta}{\alpha} + \left(10\frac{\beta}{\alpha} + 4\frac{\beta^2}{\alpha^2}\right)\varepsilon. \tag{17}$$

**Proof:** Re-using inequalities (84) and (89) from the proof of Theorem 6, we obtain:

$$\|\mathbf{w}_{t+1} - \mathbf{w}_\star\|^2 \leq \|\mathbf{w}_t - \mathbf{w}_\star\|^2 - \frac{1}{2\beta}(\ell_{z_t}(\mathbf{w}_t) - \ell_{z_t}(\mathbf{w}_\star)) + \frac{\delta}{4\beta^2} + \gamma_t(\ell_{z_t}(\mathbf{w}_\star) - \underline{\mathrm{B}}). \tag{118}$$

This is exactly the same result as the first line of the inequality (112) in the proof of Theorem 8. Then the rest of the proof is identical to the one of Theorem 8. ∎

## D.11 Theorem 10

**Theorem 10.** *We assume that $\mathcal{X}$ is a convex set, and that for every $z \in \mathcal{Z}$, $\ell_z$ is $\alpha$-strongly convex and $\beta$-smooth. Let $\mathbf{w}_\star$ be an $\varepsilon$-interpolation for $((\mathcal{P}), \underline{\mathrm{B}})$, and suppose that $\delta > 2\beta\varepsilon$. Further assume*

*that $\eta \leq \frac{1}{2\beta}$. Then if we apply ALI-G with a maximal learning-rate of $\eta$ to $f$, we have:*

$$f(\mathbf{w}_{T+1}) - f_\star \leq \beta \exp\left(-\frac{\alpha\eta T}{4}\right) \|\mathbf{w}_0 - \mathbf{w}_\star\|^2 + \frac{2\delta}{\alpha} + \frac{14\varepsilon\beta}{\alpha}. \tag{18}$$

**Proof :** Re-using inequalities (92) and (98) from the proof of Theorem 7, we can write:

$$\|\mathbf{w}_{t+1} - \mathbf{w}_\star\|^2 \leq \|\mathbf{w}_t - \mathbf{w}_\star\|^2 - \eta(\ell_{z_t}(\mathbf{w}_t) - \ell_{z_t}(\mathbf{w}_\star)) + \frac{\eta\delta}{2\beta} + \gamma_t(\ell_{z_t}(\mathbf{w}_\star) - \underline{\mathrm{B}}),$$

$$\leq \|\mathbf{w}_t - \mathbf{w}_\star\|^2 - \eta(\ell_{z_t}(\mathbf{w}_t) - \ell_{z_t}(\mathbf{w}_\star)) + \frac{\eta\delta}{2\beta} + \eta\varepsilon \quad (\gamma_t \leq \eta, 0 \leq \ell_{z_t}(\mathbf{w}_\star) - \underline{\mathrm{B}} \leq \varepsilon). \tag{119}$$

Furthermore, the inequality (113) gives:

$$\ell_{z_t}(\mathbf{w}_t) - \ell_{z_t}(\mathbf{w}_\star) \geq \frac{\alpha}{4}\|\mathbf{w}_t - \mathbf{w}_\star\|^2 - 2\varepsilon \tag{120}$$

Therefore, we can write:

$$\|\mathbf{w}_{t+1} - \mathbf{w}_\star\|^2 \leq \|\mathbf{w}_t - \mathbf{w}_\star\|^2 - \frac{\alpha\eta}{4}\|\mathbf{w}_t - \mathbf{w}_\star\|^2 + \frac{\eta\delta}{2\beta} + 3\eta\varepsilon,$$

$$= \left(1 - \frac{\alpha\eta}{4}\right)\|\mathbf{w}_t - \mathbf{w}_\star\|^2 + \frac{\eta\delta}{2\beta} + 3\eta\varepsilon. \tag{121}$$

Then a trivial induction gives that:

$$\|\mathbf{w}_{T+1} - \mathbf{w}_\star\|^2 \leq \left(1 - \frac{\alpha\eta}{4}\right)^T \|\mathbf{w}_0 - \mathbf{w}_\star\|^2 + \left(\frac{\eta\delta}{2\beta} + 3\eta\varepsilon\right)\sum_{t=0}^{T}\left(1 - \frac{\alpha\eta}{4}\right)^t,$$

$$\leq \left(1 - \frac{\alpha\eta}{4}\right)^T \|\mathbf{w}_0 - \mathbf{w}_\star\|^2 + \left(\frac{\eta\delta}{2\beta} + 3\eta\varepsilon\right)\sum_{t=0}^{\infty}\left(1 - \frac{\alpha\eta}{4}\right)^t,$$

$$= \left(1 - \frac{\alpha\eta}{4}\right)^T \|\mathbf{w}_0 - \mathbf{w}_\star\|^2 + \left(\frac{\eta\delta}{2\beta} + 3\eta\varepsilon\right)\frac{1}{1 - \left(1 - \frac{\alpha\eta}{4}\right)}, \tag{122}$$

$$= \left(1 - \frac{\alpha\eta}{4}\right)^T \|\mathbf{w}_0 - \mathbf{w}_\star\|^2 + \frac{2\delta}{\alpha\beta} + \frac{12\varepsilon}{\alpha}.$$

We now re-use the inequality (116) to write:

$$f(\mathbf{w}_{T+1}) - f_\star \leq \beta\|\mathbf{w}_{T+1} - \mathbf{w}_\star\|^2 + \frac{2\beta\varepsilon}{\alpha},$$

$$\leq \beta\left(1 - \frac{\alpha\eta}{4}\right)^T \|\mathbf{w}_0 - \mathbf{w}_\star\|^2 + \frac{2\delta}{\alpha} + \frac{14\varepsilon\beta}{\alpha}, \tag{123}$$

$$\leq \beta\exp\left(\frac{-\alpha\eta T}{4}\right)\|\mathbf{w}_0 - \mathbf{w}_\star\|^2 + \frac{2\delta}{\alpha} + \frac{14\varepsilon\beta}{\alpha}. \qquad \blacksquare$$

# E   ADDITIONAL EXPERIMENTAL DETAILS

## E.1   STANDARD DEVIATION OF CIFAR RESULTS

| Task | Optimizer | Avg | Std |
|------|-----------|-----|-----|
| DN10 | ADAMW | 92.6 | 0.08 |
| DN10 | ALIG | 95.0 | 0.16 |
| DN10 | AMSGRAD | 91.7 | 0.25 |
| DN10 | DFW | 94.6 | 0.22 |
| DN10 | L4ADAM | 90.8 | 0.09 |
| DN10 | L4MOM | 91.9 | 0.17 |
| DN10 | SGD | 95.1 | 0.21 |
| DN10 | YOGI | 92.1 | 0.38 |
| DN100 | ADAMW | 69.5 | 0.54 |
| DN100 | ALIG | 76.3 | 0.14 |
| DN100 | AMSGRAD | 69.4 | 0.41 |
| DN100 | DFW | 73.2 | 0.29 |
| DN100 | L4ADAM | 60.5 | 0.64 |
| DN100 | L4MOM | 62.6 | 1.98 |
| DN100 | SGD | 76.3 | 0.22 |
| DN100 | YOGI | 69.6 | 0.34 |
| WRN10 | ADAMW | 92.1 | 0.34 |
| WRN10 | ALIG | 95.2 | 0.09 |
| WRN10 | AMSGRAD | 90.8 | 0.31 |
| WRN10 | DFW | 94.2 | 0.19 |
| WRN10 | L4ADAM | 90.5 | 0.09 |
| WRN10 | L4MOM | 91.6 | 0.24 |
| WRN10 | SGD | 95.3 | 0.31 |
| WRN10 | YOGI | 91.2 | 0.27 |
| WRN100 | ADAMW | 69.6 | 0.51 |
| WRN100 | ALIG | 75.8 | 0.29 |
| WRN100 | AMSGRAD | 68.7 | 0.70 |
| WRN100 | DFW | 76.0 | 0.24 |
| WRN100 | L4ADAM | 61.7 | 2.17 |
| WRN100 | L4MOM | 61.4 | 0.86 |
| WRN100 | SGD | 77.8 | 0.13 |
| WRN100 | YOGI | 68.7 | 0.47 |

Table 4: *Test Accuracy (%) on CIFAR including standard deviations. Each experiment was run three times.*

