# OpenReview forum: "Training Neural Networks for and by Interpolation"
_ICLR.cc/2020/Conference — Reject_

### Official Review · AnonReviewer1 · 2019-10-22
**Official Blind Review #1**

**Rating:** 1

**Review:**

This paper proposes a new gradient descent methods for training deep neural network which can take the adaptive step size with only one hyper-parameter to tune -- the maximum learning rate -- and achieve comparable results to stochastic gradient descent (SGD) on various tasks and models. In order to achieve that, they develop a stochastic extension of the Polyak step-size for the non-convex setting, namely the adaptive learning-rates for interpolation with gradients (ALI-G), in which the minimal value of the objective loss is set to 0 due to interpolation in neural networks and the learning rates are clipped by a chosen maximal value. The problem is formulated clearly, and the review on the Polyak step-size and related works are well done. Another main contribution of the paper is to provide the convergence guarantees for ALI-G in the convex setting where the objective loss is Lipschitz-continuous (Theorem 1 in the paper). Their theorem also takes into account the error in the estimate of the minimal value of the objective loss. In addition, they derive the connections between  ALI-G and SGD and show that compared to SGD, ALI-G take into consideration that the objective loss is non-negative and set the loss to 0 when it is negative. They perform empirical study to compare their algorithm with other methods including Adagrad, Adam, DFW, L4Adam and SGD on learning a differentiable neural computer, object recognition, and a natural language processing task. Their experimental results show that ALI-G performance is comparable with that of SGD with schedule learning rate.

Overall, this paper could be an interesting contribution. However, some points in the theory and experiments need to be verified. I weakly reject this paper, but given these clarifications in an author response, I would be willing to increase the score.

For the algorithm and theory, there are some points that need to be verified and further clarification on novelty:

1. When there are regularization such as the weight decay regularization, the minimal objective loss will not be 0. In such cases, Theorem 1 in the paper only guarantees that ALI-G reaches an objective loss less than or equal to a multiple of the estimate error \epsilon of the true minimal objective loss. It cannot guarantee that the objective loss reached by ALI-G can converge to the true minimall objective loss. Furthermore, when training neural networks with, for example, the weight decay regularization, often times the value of the regularization loss, i.e.  the estimate error \epsilon, is not small. Therefore, the upper bound given by Theorem 1 is rather loose.

2.  The paper mentions that when no regularization is used, ALI-G and Deep Frank-Wolfe (DFW) are identical algorithms. The difference between the two algorithms are when regularization is used. However, given my concern for Theorem 1 above, the convergence of ALI-G and advantage of ALI-G over DFW in this setting is questionable, and the claim that “ALI-G can handle arbitrary (lower-bounded) loss functions” also needs to be verified.

For the experiments, the following should be addressed:
1. In the experiment with the differentiable neural computers, even though ALI-G obtains better performance for a large range of \eta, its best objective loss is still worse than RMSProp, L4Adam, and L4Mom.

2. Given the merit of Theorem 1 is that the convergence guarantee takes into account the estimate error of the minimal objective loss, an ablation study that compares ALI-G with other methods in the same setting with and without regularization are needed. For example, it would be more convincing if similar results to those in Table 2 or 3 but without regularization are provided and discussed.

3. In Section 5.5, given that ALI-G and DFW are related, why is there no result for DFW in Figure
4.?

4.  As the paper mentions, AProx algorithm and ALI-G are related, why is there no comparison with AProx in the experiments?
Things to improve the paper that did not impact the score:
1. In all experiments, the performance differences between ALI-G and competitive methods are small. Thus, error bars are needed for these results.

2. In Table 3, the gap between SGD and ALI-G can be significantly different on different architectures. For example, in CIFAR-100 experiments, while ALI-G achieves the same result as SGD with the DenseNet, ALI-G’s performance on Wide ResNet is much worse than SGD. Do you have any explanation for this?

3. How is ALI-G compared with the methods proposed in the paper “Stochastic Gradient Descent with Polyak's Learning Rate” (https://arxiv.org/abs/1903.08688)


The following paper also proved the convergence of adaptive gradient methods for nonconvex optimization:
Dongruo Zhou*, Yiqi Tang*, Ziyan Yang*, Yuan Cao, Quanquan Gu. On the Convergence of Adaptive Gradient Methods for Nonconvex Optimization.

-------------------------------------------------------------
After rebuttal:
I noticed that the authors:

1. did not compare with SGD or Adam with good hyperparameters (https://arxiv.org/abs/1907.08610)

2. did not test on the large scale datasets, e.g., imagenet

3. the proposed algorithm is not as good as SGD on most numerical experiments.

For the above reasons, I think this paper should be rejected.


**Experience Assessment:**

I have read many papers in this area.

**Review Assessment: Checking Correctness Of Derivations And Theory:**

I carefully checked the derivations and theory.

**Review Assessment: Checking Correctness Of Experiments:**

I carefully checked the experiments.

**Review Assessment: Thoroughness In Paper Reading:**

I read the paper thoroughly.

---

> ### Author Response · Authors · 2019-11-14
> **Response to Review #1 (1/2)**
>
> We thank the reviewer for their comments, which we answer below:
>
> * “When there are regularization such as the weight decay regularization, the minimal objective loss will not be 0.”; “Therefore, the upper bound given by Theorem 1 is rather loose. “; “given my concern for Theorem 1 above, the convergence of ALI-G and advantage of ALI-G over DFW in this setting is questionable [...].”
>
> We believe that this concern stems from a misunderstanding of our method. As described in the paragraph “Regularization” in section 2.1, we express the regularization as a constraint on the feasible set rather than a soft penalty in the objective function. This allows us to realistically assume that even for regularized models, the minimal objective value is close to zero. This is verified in practice in our experiments, where regularized models achieve near zero objective value while remaining within the feasible set. Furthermore, please note that our theoretical results have also been proven for the constrained setting, i.e., no assumption is made regarding the feasible region other than the standard ones (convexity, availability of efficient projection algorithms).
>
> * “the claim that “ALI-G can handle arbitrary (lower-bounded) loss functions” also needs to be verified.”
>
> The above answer shows that we can use arbitrary loss functions that have an infimum of 0.  In the more general case, if the loss function $\ell$ is just lower-bounded, we can recover the previous case of an infimum of 0 by using the function shifted by a constant $\ell - \inf \ell$, which does not change the solution of the problem (using $inf$ is well-defined because the function is real-valued and lower-bounded).
>
> * “Given the merit of Theorem 1 is that the convergence guarantee takes into account the estimate error of the minimal objective loss, an ablation study that compares ALI-G with other methods in the same setting with and without regularization are needed. For example, it would be more convincing if similar results to those in Table 2 or 3 but without regularization are provided and discussed.”
>
> In Tables 2 and 3, we focus on generalization performance, which is why we keep the regularization activated for a fair comparison between all methods. In contrast, sections 5.1 and 5.5 do not use l2 regularization because they focus on training performance, and section 5.3 does not either because the corresponding model from (Conneau et al., 2017) did not originally use any regularization. Thus our set of experiments arguably cover varied settings in terms of regularization.
>
> * “ In the experiment with the differentiable neural computers, even though ALI-G obtains better performance for a large range of \eta, its best objective loss is still worse than RMSProp, L4Adam, and L4Mom.”
>
> By being more careful about numerical precision (a bottleneck for very accurate optimization on this task), we have been able to improve the performance of ALI-G. ALI-G, the L4 methods and RMSProp obtain approximately the same performance -- the difference is insignificant due to the limit of numerical precision on single-float numbers. We have updated the results and added a comment about numerical precision.
>
> * “In Section 5.5, given that ALI-G and DFW are related, why is there no result for DFW in Figure 4.?”
>
> This is because DFW does not support the use of the cross-entropy loss, and therefore its performance cannot be fairly compared on a cross-entropy objective function.
>
> * “As the paper mentions, AProx algorithm and ALI-G are related, why is there no comparison with AProx in the experiments?”
>
> The aProx algorithm coincides with the ALI-G algorithm when the exponential decay rate of aProx is set to zero. In the general case, this decay rate is an additional hyper-parameter that has to be chosen by the user (which adds to the tuning burden of tuning hyper-parameters). This work shows that this hyper-parameter can be avoided since ALI-G provides good performance in practice without needing to tune it — this is also visible in our convergence results that do not require it either. In other words, ALI-G shows that a simplified version of aProx with one fewer hyper-parameter provides good performance in practice. Finally, we point out that given the extent of our experiments and our available hardware, tuning a method with two hyper-parameters may require multiple weeks of computation if at all feasible.

---

> > ### Author Response · Authors · 2019-11-14
> > **Response to Review #1 (2/2)**
> >
> > * “In all experiments, the performance differences between ALI-G and competitive methods are small. Thus, error bars are needed for these results.”
> >
> > All results of Table 3 (which seem to have the largest variance among our set of experiments) are averaged over three independent runs. The standard deviation of these runs are reported in Appendix E.1 of the revised paper. One can check that the difference of performance between ALI-G and other adaptive methods (in particular AMSGrad and L4 methods) is statistically significant.
> >
> > * “In Table 3, the gap between SGD and ALI-G can be significantly different on different architectures. For example, in CIFAR-100 experiments, while ALI-G achieves the same result as SGD with the DenseNet, ALI-G’s performance on Wide ResNet is much worse than SGD. Do you have any explanation for this?”
> >
> > The WRN architecture has a significantly larger number of parameters than the DN architecture (8M vs 2M). As a consequence, optimizers tend to overfit more easily on WRN, and SGD benefits more from the implicit regularization of its manually tuned schedule with many iterations at a large learning-rate. In contrast, ALI-G tends to converge fast on this problem which limits its implicit regularization. This could probably be mitigated by using a finer grain tuning of the l2 regularization, or “locking” the learning-rate of ALI-G to its maximal value for a large number of steps before letting it decay.
> >
> > * “ How is ALI-G compared with the methods proposed in the paper “Stochastic Gradient Descent with Polyak's Learning Rate” (https://arxiv.org/abs/1903.08688)”.
> >
> > In contrast to ALI-G which uses only $\ell_{z_t}(w_t)$ and $\nabla \ell_{z_t}(w_t)$ (loss function and its gradient for a sample / mini-batch) to compute its learning-rate, the aforementioned work requires $f(w_t)$ (objective function over entire dataset) to compute its learning-rate. In addition, since they do not do exploit the interpolation setting (and the fact that regularization can be expressed as a constraint), they also require the optimal objective function $f*$ in advance. We have added this comparison to the revised version of the paper.

---

### Official Review · AnonReviewer2 · 2019-10-23
**Official Blind Review #2**

**Rating:** 6

**Review:**

This paper proposes a new adaptive learning rate method which is tailored to the optimization of deep neural networks. The motivating observation is that over-parameterized DNNs are able to interpolate the training data (i.e. they are able to reach near-zero training error). This enables application of the Polyak update rule to stochastic updates and a simplification by assuming a zero minimal training loss. A number of proofs for convergence in various convex settings are provided, and empirical evaluation on several benchmarks demonstrates (a) ability to optimize complex architectures, (b) performance improvements over, and (c) performance close to manually tuned SGD learning rates.

I vote for accepting this paper. The approach is well-motivated, the method is described clearly and detail, and the experiments support the paper's claims well. What I would still like to see are a few additional details regarding the experimental protocol. In particular, did you train a single or multiple models for each result that is reported? Do different runs start from the exact same weight initialization? What condition was used to stop the training? The results in section 5.2. are all very close to each other, and it would be helpful to have a sense of the variability of the different methods. The graphs in Figure 4 do look like the models did not converge yet.

Generally, it would be nice to examine the behavior of the method in cases where the neural network is underparameterized or is otherwise unable to effectively interpolate the training data. Does the method lead to divergence in this case, or is it subpar to other methods? I think section 2.2. could benefit from a short motivational introduction; on the first read, I was not clear about the purpose of introducing the Polyak step size as it is not mentioned explicitly in the text leading to it.

**Experience Assessment:**

I do not know much about this area.

**Review Assessment: Checking Correctness Of Derivations And Theory:**

I did not assess the derivations or theory.

**Review Assessment: Checking Correctness Of Experiments:**

I assessed the sensibility of the experiments.

**Review Assessment: Thoroughness In Paper Reading:**

I made a quick assessment of this paper.

---

> ### Author Response · Authors · 2019-11-14
> **Response to Review #2**
>
> We thank the reviewer for their comments, which we answer below:
>
> * “did you train a single or multiple models for each result that is reported?”
>
> For all datasets except CIFAR, we train a single model. On the CIFAR tasks (the ones with the largest possible variance), we now provide the standard deviation over three independent runs in the appendix — these remain lower than 0.3 for ALI-G and SGD. We have added these experimental details in the revised submission.
>
> * “Do different runs start from the exact same weight initialization?”
>
> For all experiments, the model starts with a random initialization that may be different: the random seed is itself randomly generated for each experiment. We believe that such tasks tend to be relatively robust to using different random seeds (see answer above).
>
> * “What condition was used to stop the training?”, “The graphs in Figure 4 do look like the models did not converge yet”
>
> In each experiment, the training loop terminates after a fixed number of iterations/ epochs, set as follows:
> 10k steps for DNC (following (Graves et al., 2016);
> 160 epochs for WRN on SVHN (following (Zagoruyko & Komodakis, 2016));
> 20 epochs for SNLI (following Conneau et al., 2017);
> 200 epochs for WRN on CIFAR (following ( Zagoruyko & Komodakis, 2016));
> 300 epochs for DN on CIFAR (following (Huang et al., 2017)).
>  We have added these experimental details in the revised submission.
>
> * “Does the method lead to divergence in this case, or is it subpar to other methods?”
>
> When the interpolation assumption is not satisfied, the learning-rate of ALI-G falls back to its maximal value, and then ALI-G behaves exactly as SGD. Indeed, rephrasing Proposition 1, ALI-G can be seen as an extension of SGD which exploits the fact that the loss is non-negative to adapt its learning-rate. When the interpolation property is satisfied, this knowledge is useful for convergence of the algorithm, because the lower bound of zero corresponds to the minimum value that can be attained. When the interpolation property is not satisfied, the lower bound is too loose to be useful and therefore ALI-G behaves exactly like non-adaptive SGD.
>
> * “I think section 2.2. could benefit from a short motivational introduction”
>
> We thank the reviewer for the useful suggestion, we have updated the paper accordingly.

---

### Official Review · AnonReviewer3 · 2019-11-04
**Official Blind Review #3**

**Rating:** 6

**Review:**

This paper addresses designing and analyzing an optimization algorithm. Like SGD, it maintains a low computational cost per optimization iteration, but unlike SGD, it does not require manually tuning a decay schedule. This work uses the interpolation property (that the empirical loss can be driven to near zero on all samples simultaneously in a neural network) to compute an adaptive learning rate in closed form at each optimization iteration, and results show that this method produces state-of-the-art results among adaptive methods. I can say the paper was very well written and easy to follow along/understand. Prior work seems comprehensive, and the intuitive comparisons to the prior methods were also useful for the reader.

My current decision is a weak accept, for a well-written paper, thorough results including meaningful baselines and numerous hyperparameter searches, and a seemingly high-impact tool. Some concerns are listed as follows:
1-	Convergence was only discussed in the stochastic convex setting, which seems limiting because we rarely deal with convex problems in problems requiring neural networks.
2-	Regularization of the weights during the optimization is dealt with by projecting onto the feasible set of weights, but it seems like there are other types of losses that don’t necessarily to go 0. For example, terms in the objective such as entropy seem worrisome.
3-	One detail that I did not fully follow along with is section 3.1. How does Theorem 1 (Regarding convexity) related to the “each training sample can use its own learning rate without harming progress on the other ones” and/or “allow the updates to rely on the stochastic estimate rather than the exact”?
4-	Unfortunately, I am not an expert in this particular area, so I’m not confident about the novelty. For example, the difference between L4 and this is stated to be the utilization of the interpolation policy (which just sets f*=0) and the maximal learning rate, and the stated benefit of convergence guarantees in stochastic convex settings seems poor since most problems will not be convex anyway. More generally, it seems like all details of the algorithm came from elsewhere, although the presented synthesis of ideas does have clear benefits.

After reading the author response:
-I'm fine with points 4 and 3.
-My feelings about 1 are still the same.
-My comment on 2 wasn't about cross-entopy loss, but rather other types of objectives that people are often interested in optimizing (such as max-ent RL, where we aim for maximizing rewards as well as maximizing entropy of the policy), in which case, it's not clear to me how we could apply this optimizer.
-My decision stays as a weak accept

**Experience Assessment:**

I have read many papers in this area.

**Review Assessment: Checking Correctness Of Derivations And Theory:**

I assessed the sensibility of the derivations and theory.

**Review Assessment: Checking Correctness Of Experiments:**

I assessed the sensibility of the experiments.

**Review Assessment: Thoroughness In Paper Reading:**

I read the paper at least twice and used my best judgement in assessing the paper.

---

> ### Author Response · Authors · 2019-11-14
> **Response to Review #3**
>
> We thank the reviewer for their detailed comments, which we answer below:
>
> * “[...] but it seems like there are other types of losses that don’t necessarily to go 0. For example, terms in the objective such as entropy seem worrisome.”
>
> We reassert that the cross-entropy loss is well-handled by our framework: it is lower bounded by zero, and when the model can “separate” the data with an arbitrary margin in logits space, its value can get arbitrarily close to zero. In addition, the proximity between the actual value and the ideal value of zero is captured by the interpolation tolerance $\varepsilon$. For a practical example of the cross-entropy loss going near zero, we refer the reviewer to Figure 4, which plots the evolution of the cross-entropy loss during training: final values obtained by ALI-G are indeed very close to zero.
>
> * “How does Theorem 1 (Regarding convexity) related to the ‘each training sample can use its own learning rate without harming progress on the other ones’ and/or ‘allow the updates to rely on the stochastic estimate rather than the exact’.”
>
> These statements refer to the fact that, as shown in Theorem 1, by assuming interpolation ALI-G provably converges while using only $\ell_{z_t}(w_t)$ and $\nabla \ell_{z_t}(w_t)$ (stochastic estimation per sample) to compute its learning-rate. In contrast, the Polyak step-size, which does not exploit interpolation, would use $f(w_t)$ and $\nabla f(w_t)$ to compute the learning-rate (exact / non-stochastic computation over all training samples). This constitutes a major computational advantage of ALI-G over the usual Polyak step-size. We have added the above clarification in the new version of the paper.
>
> * On novelty:
>
> Beyond our theoretical analysis — which required a significant amount of work by itself, it is important to note that ALI-G requires only one hyper-parameter instead of four for L4. Since hyper-parameter sweeps are often the computational bottleneck for applied machine learning (the cost of extensive tuning grows exponentially with the number of hyper-parameters), this constitutes a considerable advantage for practical applications.
>
> In addition, L4 is sensitive to noise,which makes it very difficult to tune: we observed in practice that L4 sometimes fails to learn anything or even diverges in some cases — similar issues have been reported in (Vaswani et al., 2019b). In contrast, ALI-G consistently provides reliable optimization in the interpolation setting at considerably less tuning cost.

---

### Official Review · AnonReviewer4 · 2019-11-04
**Official Blind Review #4**

**Rating:** 6

**Review:**

Thanks for the responses and my concerns seem to be addressed. But since I do not know much about this area, I would like to stick to my initial rating 6.
==================================================================================================
This work designs a new optimization SGD algorithm named ALI-G for deep neural network with interpolation property.
This algorithm only has a single hyper-parameter and doesn’t have a decay schedule. The authors provide the convergence guarantees of ALI-G in the stochastic convex setting as well as the experiment results on four tasks.This paper shows state-of-the-art results but I still have have two concerns.

My main concern is that the performances of other SGD algorithms may be potentially better than the results showed in section 5 because it is not easy to tune the parameter. It would be better if the authors can tune the hyper-parameter more carefully. Take figure 3 as an example. The settings where step size bigger than 1e+1 can hardly shows something, because the step-size is too big for SGD to converge. The settings where step size smaller 1e-2 can also hardly shows something, because the step-size is too small and the experiments only runs 10k steps. It would be better if authors can do more experiments in the settings where step size is from 1e-3 to 1e+0. Moreover, the optimal step size of different optimization algorithms may differ a lot. It would be much more fair if the authors can compare the best performance of different algorithms.

Another concern is that the authors only give the convergence rate of ALI-G in section 3 but haven’t make any comparisons. For example, it would be better if the authors can show that ALI-G has better convergence result than vanilla SGD without decay schedule.


**Experience Assessment:**

I do not know much about this area.

**Review Assessment: Checking Correctness Of Derivations And Theory:**

I did not assess the derivations or theory.

**Review Assessment: Checking Correctness Of Experiments:**

I did not assess the experiments.

**Review Assessment: Thoroughness In Paper Reading:**

I made a quick assessment of this paper.

---

> ### Author Response · Authors · 2019-11-14
> **Response to Review #4**
>
> We thank the reviewer for their comments, which we answer below:
> * “My main concern is that the performances of other SGD algorithms may be potentially better than the results showed in section 5 because it is not easy to tune the parameter.”
>
> We respectfully disagree with the statement that SGD is not a strong baseline for our experimental setting. For each of the SVHN, SNLI and CIFAR tasks, SGD uses a schedule of the learning-rate that has been tuned by an experienced practitioner on the particular combination of dataset and architecture. More specifically, these schedules come from (Zagoruyko & Komodakis, 2016), (Conneau et al., 2017), and (Huang et al., 2017), each of which tuned the learning-rate schedule to obtain then state-of-the-art results with their model. To the best of our knowledge, this constitutes the strongest baseline available in the literature.
>
> We do agree that in general SGD is difficult to tune, and this is a major motivation for this work. We hope to show that ALI-G alleviates this empirical difficulty in the interpolation setting by providing good performance while requiring little tuning.
>
> * “The settings where step size bigger than 1e+1 can hardly shows something, because the step-size is too big for SGD to converge. The settings where step size smaller 1e-2 can also hardly shows something, because the step-size is too small and the experiments only runs 10k steps. It would be better if authors can do more experiments in the settings where step size is from 1e-3 to 1e+0”.
>
> We have run the experiment with a finer grid-search and restricted the range of learning-rates in the plot, please see Figure 3 in the updated paper.
>
> * “ It would be much more fair if the authors can compare the best performance of different algorithms.”
>
> While the main point of Figure 3 is to evaluate robustness to hyper-parameters, the best performance can still be compared by comparing the best value per row. We hope that the figure in the revised paper makes this more visible by having fewer cells per row.
>
> * “it would be better if the authors can show that ALI-G has better convergence result than vanilla SGD without decay schedule.”
>
> We are not aware of convergence guarantees for vanilla stochastic gradient descent with a constant learning-rate. To the best of our knowledge, the closest result to it in the interpolation setting is SGD with line-search (Vaswani et al., 2019b), which comes at the cost of (i) additional forward passes and (ii) up to four hyper-parameters instead of one for ALI-G.
>
> In more detail, SGD with line-search provides the following convergence results:
> Smooth setting: O(1/T) in Theorem 1 of (Vaswani et al., 2019b), to be compared to O(1/T) in Theorem 4 of this work.
> Smooth and strongly-convex setting: O(exp(-K T)) in Theorem 2 of (Vaswani et al., 2019b), to be compared to O(exp(-K T / 8)) in Theorem 8 of this work.
>
> It is also worth noting that, without assuming smoothness or strong convexity, the rate of convergence of O(1/sqrt(T)) in Theorem 1 is optimal among first-order methods in Nemirovski / Nesterov sense (Theorem 3.13 in Bubeck, 2015).

---

### Decision · Program_Chairs · 2019-12-19

**Decision:**

Reject

**Comment:**

This paper uses the interpolation property to design a new optimization algorithm for deep learning, which computes an adaptive learning-rate in closed form at each iteration. The authors also analyzed the convergence rate of the proposed algorithm in the stochastic convex optimization setting. Experiments on several benchmark neural networks and datasets verify the effectiveness of the proposed algorithm. This is a borderline paper and has been carefully discussed. The main objection of the reviewers include: (1) The interplay between regularization and the interpolation property is not clear; and (2) the proposed algorithm  is no better than SGD in any of the benchmarks except one, where SGD's learning rate is set to be a constant. After the author response, this paper still does not gather sufficient support. So I encourage the authors to improve this paper and resubmit it to future conference.